# Simulating multiple variability in spatially resolved transcriptomics with scCube

Jingyang Qian[1,2,5], Hudong Bao[1,5], Xin Shao [1,2], Yin Fang [3], Jie Liao [1,2], Zhuo Chen [3], Chengyu Li [1,2], Wenbo Guo[1,2], Yining Hu[1,2], Anyao Li[1,2], Yue Yao[1,2], Xiaohui Fan [1,2,4] ✉ & Yiyu Cheng [1,2] ✉

A pressing challenge in spatially resolved transcriptomics (SRT) is to benchmark the computational methods. A widely-used approach involves utilizing simulated data. However, biases exist in terms of the currently available simulated SRT data, which seriously affects the accuracy of method evaluation and validation. Herein, we present scCube (https://github.com/ZJUFanLab/scCube), a Python package for independent, reproducible, and technology-diverse simulation of SRT data. scCube not only enables the preservation of spatial expression patterns of genes in reference-based simulations, but also generates simulated data with different spatial variability (covering the spatial pattern type, the resolution, the spot arrangement, the targeted gene type, and the tissue slice dimension, etc.) in reference-free simulations. We comprehensively benchmark scCube with existing single-cell or SRT simulators, and demonstrate the utility of scCube in benchmarking spot deconvolution, gene imputation, and resolution enhancement methods in detail through three applications.

The emergence and rapid development of spatially resolved transcriptomics (SRT) technologies offer an unprecedented opportunity to understand the cell composition, molecular architecture, and functional details of tissues at spatial levels[1]. Various emerging cutting-edge technologies, including imaging-based[2–4], spatial barcoding next-generation sequencing (NGS)-based[5–7], and laser capture micro-dissection (LCM)-based[8,9] approaches, have been successfully employed across diverse biological fields, such as developmental biology[10], cancer[11], and neuroscience[12,13].

As SRT data have become available, there has been an increasing advancement of computational tools for downstream analysis. Currently, more than 300 software packages have been developed for various spatial transcriptomic data analysis[14], such as spatially variable gene detection[15–18], cell type deconvolution[19–26], unmeasured gene imputation[27–31], spatial domain identification[18,32,33], and spatial cell-cell interaction inference[34,35]. While these computational methods are

often based on reasonable assumptions, it is difficult to benchmark and evaluate their performance without gold standards.

Currently, a wildly-used approach for assessing the performance of computational methods involves utilizing simulated SRT data constructed from scRNA-seq data. However, variations exist in terms of the scRNA-seq data selection and the approach used to generate simulated SRT data across different benchmarking experiments. In most cases, the simulated data are constructed specifically based on the assumptions that underlie the computational method being evaluated, which may introduce bias in the evaluation process. Additionally, the lack of reproducibility in the description of most simulation steps can impede the potential reuse of these methods by other researchers. Several methods have recently emerged to generate synthetic SRT data. For example, SRTsim simulates the gene expression counts with a count model inferred from the reference data and then allocates the simulated counts to spatial locations in the synthetic

[1]College of Pharmaceutical Sciences, Zhejiang University, Hangzhou 310058, China. [2]National Key Laboratory of Chinese Medicine Modernization, Innovation Center of Yangtze River Delta, Zhejiang University, 314100 Jiaxing, China. [3]College of Computer Science and Technology, Zhejiang University, Hangzhou 310013, China. [4]Zhejiang Key Laboratory of Precision Diagnosis and Therapy for Major Gynecological Diseases, Women's Hospital, Zhejiang University School of Medicine, Hangzhou 310006, China. [5]These authors contributed equally: Jingyang Qian, Hudong Bao. ✉e-mail: fanxh@zju.edu.cn; chengyy@zju.edu.cn

data[36]. scDesign3 applies a probabilistic model which can incorporate diverse cell covariates to simulate the gene expression changes across spatial locations[37]. However, both SRTsim and scDesign3 make the simulations based on the SRT data, leading to inherent limitations of the generated data, such as limited gene detection and low cellular resolution. Furthermore, scDesign3 cannot simulate the SRT data with new spatial locations specified by users, which greatly restricts its application; SRTsim, though, can assign simulated expression counts for each new location based on its $k$ nearest neighboring locations measured in the reference data, this strategy may fail to preserve the spatial expression patterns of genes when there is a large discrepancy in tissue shape or cell type spatial distribution between the reference and synthetic data, such as the slices of different brain donors in the DLPFC datasets by Maynard et al.[38]. Therefore, a general simulation framework that can simulate the independent, reproducible, and various SRT data to better facilitate the development of spatial transcriptomic data analysis methods is still in demand.

To this end, we introduce scCube, an SRT simulator for simulating multiple spatial variability in spatial resolved transcriptomics and generating unbiased simulated SRT data. Based on the variational autoencoder (VAE) framework, scCube can simulate the gene expression profiles of different cell (or spot) populations in scRNA-seq (or SRT) data. Next, the spatial distribution patterns for specific populations can then be generated by the reference-based or reference-free strategy. We evaluated the simulation performance of scCube with existing single-cell or SRT simulators across various real SRT datasets, and demonstrated the utility of scCube in three benchmarking applications. The results indicated that scCube is a user-friendly framework to simulate unbiased SRT data, enabling researchers to benchmark and evaluate different computational methods more lightly and accurately.

## Results

### Design concept of scCube
The workflow of scCube consists of two components: 1) gene expression simulation, and 2) spatial pattern simulation (Fig. 1). In the gene expression simulation step, scCube applies a variational autoencoder (VAE) model[39] to simulate the gene expression profiles of cell (or spot) populations within the characterized clustering spaces[20] (Fig. 1a; Methods). One of the main advantages of VAE is the ability to generate new data points compared with classical autoencoders, which is highly in line with the design requirements for a simulator. In the current version, scCube includes about 300 trained models of various tissues derived from four human and mouse scRNA-seq atlas (Tabula Muris[40], Tabula Sapiens[41], MCA[42], and HCL[43]) as well as eight high-quality SRT datasets (Supplementary Data 1). These models can be conveniently employed through the Python package by users to generate the new gene expression profiles of specific tissues. Additionally, scCube is also equipped to accommodate external scRNA-seq or SRT datasets provided by users to train new models. Next, scCube will simulate unique spatial distribution patterns for the specific cell (or spot) populations set by users. In this step, scCube provide two strategies, reference-based and reference-free, to simulate the various spatial patterns in real SRT data. In the reference-based simulations, for each cell (or spot) population, scCube constructs a mapping between the cells (or spots) in generated data and the positions in the spatial reference using the optimal transport algorithm, and then maps the generated cells (or spots) to positions with the maximum likelihood of spatial origin (Fig. 1b; Methods). In the reference-free simulations, scCube further provides two strategies. Users can either choose to generate the random spatial patterns with the default spatial autocorrelation function, or flexibly simulate the highly customized complex spatial patterns by combining different types as well as numbers of basis patterns provided by scCube (Fig. 1c; Methods). By combining the simulated gene expression profiles and spatial patterns, scCube can generate unbiased SRT data corresponding to diverse spatial

variations, such as the spatial pattern continuity of cell populations, the resolution for spot-based SRT data, the number and type of targeted genes for imaging-based SRT data, the dimension of the tissue slice, and so on (Supplementary Fig. S1). What's more, scCube also builds in a set of visualization functions, which helps users investigate simulated datasets more intuitively.

### Simulation performance evaluation of the reference-based strategy of scCube
We first comprehensively compared the simulation performance of the reference-based strategy of scCube with other existing simulators, including two SRT simulators, SRTsim[36] and scDesign3[37], as well as six single-cell simulators, scDesign2[44], SymSim[45], ZINB-WaVE[46], and three variations of Splatter[47] (Splat, Splat Simple, and Kersplat). To achieve this, 29 real SRT datasets from seven different tissues generated by six sequencing technologies that include 10X Visium, ST, Stereo-seq, MERFISH, STARmap, and 10X Xenium were utilized as the benchmark datasets (Supplementary Fig. S2 and Supplementary Data 2). The simulation performance was evaluated by the Pearson correlation coefficient ($PCC_{GEV}$) and mean absolute error ($MAE_{GEV}$) values between the gene expression vector for each gene across spatial positions in real and simulated data, as well as the Pearson correlation coefficient ($PCC_{GBM}$) values between the gene expression value for each gene from the simulated data's spatial locations predicted by the two generalized boosted regression models (GBMs) trained on the real data and simulated data separately (Methods). As shown in Fig. 2a, b and Supplementary Figs. S3-4, scCube outperformed other SRT and single-cell simulators over most of the seven benchmark datasets from different tissues, with the highest average $PCC_{GEV}$ and $PCC_{GBM}$ values as well as the lowest average $MAE_{GEV}$ values (except for the human DLPFC 10X Visium and zebrafish embryo Stereo-seq datasets) between the real spatial expression patterns of all genes and the corresponding simulated spatial expression patterns. Consistently, scCube also exhibited excellent simulation performance over all tissue slices of the human DLPFC 10X Visium and mouse hypothalamus MERFISH datasets (Supplementary Fig. S5). We further demonstrated the spatial expression patterns of several representative marker genes in the real and simulated data. As illustrated in Fig. 2c and Supplementary Figs. S6–12, scCube accurately preserved the spatial expression patterns of these marker genes across different datasets. In contrast, although the two other SRT simulators, SRTsim and scDesign3, also achieved this relatively successfully, confirming the superiority of SRT simulators in simulating spatially resolved transcriptomics compared with single-cell simulation methods, scCube was superior to both of them with the highest $PCC_{GEV}$ and $PCC_{GBM}$ values.

We next evaluated the overfitting issues of scCube and other two SRT simulators in three scenarios (Supplementary Data 3). In the first two scenarios, we constructed the training and test data from single datasets using two different data splitting strategies (countsplit[48] and random splitting), respectively; while in the third scenario, we considered a more general case where the training and test data come from different tissue slices of the same sample or experiment, rather than just being split from the same datasets. Each simulator was trained on the training data, and the simulated data generated by different simulators was subsequently compared with the test data for overfitting evaluation. Although all three SRT simulators accurately simulated the spatial expression patterns of genes in the test data in the first two scenarios and didn't have significant overfitting issues (Supplementary Fig. S13–28), scCube exhibited a unique strength in the third scenario, i.e., its scalability to simulate SRT data with a large discrepancy in tissue shape or cell type spatial distribution from the spatial reference (Supplementary Fig. S29). To be specific, we selected two tissue slices with different spatial distributions of cell types from the mouse hypothalamus MERFISH dataset as an example, where the "Bregma +0.06" and "Bregma −0.29" slices are near the anterior and

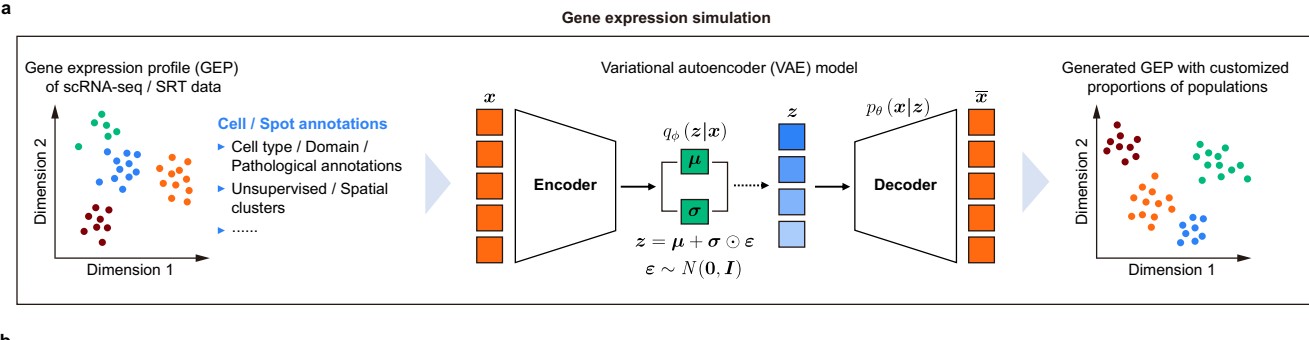

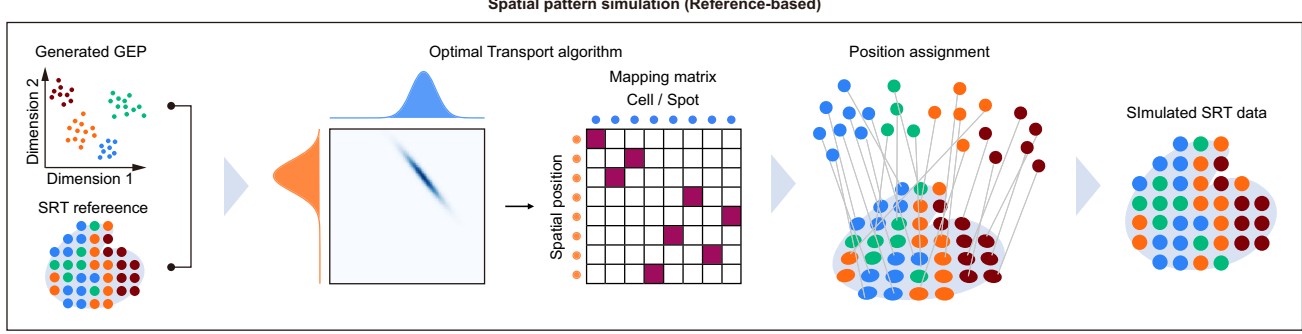

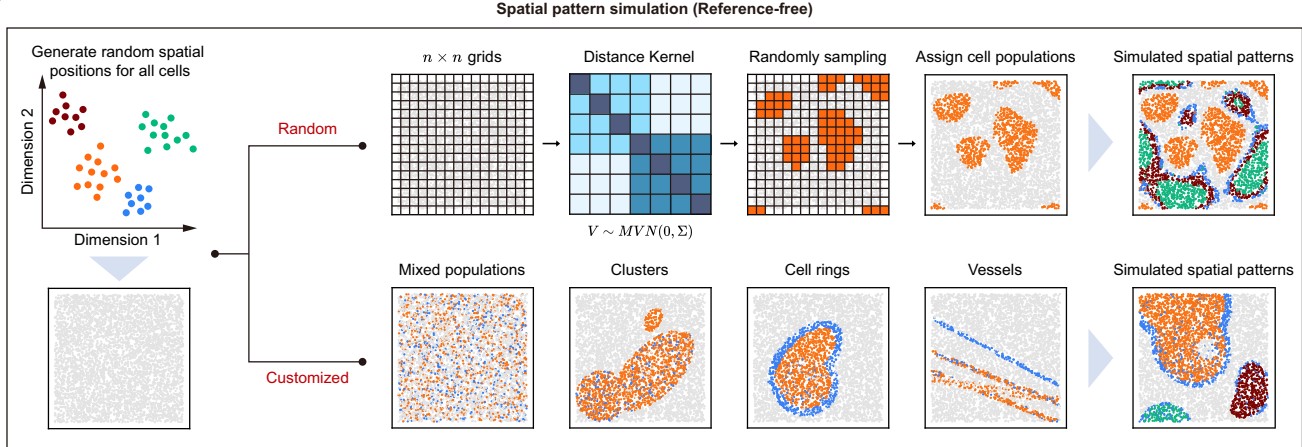

**Fig. 1 | Schematic workflow of scCube. a** Conceptual framework of gene expression simulation step with scCube. A variational autoencoder (VAE) model is applied to simulate the gene expression profiles of cell (or spot) populations in scRNA-seq (or SRT) data. **b, c** Conceptual framework of spatial pattern simulation step with scCube. In the reference-based simulations (**b**), a mapping between the cells (or spots) in generated data and the positions in the spatial reference is first constructed using the optimal transport algorithm, and the generated cells (or spots) then are mapped to positions with the maximum likelihood of spatial origin. In the reference-free simulations (**c**), the spatial positions with the same number as the cells (or spots) in the generated data are randomly generated first. Then, scCube generates the random spatial patterns with the default spatial autocorrelation function, or the customized complex spatial patterns by combining different types as well as numbers of basis patterns.

posterior positions, respectively (Fig. 2d). For both scCube and SRTsim, we simulated the gene expression over the coordinates of the "Bregma −0.29" slice using the "Bregma +0.06" data as the spatial reference. As illustrated in Fig. 2e and Supplementary Fig. S30, scCube can still accurately simulate the spatial expression patterns of marker genes on the "Bregma -0.29" slice when using the different tissue slice (the "Bregma +0.06") as the spatial reference. In contrast, SRTsim suffered from significant overfitting and failed to preserve these spatial expression patterns well. One possible reason is that for each location on the "Bregma -0.29" slice, SRTsim assigned the simulated expression counts based on the expression counts of its $k$ nearest neighboring locations on the "Bregma +0.06" slice. Since the spatial distribution of cell types between these two slices is quite different, this strategy may lead to the spatial expression patterns of marker genes on the simulated data being heavily influenced by the spatial reference. Similar

results were reproduced on the human DLPFC 10X Visium dataset. We used the "Slice 151507" from donor 1 as the spatial reference and applied scCube and SRTsim to simulate the gene expression over the coordinates of the "Slice 151676" from donor 3 respectively (Supplementary Fig. S31a), and the results showed that SRTsim still failed to exactly preserve these spatial expression patterns due to the large discrepancy in tissue shape between slices (Supplementary Fig. S31b). All these results further demonstrated the superiority and general applicability of scCube in simulating the SRT data without significant overfitting issues.

In addition, we further evaluated the biologically interpretable of the scCube-simulated data by comparing the benchmark results of three types of SRT computational methods (spot deconvolution, gene imputation, and spatial domain identification) on the real data and scCube-simulated data. As shown in Supplementary Fig. S32, the

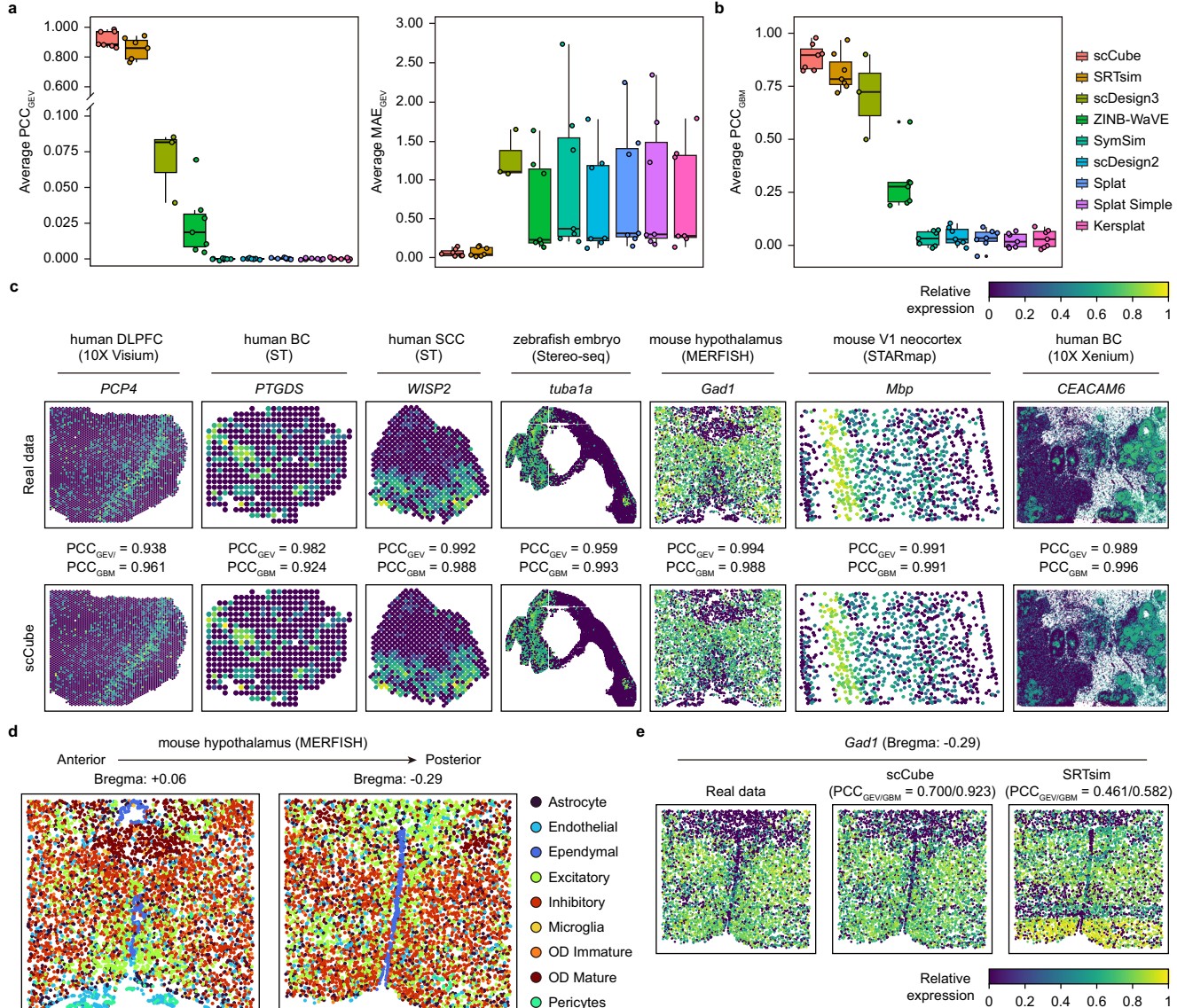

**Fig. 2 | Superior performance of scCube over other simulators. a, b** Performance comparison of scCube with other SRT and single-cell simulators across seven benchmark datasets. The simulation results for all genes of scDesign3 are not provided in the Stereo-seq data, 10X Visium data, and ST data due to speed constraints of the training step. Data are presented as boxplots (minima, 25th percentile, median, 75th percentile, and maxima). The number of data points are 7 for scCube, SRTsim, ZINB-WaVE, SymSim, scDesign2, Splat, Splat Simple, and Kersplat, and 3 for scDesign3. **c** The spatial expression patterns of representative marker genes in the real data and the simulated data generated by scCube across seven benchmark datasets. **d** Two tissue slices with different spatial distributions of cell types from the mouse hypothalamus dataset, where the "Bregma +0.06" slice is near the anterior position and the "Bregma −0.29" slice is near the posterior position. **e** The spatial expression patterns of *Gad1* (the marker of inhibitory) in the real data (Bregma: −0.29) and the simulated data generated by scCube and SRTsim. Source data are provided as a Source Data file.

deconvolution result of each spot deconvolution method was highly consistent when using the real or scCube-simulated data, as were the benchmark results of all methods. Conversely, this consistency was not observed when using random-simulated data (Supplementary Fig. S33). Similar results were reproduced on gene imputation and spatial domain identification computational methods (Supplementary Figs. S34 and 35), indicating that the simulated SRT data generated by scCube preserves the biological characteristics of the real data.

We also surveyed the effect of utilizing different types of cell (or spot) annotations in the gene expression simulation step on the reference-based simulation performance of scCube. As shown in Supplementary Fig. S36, we selected four human breast cancer 10X Visium datasets and performed the reference-based simulation of scCube using the pathological annotation provided by the authors, three unsupervised clustering results with different clustering

resolutions (implemented by Seurat[29], resolution = 0.3, 0.5, and 0.7), and one spatial domain clustering result (implemented by BayesSpace[49]) as the spot annotations, respectively. As shown in Supplementary Figs. S37–39, scCube accurately simulated the spatial expression patterns of genes in all four SRT datasets, whether using the pathological labels or the clustering results as the spot annotations, and the average PCC values between the real spatial expression patterns of all genes and the corresponding simulated spatial expression patterns are both greater than 0.9 (Supplementary Fig. S40). What's more, the cell type composition with each spot of the scCube-simulated data generated with different spot annotations was highly consistent with the real data, with the average PCC values greater than 0.98 (Supplementary Fig. S41). These results suggest that scCube is robust to the choice of cell (or spot) annotations in the gene expression simulation step.

Finally, we systematically provided the execution time of scCube when achieving relatively stable simulation performance across seven benchmark datasets of different sizes (Supplementary Fig. S42). Notably, as shown in Supplementary Fig. S43, the execution time of scCube mainly concentrated in the training step. Therefore, if there are trained models provided by scCube matching the target SRT data, users can directly download the corresponding models to generate the simulated data within a very short period of time without additional training steps.

### Simulating multiple variability in SRT data by the reference-free strategy with scCube

In this section, we illustrated the high flexibility of scCube in simulating multiple variability in SRT data with the reference-free strategy. In brief, we further subdivided the variability in SRT data into two types: gene expression variability and spatial pattern variability. For the former, scCube can select a user-specified number or type of genes based on the simulated gene expression profiles to simulate the variability of targeted genes in imaging-based SRT data. For the latter, scCube can generate simulated spatial patterns of cell types varying in number, type, and dimension with the default spatial autocorrelation function, and further simulate the variability of spatial patterns within cell types (such as cell subtypes), as well as the variability of resolution and spot arrangement in spot-based SRT data. We demonstrated it in detail by the following three examples.

**Simulation of the variability of spatial patterns of cell types.** A basic function of scCube is its ability to generate the spatial distributions of cell types with diverse patterns. In scCube, there are two vital parameters, $\lambda$ and $\delta$, which control the fuzzy degree and the continuity of generated spatial patterns, respectively. The value of $\lambda$ ranges from 0 to 1, and larger values will tend to form clearer spatial patterns; the value of $\delta$ should be greater than 0, and larger values will tend to form spatial patterns with greater connectivity (Fig. 3a). Users thus can apply scCube to generate specific patterns by setting and combining different $\lambda$ and $\delta$ values for simulating the spatial architecture of cells in different tissues.

Another function of scCube is its ability to set the number of spatial patterns and simulate spatial patterns for only a specific set of cell types (Fig. 3c). This feature of scCube allows users to better simulate different situations in real tissues, such as spatial domains for specific cell types or the uniform colocalization distribution of several cell types[50]. In the example, we applied scCube to generate spatial patterns for only three cell types (astrocyte, brain pericyte, and neuron) (Fig. 3b). As expected, the Moran's *I* values for the spatial distribution of these three cell types were much higher than those of the remaining cell types (Fig. 3c), supporting the accuracy of scCube in spatial pattern simulation. We further examined the spatial distribution of each cell type as well as the spatial expression pattern of its top 50 marker genes, as illustrated in Fig. 3d and Supplementary Fig. S44a, b, astrocyte, brain pericyte, and neuron showed distinct spatial patterns, while the other four cell types without specific spatial patterns followed similar uniform distributions. Unsurprisingly, the top 50 marker genes of astrocyte, brain pericyte, and neuron also had significantly higher Moran's *I* values, which was consistent with the results of the spatial distribution of cell types (Supplementary Fig. S44c). Moreover, users can apply this function to specify whether a specific cell type has the spatial distribution pattern, which in turn achieves the manual specification of the types of spatial expression patterns for the marker genes of this cell type. As shown in Fig. 3e, we used scCube to generate two simulated SRT datasets, in which astrocytes were manually specified to be present and absent spatial patterns, respectively. Compared with Dataset 2, the marker genes of astrocyte in Dataset 1 showed obvious spatial expression patterns and

had significantly higher Moran's *I* values (Fig. 3f and Supplementary Fig. S45).

The additional function of scCube is its ability to simulate SRT data of three-dimensional tissues (Fig. 4a, b). Recently, computational methods for three-dimensional structures of tissue construction by integrating multi-slices are gradually emerging, such as PASTE[51], PRECAST[52], and STalign[53]. However, real three-dimensional SRT datasets for benchmarking these methods are still scarce. A widely used trade-off is to utilize the two-dimensional SRT data from a series of continuous slices, and then compare the alignment results for the same domain between them. Although this strategy is applied in the performance evaluation of most methods, it may introduce too much noise since slices are not exactly adjacent to each other. In contrast, the simulated three-dimensional SRT data generated by scCube is unbiased, and thus suitable enough as the ground truth in benchmarking datasets. Furthermore, scCube provides a "split" function that can split the three-dimensional SRT data into a user-defined number of two-dimensional SRT data along any axis of coordinates of the simulated data (Fig. 4c). With this function, users can investigate the spatial variability along a series of adjacent two-dimensional slices, such as the spatial distribution of specific cell types (endothelial cell in Fig. 4c) and spatial expression pattern of its marker genes. In addition, by jointly utilizing simulated three-dimensional SRT data and a series of derived two-dimensional SRT data, users thus can evaluate the computational methods for three-dimensional tissues reconstruction more accurately.

**Simulation of the variability of spatial patterns within cell types.** scCube also provides a separate function to consider the heterogeneity within cell types and generate the spatial patterns of cell subtypes flexibly. As shown in Fig. 5a, b, we first applied scCube to generate the spatial patterns for each cell type from a scRNA-seq dataset, in which macrophages contain two subtypes. Subsequently, for these two macrophage subtypes, we simulated two different types of spatial patterns to disregard or consider the heterogeneity within cell types (Fig. 5c). Compared with the former, where the spatial distribution of each subtype followed a random distribution, the latter further simulated a specific spatial pattern for each subtype to better account for intra-cell-type gene expression variation in space (Fig. 5d).

**Simulation of the variability of resolution and spot arrangement.** scCube further provides the optional parameters for setting the number of cells per spot ($n$) and the arrangement and neighborhood structure of all spots, which makes it possible to simulate spot-based SRT data with different resolutions and even different technology platforms. As shown in Fig. 5e, the resolution of the simulated spatial patterns decreased with the value of the parameter $n$ gradually increased. In contrast, the average number of cells per spot increased and approximated the setting value of $n$. We next demonstrated the ability of scCube in simulating spot-based SRT data with different spot arrangements corresponding to the three current mainstream technology platforms, including Slide-seq[6], 10X Visium, and ST[5]. As shown in Supplementary Fig. S46a, the simulated Slide-seq data followed a random neighborhood structure of spots, while the simulated 10X Visium and ST data were opposite, having regular hexagonal and square neighborhood structures of spots, respectively. Additionally, the average numbers of cells per spot in simulated Slide-seq, 10X Visium, and ST data was 2.34, 10.5, and 34.01, respectively, which is consistent with the real data. Specifically, we examined the spatial distribution of astrocyte and the spatial expression pattern of its marker gene *Slc6a11* and found that they were consistent well with each other in all three types of simulated data, further suggesting the robustness of scCube in simulation (Supplementary Fig. S46b).

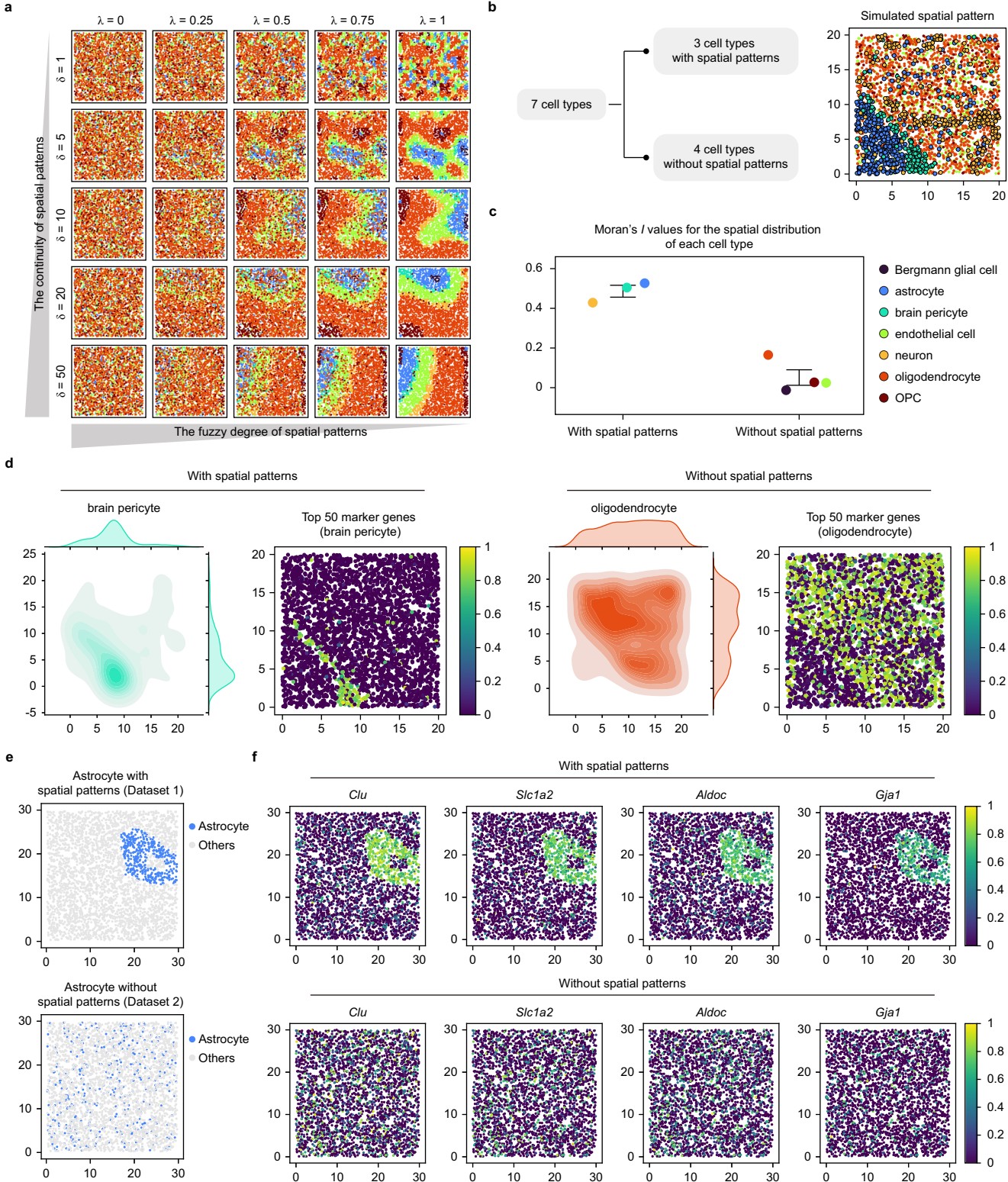

**Fig. 3 | scCube generates spatial patterns with diverse types and numbers.**
**a** The spatial patterns with different fuzzy degree and continuity generated by
scCube. **b** Illustration of simulating specified number of spatial patterns with
scCube. Three cell types (astrocyte, brain pericyte, and neuron) are set as "with
spatial patterns" while other four cell types are set as "without spatial patterns" and
follow uniform distributions. **c** Moran's *I* values for the spatial distribution of dif-
ferent cell types. Data are presented as the mean values ± standard deviation. The
number of data points are 3 and 4 for cell types with and without spatial patterns,

respectively. **d** The spatial distributions of brain pericyte (with spatial patterns) and
oligodendrocyte (without spatial patterns) and the spatial expression patterns for
top 50 marker genes of them. **e** Two simulated SRT datasets generated by scCube,
in which astrocytes were manually specified to be present (Dataset 1, top) and
absent (Dataset 2, bottom) spatial patterns. **f** Boxplots of Moran's *I* values for spatial
expression patterns of marker genes of astrocytes in two simulated datasets.
Source data are provided as a Source Data file.

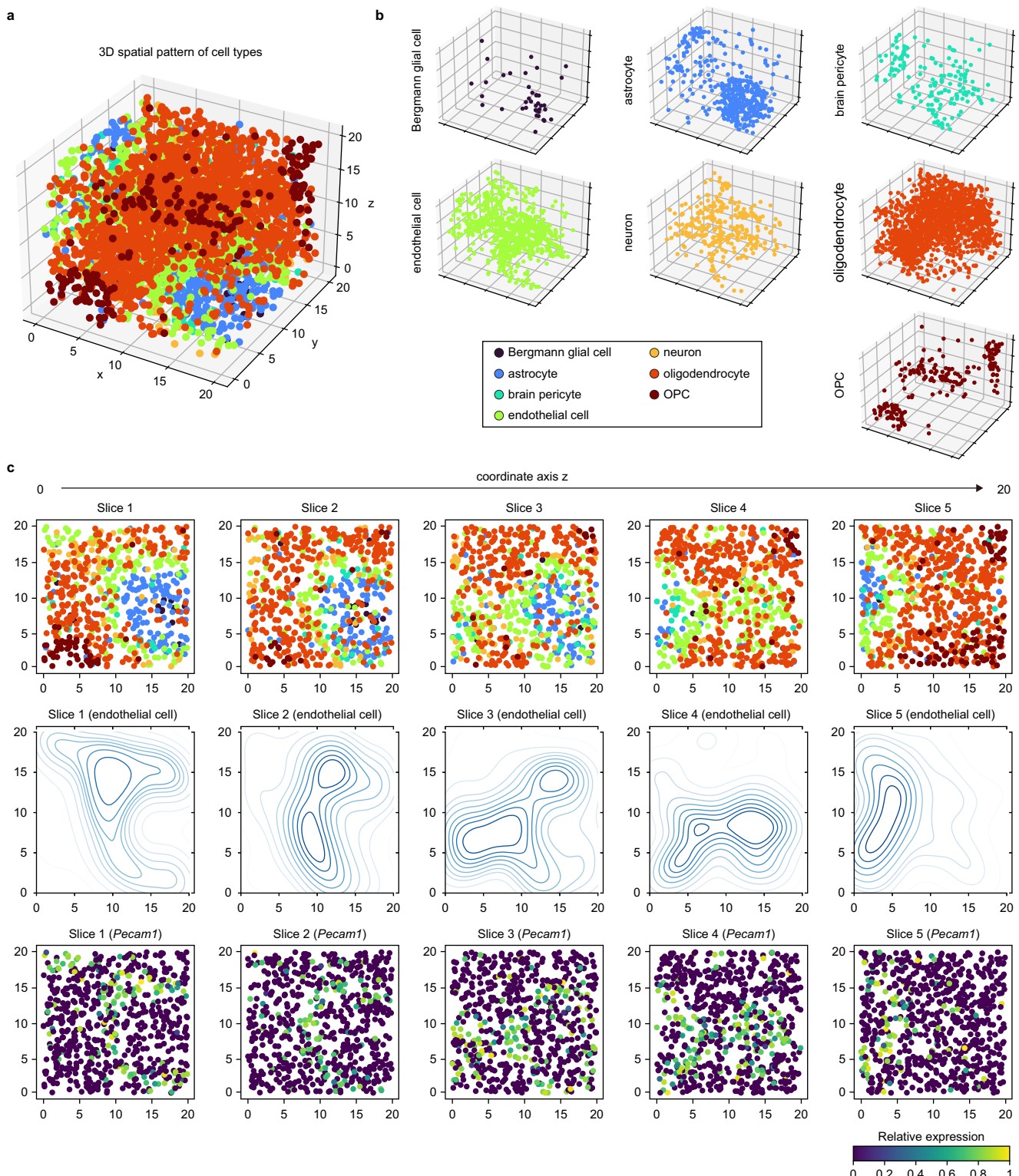

**Fig. 4 | scCube generates spatial patterns of three-dimensional tissues. a, b** The three-dimensional spatial pattern generated by scCube (**a**) and the spatial distribution of each cell type (**b**). **c** A series of two-dimensional spatial patterns of slices split from the three-dimensional spatial pattern along the z axis of coordinates. The overview of spatial patterns (top), the spatial distribution of endothelial cell (middle), and the spatial expression pattern of *Pecam1* (bottom) are showed respectively. Source data are provided as a Source Data file.

## Simulating biologically interpretable spatial patterns by the reference-free strategy with scCube

In the reference-free spatial pattern simulation, scCube further considers generating more interpretable spatial patterns in a customized manner. Users can first simulate a series of biologically interpretable spatial basis patterns, including unstructured mixed or clustered cell populations, cell rings encircling tissue structures, and some external structures such as vessels[54], and then flexibly generate highly customized complex spatial patterns by combining different types as well as numbers of these basis patterns (Fig. 6a, b).

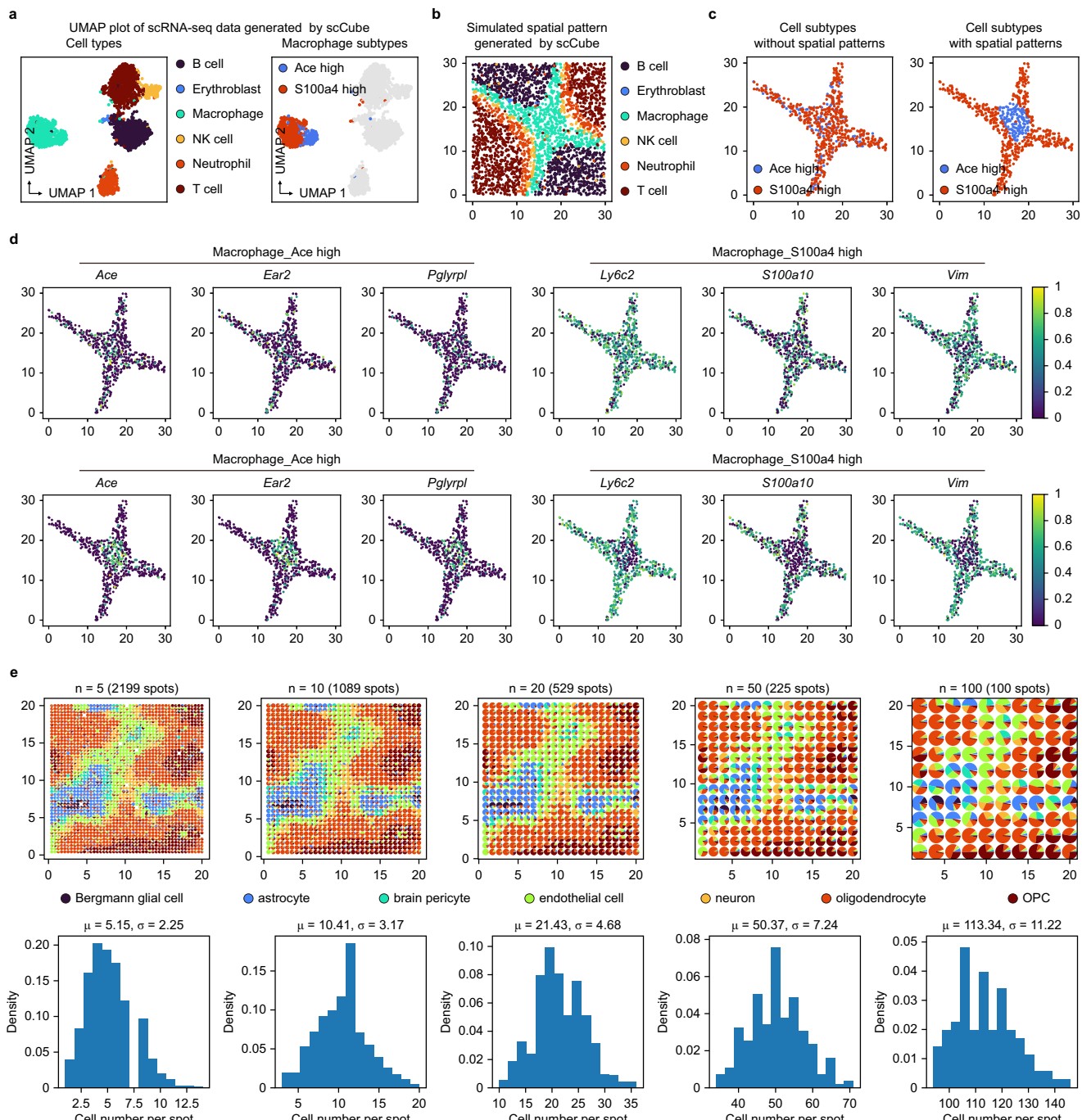

**Fig. 5 | Using scCube to simulate the variability of spatial patterns within cell types and the variability of resolution and spot arrangement in spot-based SRT data. a** The simulated gene expression data of a scRNA-seq dataset generated by scCube, in which macrophages contain two subtypes. **b** The simulated spatial pattern of cell types generated by scCube. **c** The spatial distributions of macrophage subtypes in the settings of "cell subtypes without spatial patterns" (left) and "cell subtypes with spatial patterns" (right). **d** The spatial expression patterns of marker genes for macrophage subtypes in the settings of "cell subtypes without spatial patterns" (top) and "cell subtypes with spatial patterns" (bottom). **e** The simulated spatial patterns with diverse resolutions generated by scCube (top) and frequency histograms of the cell number per spot (bottom). Source data are provided as a Source Data file.

We applied scCube to the simulation of the tumor-immune microenvironment of 3 triple negative breast cancer (TNBC) patients, which corresponded to three archetypical subtypes of tumor-immune interactions: cold, mixed, and compartmentalized, respectively as a detailed example (Fig. 6c). It has been reported that the cold subtype shows the low immune infiltration, the mixed subtype shows the high mixability between tumor and immune cells, and the compartmentalized subtype shows a series of regions comprised predominantly of

either immune or tumor cells[55]. As shown in Fig. 6c, the spatial patterns simulated by scCube were very similar to the corresponding tumor-immune microenvironments. Furthermore, unlike spaSim[54], which can only generate spatial pattern images, scCube also combines this customized reference-free spatial pattern simulation step with the gene expression simulation step to generate the corresponding gene expression profiles that match the simulated spatial patterns, which more comprehensively simulates the real SRT data (Fig. 6d, e).

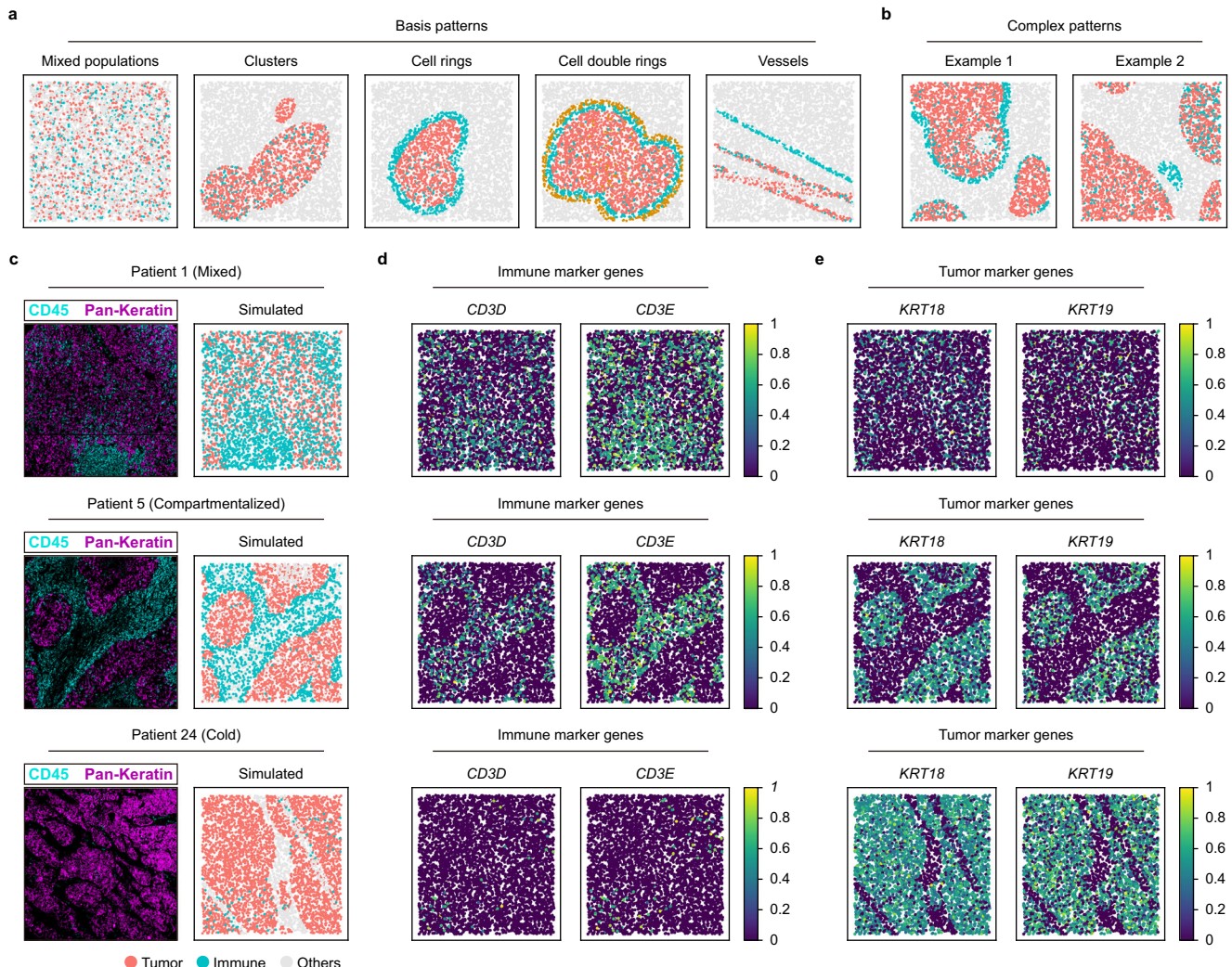

**Fig. 6 | Using scCube to flexibly simulate the biologically interpretable spatial patterns. a**, **b** The biologically interpretable spatial basis (**a**) and complex (**b**) patterns simulated by scCube. **c** The real and scCube-simulate tumor-immune microenvironments of mixed, compartmentalized, and cold subtypes of TNBC. **d**, **e** The spatial expression patterns of immune (**d**) and tumor (**e**) marker genes in the scCube-simulated SRT data.

## Using scCube to benchmark spot deconvolution methods

We began by demonstrating the utility of scCube in benchmarking the spot deconvolution methods. To this end, we first generated the simulated SRT data with single-cell resolution from a mouse brain scRNA-seq dataset, and then simulated six spot-based SRT datasets with different resolutions by merging the different numbers of cells into specific spots (Fig. 7a). The cell type composition of each spot in these six simulated datasets was known and can be used as the ground truth when evaluating the deconvolution performances of different methods. A total of nine spot deconvolution methods were benchmarked, including Cell2location[19], DestVI[21], DSTG[22], RCTD[25], Seurat[29], spatialDWLS[24], SPOTlight[23], Stereoscope[26], and Tangram[27]. The accuracy of the cell type composition of each spot was evaluated. We first explored the deconvolution performance of each method on the simulated SRT data with the high resolution (n = 5), and as shown in Fig. 7b, Cell2location, DestVI, and spatialDWLS, had the top 1, 2, and 3 rankings average Pearson correlation coefficient (PCC) and Spearman rank-order correlation coefficient (SRCC) values (0.930/0.913, 0.900/0.870, and 0.889/0.869, respectively), followed by RCTD (0.814/0.771), Seurat (0.809/0.778), Tangram (0.750/0.652), SPOTlight (0.734/0.597), Stereoscope (0.574/0.523), and DSTG (0.533/0.431). Consistently, the average root mean squared error (RMSE) and Jensen-Shannon divergence (JS) values of Cell2location, DestVI, and spatialDWLS were 0.063/0.166, 0.095/0.264, and 0.115/0.194, respectively, also lower than those of the other methods.

We further examined the effect of the resolution on the performance of spot deconvolution, and benchmarked these methods on a series of simulated datasets with decreasing resolutions (n = 10, 20, 30, 50, and 100) (Fig. 7c and Supplementary Fig. S47). Interestingly, we found that RCTD, Tangram, and Cell2location were relatively robust to the resolution of spot-based SRT data, the differences between the maximum and minimum of the average PCC and SRCC values were 0.034/0.025, 0.047/0.039, and 0.070/0.095 (Fig. 7c, d). In contrast, other methods showed a preference for specific resolutions. As the resolution decreased, the performance of DestVI, Seurat, and Spotlight degraded, while spatialDWLS performed the opposite (Fig. 7c, d). Additionally, Stereoscope achieved higher average PCC/SRCC values as well as lower average RMSE/JS values when n = 30 and 50 (Fig. 7c). We also used the accuracy score (AS) proposed by Li et al.[56] to evaluate the performance of these methods on the six simulated datasets comprehensively, and found that Cell2location outperformed other methods with the highest average AS value (0.954), followed by spatialDWLS (0.778), RCTD (0.718), DestVI (0.676), Stereoscope (0.565), Tangram (0.472), Seurat (0.440), SPOTlight (0.255), and DSTG (0.144) (Fig. 7e).

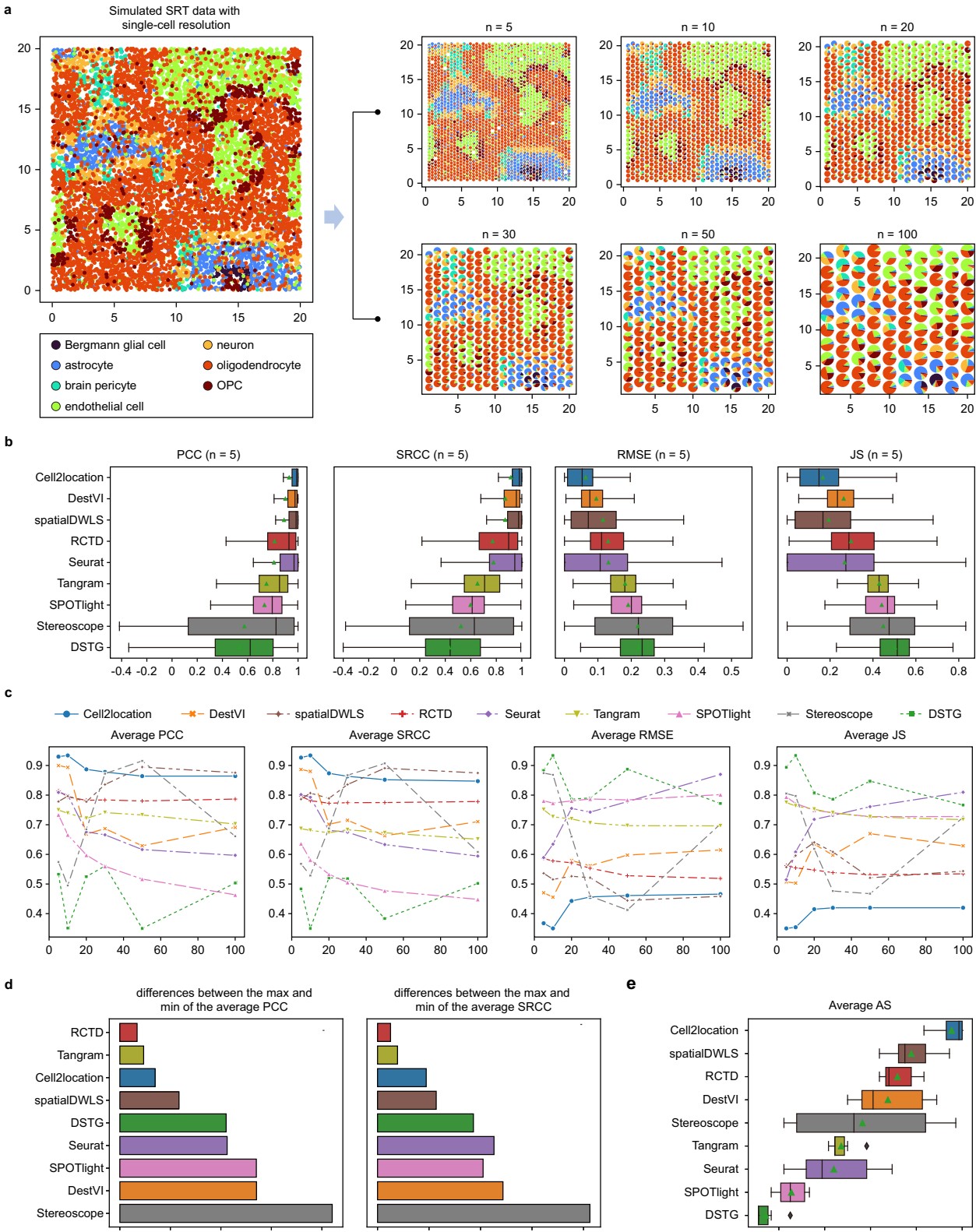

**Fig. 7 | Using scCube to benchmark spot deconvolution methods. a** Illustration of simulating SRT data with diverse resolutions by scCube. A total of six simulated SRT datasets with different resolutions are generated as the benchmarking datasets. **b** Boxplots of PCC, SRCC, RMSE, and JS values of the nine spot deconvolution methods for the high-resolution ($n = 5$) simulated SRT dataset. Data are presented as boxplots (minima, 25th percentile, median, 75th percentile, and maxima). The number of data points are 2013 for each method. **c** The average PCC, SRCC, RMSE, and JS values of the nine spot deconvolution methods for all six simulated SRT datasets. **d** Bar plots of the differences between the maximum and minimum of the average PCC (left) and SRCC (right) values of the nine spot deconvolution methods across all six simulated SRT datasets. **e** Boxplots of average AS values of the nine spot deconvolution methods for all six simulated SRT datasets. Data are presented as boxplots (minima, 25th percentile, median, 75th percentile, and maxima). The number of data points are 6 for each method. Source data are provided as a Source Data file.

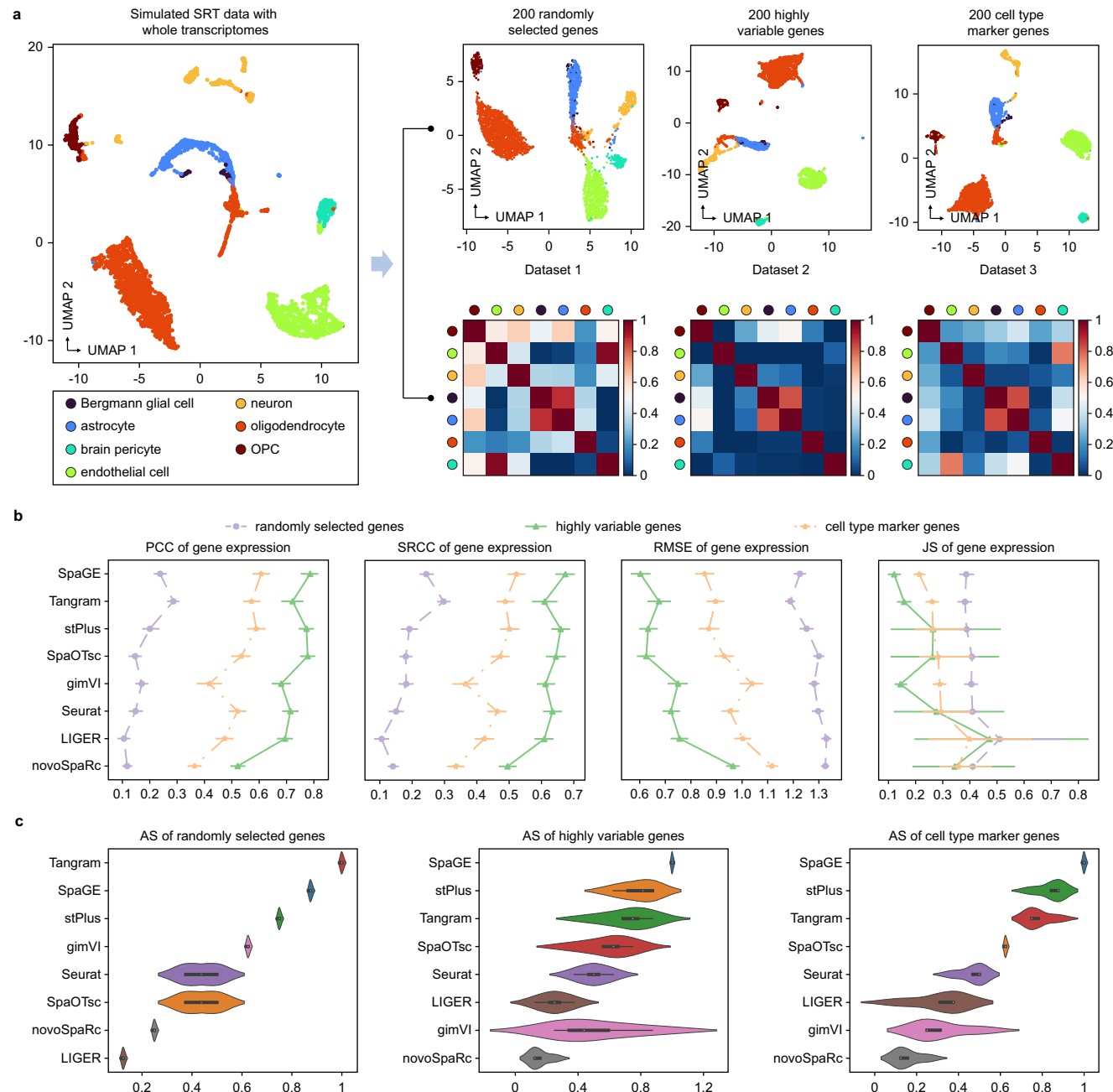

**Fig. 8 | Using scCube to benchmark gene imputation methods. a** Illustration of simulating SRT data with diverse types of targeted genes by scCube. A total of three simulated SRT datasets with different types of targeted genes are generated as the benchmarking datasets. **b** The PCC, SRCC, RMSE, and JS values of the eight gene imputation methods for all three simulated SRT datasets. Data are presented as mean values ± 95% confidence intervals. The number of data points are 200 for each type of targeted genes of each method. **c** Boxplots of AS values of the eight gene imputation methods for all three simulated SRT datasets. Data are presented as boxplots (minima, 25th percentile, median, 75th percentile, and maxima). The number of data points are 3 for each type of targeted genes of each method. Source data are provided as a Source Data file.

## Using scCube to benchmark gene imputation methods

We next demonstrated the utility of scCube in benchmarking the gene imputation methods. In this application, we focused on examining the effect of the quality of detected transcriptomes on the performance of gene imputation. Specifically, we first generated the simulated SRT data with whole transcriptomes from a mouse brain scRNA-seq dataset, and then simulated three imaging-based SRT datasets with different types of genes by selecting 200 random genes (Dataset 1), 200 highly variable genes (Dataset 2), and 200 cell type marker genes (Dataset 3), respectively (Fig. 8a). The gene expression value of each gene in these three simulated datasets was known and can be used as

the ground truth when evaluating the imputation performances of different methods. We benchmarked eight gene imputation methods, including SpaGE[28], Tangram[27], stPlus[31], gimVI[57], Seurat[29], SpaOTsc[58], novoSpaRc[59], and LIGER[30] by calculating the PCC, SRCC, RMSE, and JS values between the expression vector of a gene in the ground truth and the expression vector for the same gene in the imputed results predicted by different methods. As illustrated in Fig. 8b, all methods performed well on Dataset 2 and 3 but poorly on Dataset 1. This is expected and reasonable, as a crucial step in all gene imputation methods is the integration of imaging-based SRT data and scRNA-seq reference using the shared genes between these two types of data.

Compared with randomly selected genes, highly variable genes and cell type marker genes cover more information and contain less noise, making it possible for these methods to perform better in gene imputation.

We further compared the performance of all methods on three simulated datasets, and found that SpaGE, stPlus, and Tangram performed better than other methods, with the average PCC and SRCC values ranking in the top 4 on all datasets (Supplementary Fig. S48). The performance of SpaOTsc was influenced by the Datasets, with the top 2 on Dataset 2 but the bottom 3 ranking on Dataset 1 (Supplementary Fig. S48). Moreover, the results of AS values were similar to those of PCC and SRCC values, where Tangram, SpaGE, and stPlus achieved the top 3 rankings on Dataset 1 (1, 0.875, and 0.75, respectively); SpaGE, SpaOTsc, and stPlus achieved the top 3 rankings on Dataset 2 (1, 0.781, and 0.719, respectively); SpaGE, stPlus, and Tangram achieved the top 3 rankings on Dataset 3 (1, 0.844, and 0.781, respectively) (Fig. 8c).

### Using scCube to benchmark resolution enhancement methods

In the last application, we demonstrated the utility of scCube in benchmarking two existing resolution enhancement methods, BayesSpace[49] and CARD[60], both of which constructing the high-resolution spatial maps of gene expression for spot-based SRT data. Specifically, we utilized scCube to simulate two pairs of spot-based SRT data with low (121 spots) and high (1089 spots) resolution, where each spot in the low-resolution data was generated by merging nine nearby spots in the high-resolution data (Fig. 9a). The cell type label of each cell in the simulated single-cell resolution SRT data and the gene expression profile of each spot in the simulated high-resolution spot-based SRT data were known and can be used as the ground truth for spot deconvolution and resolution enhancement, respectively. We first evaluated the spot deconvolution performance of CARD on both low- and high-resolution SRT data. As shown in Fig. 9b, CARD performed better on the low-resolution than on the high-resolution SRT data, the average PCC were 0.996 and 0.735, respectively. We further examined the predicted result of each spot of the high-resolution ST data, and found that CARD failed to accurately deconvolute the cell type compositions of the spots in the region located between two different cell type domains (Fig. 9c). The poor deconvolution performance on spots located at the boundary may be due to the simplistic assumption of CARD. The key idea of CARD to impute and construct high-resolution spatial maps for cell type composition and gene expression is modeling the spatial correlation structure as a multivariate normal distribution. This strategy leads to the cell type composition on the new locations being effectively represented as a weighted summation of the nonnegative cell-type proportions on the original locations[60], forming a "transition" state at the boundary. However, in some cases, the cell type composition of spots at the boundary of two large cell type domains often differs greatly from the composition of spots within the two domains because of the presence of specific new cell types (such as astrocyte in the simulated data), which is contrary to the assumption of CARD.

We also compared the accuracy of the high-resolution spatial maps of gene expression constructed by CARD and BayesSpace. Similar to the predicted results of cell type composition, the high-resolution spatial maps of gene expression constructed by CARD also presented an obvious "diffusion" trend, which were inconsistent with the ground truth. In contrast, the maps constructed by BayesSpace were more accurate (Fig. 9d and Supplementary Fig. S49). Additionally, as one might expect, both CARD and BayesSpace were better at predicting the high-resolution spatial maps of marker gene expression for cell types with large populations than for rare cell types. As shown in Fig. 9e, both CARD and BayesSpace achieved the highest average PCC values in predicting the high-resolution spatial maps of top 50

marker genes of endothelial cell (0.772 and 0.793, respectively), followed by oligodendrocyte (0.694 and 0.736, respectively), neuron (0.670 and 0.651, respectively), astrocyte (0.586 and 0.589, respectively), brain pericyte (0.523 and 0.547, respectively), OPC (0.501 and 0.480, respectively), and Bergmann glial cell (0.473 and 0.438 respectively).

## Discussion

In this study, we have presented a spatially resolved transcriptomics simulator, scCube, for simulating multiple spatial variability in spatial resolved transcriptomics and generating unbiased simulated SRT data. We have demonstrated the capability of scCube to preserve the spatial expression patterns of genes in real SRT data, and illustrated the utility of scCube in benchmarking diverse SRT-analysis computational methods, including spot deconvolution, gene imputation, and resolution enhancement.

scCube is designed to consist of two steps for SRT simulation: gene expression simulation and spatial pattern simulation. The former step is mainly based on the VAE model, which has become a popular tool for single-cell omics data modeling[20,61–63]. We have also demonstrated that the VAE model can accurately simulate various characteristics of the original data, such as the sparsity of scRNA-seq or SRT data (Supplementary Figs. S50–52). Meanwhile, we have provided about 300 trained models of various tissues, which are coupled with the scCube Python package, enabling users to simulate different spatial architectures of cells in real tissues more conveniently and quickly without the additional training steps. One potential limitation of scCube is that the ground truth provided by its simulated spot-based SRT data is restricted to the cell population level since it assigns each spot a cell population first when generating synthetic data. This may not be suitable for direct use in benchmarking studies of some SRT-analysis methods that operate at a finer resolution. However, benefiting from the robustness of scCube in cell population annotations selection (Supplementary Figs. S36–41), users can flexibly choose an appropriate granularity of cell population annotation during simulation. In addition, since the gene expression profiles generated by scCube remain at the spot level, users can also refine the annotation granularity of each spot through relevant downstream analysis (such as spot deconvolution).

In the spatial pattern simulation step, scCube provides two strategies: reference-based and reference-free. The reference-based strategy uses the SRT reference and aims to simulate the spatial expression pattern of genes across positions. The benchmark results across the real SRT data from diverse tissues generated by different sequencing platforms showed the superiority and broad scalability of scCube over other SRT or single-cell simulators. Moreover, we highlight the unique strength of scCube compared with SRTsim[36], a current state-of-the-art SRT simulator. Specifically, scCube first constructs a mapping between the cells (or spots) in simulated data and the positions in the spatial reference by solving an optimal transport problem[64] based on the gene expression, and then maps the cells (or spots) to positions with the maximum likelihood of spatial origin. This strategy effectively avoids the interference of external noise, such as tissue slice shape, cell type proportion, and distribution, etc., and thus can robustly simulate the spatial expression pattern of genes whether in the same or across different slices. In contrast, SRTsim relies heavily on the coordinate system (i.e., slice shape) of the spatial reference in simulation, which can indeed preserve the spatial expression pattern of genes when the shape of targeted simulation data is the same or sufficiently similar to that of the spatial reference. However, the accuracy of simulation drops dramatically when simulating data with a large discrepancy from the spatial reference (Fig. 2d, e and Supplementary Figs. S29–31).

For the reference-free simulations, scCube aims to generate random or customized spatial patterns for cell populations and combine

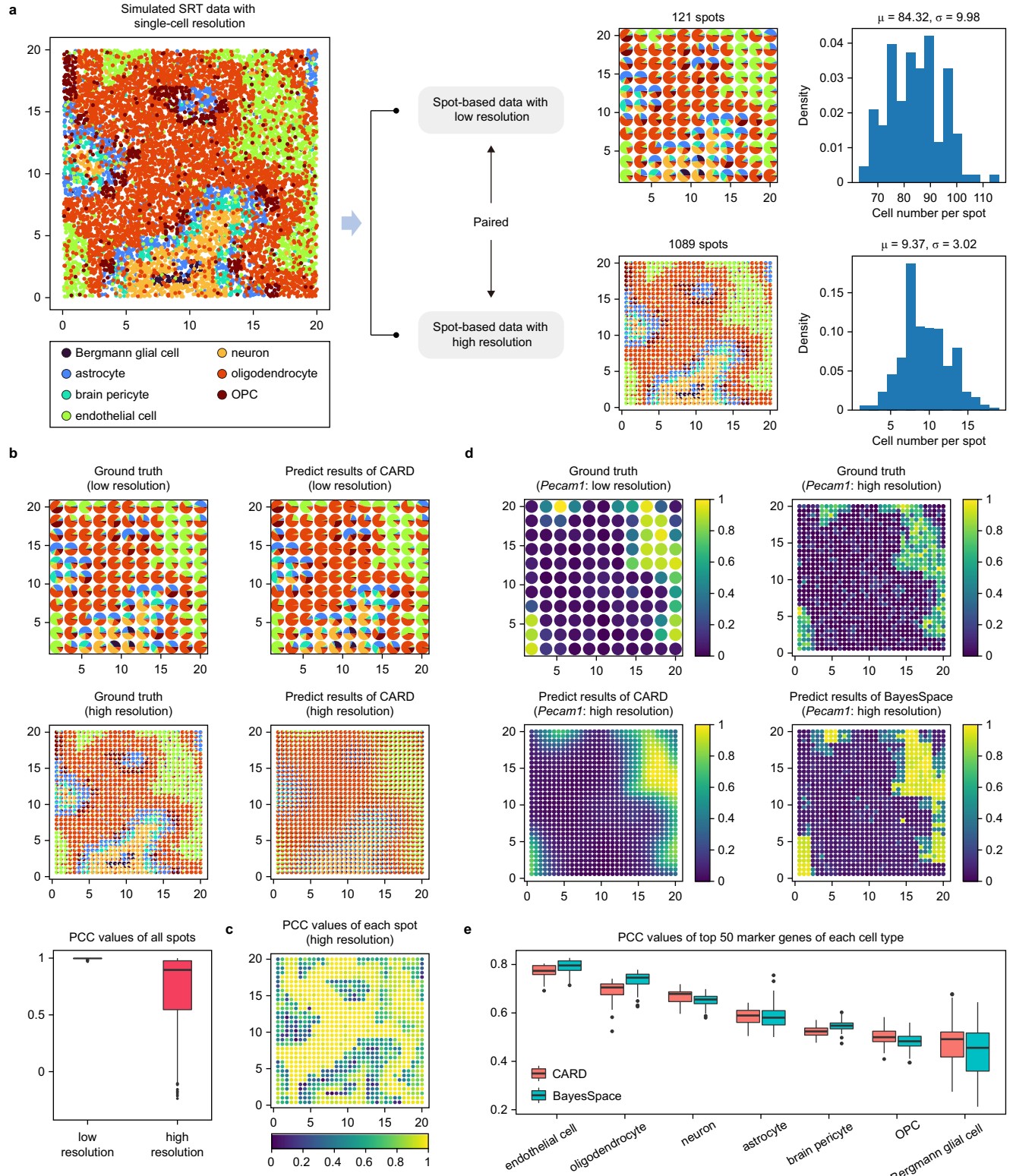

**Fig. 9 | Using scCube to benchmark resolution enhancement methods.**
**a** Illustration of simulating two pairs of spot-based SRT data with low (121 spots) and high (1089 spots) resolution by scCube. **b** The deconvolution performance of CARD on the low- and high-resolution SRT data. Data are presented as boxplots (minima, 25th percentile, median, 75th percentile, and maxima). The number of data points are 121 and 1089 for the low- and high-resolution SRT data, respectively.

**c** The PCC values of CARD on each spot for the high-resolution SRT data. **d** The high-resolution spatial maps of *Pecam1* constructed by CARD and BayesSpace. **e** Boxplots of PCC values of CARD and BayesSpace for top 50 marker genes of each cell type. Data are presented as boxplots (minima, 25th percentile, median, 75th percentile, and maxima). The number of data points are 50 for each cell type of each method. Source data are provided as a Source Data file.

them with the simulated gene expression profiles to de novo construct complete benchmarking datasets from scRNA-seq data. Compared with simulations starting from real SRT data such as implemented in SRTsim, this strategy can generate simulated SRT data that feeds both single-cell resolution and whole transcriptomes, which is suitable as the ground truth for evaluating the performance of those integration methods, such as spot deconvolution, gene imputation, and spatial reconstruction[59,65,66]. In this paper, we have demonstrated in detail the flexibility of the reference-free spatial pattern simulation of scCube by setting specified parameter values: including the simulations of multiple variability in SRT data based on a random manner, as well as the customized generations of more biologically interpretable spatial patterns.

Additionally, a desirable feature of scCube is its ability to simulate SRT data in a user-specified manner that meets diverse benchmarking purposes while providing the ground truth that is not present in real data. Specifically, scCube allows varying one specific variable when generating the SRT data, such as the number and type of genes or the number and area of spatial patterns (Supplementary Figs. S53 and 54), which allows users to investigate the impact of certain variables on the performance of different computational methods flexibly. For instance, we have utilized scCube to generate a series of simulated spot-based SRT data with different resolutions but the same other spatial variability (such as the number, proportion and spatial distribution of cell types) and benchmarked nine widely used spot deconvolution methods. Interestingly, we found that except for RCTD, Tangram, and Cell2location, the other six methods exhibited an obvious preference for specific resolutions, which was not discussed in detail in the previous studies[19,21–24,56,67,68]. Similarly, in the other two applications, we also further discussed the spatial variability that may affect the performance of methods with the simulated data generated by scCube. Moreover, expect for the SRT-analysis methods demonstrated in the three applications, scCube should be applicable to benchmark other computational tools, such as spatial domain identification and spatial cell-cell interaction inference methods. We therefore applied scCube to benchmark the above two methods, and demonstrated that the spatial information can help to improve the accuracy of spatial domain identification and spatially proximal LR pairs inference (Supplementary Figs. S35 and S55). In summary, these results suggested that scCube can provide scalable, reproducible, and realistic simulations, which helps users to evaluate diverse methods more lightly and accurately, and better facilitates the development of spatial transcriptomic data analysis methods.

## Methods

### Data collection and processing

**Human dorsolateral prefrontal cortex (DLPFC) SRT dataset.** The human DLPFC 10X Visium dataset[38] was downloaded from http://spatial.libd.org/spatialLIBD/, containing 12 tissue slices from 3 donors.

**Mouse hypothalamus SRT dataset.** The mouse hypothalamus MER-FISH dataset[69] was downloaded from https://doi.org/10.5061/dryad.8t8s248. We filtered 5 "blank" barcodes as well as the Fos gene, whose expression value in all cells was 'NA'. A total of 12 tissue slices from the naive female animal (Animal_ID: 1) were utilized.

**Mouse neocortex V1 SRT dataset.** The mouse neocortex V1 STARmap dataset[3] was downloaded from https://zenodo.org/record/7830764#.ZDpObi-1HUI, containing 2 tissue slices from one donor.

**Human HER2 breast cancer SRT dataset.** The human HER2 breast cancer ST dataset[70] was downloaded from https://zenodo.org/record/5511763#.Y6kMduxBzUI, containing 36 tissue slices from 8 patients.

**Human skin squamous cell carcinoma (SCC) SRT dataset.** The human SCC ST dataset[71] was downloaded from GEO (GSE144240), containing 8 tissue slices from 4 patients.

**Human breast cancer SRT dataset.** The human breast cancer 10X Xenium dataset[72] was downloaded from https://www.10xgenomics.com/products/xenium-in-situ/preview-dataset-human-breast, containing 1 tissue slice from one patient.

**Zebrafish embryo SRT dataset.** The zebrafish embryo Stereo-seq dataset[73] was downloaded from Spatial Omics DataBase (SODB)[74] (https://gene.ai.tencent.com/SpatialOmics/dataset?datasetID=83). The tissue slice from the 24 h post-fertilization (hpf) embryo was utilized.

**Human breast cancer SRT and scRNA-seq datasets.** The human breast cancer 10X Visium dataset and paired scRNA-seq dataset[75] were downloaded from https://doi.org/10.5281/zenodo.4739739. A total of 4 tissue slices from 4 patients (CID44971, CID4535, CID4465, and CID4290) were utilized.

**Tabula muris dataset.** The Tabula Muris scRNA-seq dataset[40] was downloaded from https://figshare.com/projects/Tabula_Muris_Transcriptomic_characterization_of_20_organs_and_tissues_from_Mus_musculus_at_single_cell_resolution/27733, containing 96,953 cells from 31 tissues.

**Tabula sapiens dataset.** The Tabula Sapiens scRNA-seq dataset[41] was downloaded from https://figshare.com/projects/Tabula_Sapiens/100973, containing 952,796 cells from 27 tissues.

**Mouse cell atlas (MCA) dataset.** The MCA scRNA-seq dataset[42] was downloaded from https://figshare.com/articles/MCA_DGE_Data/5435866, containing 333,778 cells from 83 tissues.

**Human cell landscape (HCL) dataset.** The HCL scRNA-seq dataset[43] was downloaded from https://figshare.com/articles/HCL_DGE_Data/7235471, containing 575,745 cells from 102 tissues.

**Human triple negative breast cancer (TNBC) image sets.** The human TNBC Multiplexed Imaging (MIBI) channel images[55] were downloaded from https://mibi-share.ionpath.com. A total of 3 images from 3 patients (Patient 1, Patient 5, and Patient 24) were utilized.

All the above data except the MIBI images were pre-normalized (implemented in the Python package *SCANPY*[76]) prior to being input into scCube.

### Design of scCube

The scCube model consists of two components, 1) gene expression simulation, and 2) spatial pattern simulation. scCube first simulates the gene expression profiles of different cell (or spot) populations (for example, the cell types, spatial domains, pathological regions, etc.) in gene expression simulation step. Notably, for each population, scCube can generate gene expression profiles with a customized number of cells (or spots) based on user settings. Subsequently, scCube performs either reference-based or reference-free strategies to generate spatial patterns for cell (or spot) populations in spatial pattern simulation step, which will be described in detail later.

### Gene expression simulation

scCube applies a variational autoencoder (VAE) model[39] consisting of an encoder and a decoder, to simulate the gene expression profiles of single cells of specific populations. Given the input scRNA-seq (or SRT) data $X = \{x^{(i)}\}_{i=1}^{N}$, where $x = x_1, x_2, x_n$ represents a single cell (or spot)

and its component $x_n$ represents the gene expression value of gene $n$, the encoder utilizes the conditional distribution $q_\phi(\mathbf{z}|\mathbf{x})$ to generate the latent representation $\mathbf{z}$ from the input data $\mathbf{x}$, while the decoder leverages $p_\theta(\mathbf{x}|\mathbf{z})$ to restore $\mathbf{x}$ from the latent variable $\mathbf{z}$. Here, $\phi$ and $\theta$ symbolize the parameters symbolize the encoder and decoder, respectively.

In practice, directly computing the posterior probability $p_\theta(\mathbf{z}|\mathbf{x})$ might be challenging. To address this, we utilize the concept of variational inference, introducing an approximate distribution $q_\phi(\mathbf{z}|\mathbf{x})$ as an approximation to the true posterior probability. Specifically, we aim to maximize the Evidence Lower Bound (ELBO), which can be seen as minimizing the KL divergence between the variational approximate posterior $q_\phi(\mathbf{z}|\mathbf{x})$ and the true posterior $p_\theta(\mathbf{z}|\mathbf{x})$:

$$
\begin{aligned}
\mathcal{D}_{KL}(q_\phi(\mathbf{z}|\mathbf{x})||p_\theta(\mathbf{z}|\mathbf{x})) &= \sum_{\mathbf{z}} q_\phi(\mathbf{z}|\mathbf{x}) \log\left(\frac{q_\phi(\mathbf{z}|\mathbf{x})}{p_\theta(\mathbf{z}|\mathbf{x})}\right) \\
&= \sum_{\mathbf{z}} q_\phi(\mathbf{z}|\mathbf{x}) \log\left(\frac{q_\phi(\mathbf{z}|\mathbf{x})p_\theta(\mathbf{x})}{p_\theta(\mathbf{x},\mathbf{z})}\right) \\
&= \sum_{\mathbf{z}} q_\phi(\mathbf{z}|\mathbf{x})\left[\log(p_\theta(\mathbf{x})) + \log\left(\frac{q_\phi(\mathbf{z}|\mathbf{x})}{p_\theta(\mathbf{x},\mathbf{z})}\right)\right] \\
&= \log(p_\theta(\mathbf{x})) + \sum_{\mathbf{z}} q_\phi(\mathbf{z}|\mathbf{x}) \log\left(\frac{q_\phi(\mathbf{z}|\mathbf{x})}{p_\theta(\mathbf{x}|\mathbf{z})p_\theta(\mathbf{z})}\right) \\
&= \log(p_\theta(\mathbf{x})) + \mathbb{E}_{q_\phi(\mathbf{z}|\mathbf{x})}\left[\log\left(\frac{q_\phi(\mathbf{z}|\mathbf{x})}{p_\theta(\mathbf{z})}\right) - \log(p_\theta(\mathbf{x}|\mathbf{z}))\right] \\
&= \log(p_\theta(\mathbf{x})) + \mathcal{D}_{KL}\left(q_\phi(\mathbf{z}|\mathbf{x})||p_\theta(\mathbf{z})\right) - \mathbb{E}_{q_\phi(\mathbf{z}|\mathbf{x})}\left[\log(p_\theta(\mathbf{x}|\mathbf{z}))\right]
\end{aligned}
\tag{1}
$$

By rearranging some terms, we obtain:

$$
\begin{aligned}
\log(p_\theta(\mathbf{x})) - \mathcal{D}_{KL}\left(q_\phi(\mathbf{z}|\mathbf{x})||p_\theta(\mathbf{z}|\mathbf{x})\right) &= \mathbb{E}_{q_\phi(\mathbf{z}|\mathbf{x})}\left[\log(p_\theta(\mathbf{x}|\mathbf{z}))\right] \\
&\quad - \mathcal{D}_{KL}\left(q_\phi(\mathbf{z}|\mathbf{x})||p_\theta(\mathbf{z})\right)
\end{aligned}
\tag{2}
$$

On the left-hand side of Eq. (2), we have the log-likelihood of the data $\log(p_\theta(\mathbf{x}))$ and the KL divergence $\mathcal{D}_{KL}\left(q_\phi(\mathbf{z}|\mathbf{x})||p_\theta(\mathbf{z}|\mathbf{x})\right)$ between the approximate and the true posterior. For VAE model, we want to maximize $\log(p_\theta(\mathbf{x}))$ and minimize $\mathcal{D}_{KL}\left(q_\phi(\mathbf{z}|\mathbf{x})||p_\theta(\mathbf{z}|\mathbf{x})\right)$, which is equivalent to maximizing the right-hand side of the equation, i.e.:

$$
\mathbb{E}_{q_\phi(\mathbf{z}|\mathbf{x})}\left[\log(p_\theta(\mathbf{x}|\mathbf{z}))\right] - \mathcal{D}_{KL}\left(q_\phi(\mathbf{z}|\mathbf{x})||p_\theta(\mathbf{z})\right)
\tag{3}
$$

Equation (3) is also known as the ELBO. To maximize the Eq. (3), we assume $p_\theta(\mathbf{z})$ takes on a multivariate Gaussian $p_\theta(\mathbf{z}) = \mathcal{N}(\mathbf{z};\mathbf{0},\mathbf{I})$ and the true (but computationally intractable) posterior $p_\theta(\mathbf{z}|\mathbf{x})$ is approximated by a Gaussian form with a diagonal covariance. This assumption is reasonable as single cells from distinct cell types or domains have unique gene expression profiles that can be characterized by independent latent clustering spaces[20]. And in this case, we can let the variational approximate posterior $q_\phi(\mathbf{z}|\mathbf{x})$ be a multivariate Gaussian with a diagonal covariance structure:

$$
\log q_\phi\left(\mathbf{z}|\mathbf{x}^{(i)}\right) = \log \mathcal{N}\left(\mathbf{z};\boldsymbol{\mu}^{(i)},\boldsymbol{\sigma}^{2(i)}\mathbf{I}\right)
\tag{4}
$$

Where the mean and s.d. of the approximate posterior, $\boldsymbol{\mu}^{(i)}$ and $\boldsymbol{\sigma}^{(i)}$, can be implemented with the encoding MLP network. The KL term in Eq. (3) can be integrated analytically since both $p_\theta(\mathbf{z})$ and $q_\phi(\mathbf{z}|\mathbf{x})$ are multivariate Gaussian distributions, while the first term in Eq. (3), also termed as expected reconstruction error, requires estimation by sampling. To make sure the gradient descent algorithm can be used to train the model, we employ the reparameterization trick proposed by Kingma and Welling[39] to make the optimization function differentiable. Specifically, we don't directly sample $\mathbf{z}$ from the posterior but first sample $\boldsymbol{\epsilon}$ from the standard normal distribution and then compute

$\mathbf{z} = \boldsymbol{\mu} + \boldsymbol{\sigma} \odot \boldsymbol{\epsilon}$. Thus, the gradient can be backward propagated for updating the model parameters.

Both the encoder and decoder apply a four-layer perceptron in our model, where each layer is followed by a RELU activation except the last layer of the encoder. For the encoder, the number of neurons in each layer is 2048, 1024, 512 and 256 respectively. While for the decoder, the number of neutrons in each layer is 512, 1024, 2048 and $k$, respectively, where $k$ represents the number of genes in the dataset. The dimensionality of the latent space learned by the VAE is 128.

### Spatial pattern simulation

**Reference-based spatial pattern simulation.** In the reference-based simulation, scCube focuses on simulating the spatial expression patterns of genes over the spatial coordinates of the spatial reference data. Specifically, scCube first utilizes the VAE model trained in the gene expression simulation step to generate new cells (or spots) of different populations, whose numbers are determined by the groups in the spatial reference data. Significantly, the groups can be defined as the annotation information such as spatial domain or pathological annotation (if provided), or just as cluster labels obtained by the unsupervised or spatial domain clustering steps. Compared to the spatial reference, these generated cells (or spots) have sufficiently similar but not identical gene expression profiles. Then, scCube constructs a mapping between the cells (or spots) in generated data and the positions in the spatial reference by solving an optimal transport problem[64]:

$$
\begin{aligned}
\gamma &= \mathrm{argmin}_\gamma \langle\gamma,\mathbf{M}\rangle_F \\
&\textit{subject to } \gamma\mathbf{1} = \boldsymbol{a} \\
&\qquad\qquad \gamma^T\mathbf{1} = \boldsymbol{b} \\
&\qquad\qquad \gamma \geq 0
\end{aligned}
\tag{5}
$$

Where $\mathbf{M}$ is the metric cost matrix and $\boldsymbol{a}$, $\boldsymbol{b}$ are the sample weights. We assume $\boldsymbol{a}$ and $\boldsymbol{b}$ take uniform distributions $a = \frac{1}{n}\boldsymbol{I}_n$ and $b = \frac{1}{n}\boldsymbol{I}_n$ by default, and the metric cost matrix $\mathbf{M}$ is constructed by computing the sqeuclidean distance between the gene expression vectors of the generated cells (or spots) and the gene expression vectors of the positions in the spatial reference data.

Once we have the optimal transport plan $\gamma$, we can map the generated cells to positions with the maximum likelihood of spatial origin, and thus the spatial patterns of cell (or spot) populations and the spatial expression patterns of genes are also simulated.

**Reference-free spatial pattern simulation.** In the reference-free simulation, scCube focuses on simulating specific spatial distribution patterns for different cell (or spots) populations based on the generated data and further provides two strategies. After randomly generating spatial positions with the same number as the cells (or spots) in the generated data, users can either choose to generate the random or customized spatial patterns.

**1) Random spatial pattern generation.** To be specific, scCube applies the default spatial autocorrelation function to generate random spatial patterns of populations. The spatial range containing all positions is first uniformly divided into $n \times n$ grids. To encourage neighborhood similarity in the composition of cell type in grids, we assume that the overall distribution $V$ of grids follows a multivariate normal distribution:

$$
V \sim MVN(0,\Sigma)
\tag{6}
$$

Where the $n \times n$ covariance matrix $\boldsymbol{\Sigma}$ models the spatial correlation among the grids based on the Euclidean distance between their spatial

coordinates:

$$\boldsymbol{\Sigma} = e^{-\left(\frac{D}{\delta}\right)^2} \tag{7}$$

Where $\boldsymbol{D}$ is the Euclidean distance matrix and $\delta$ is the autocorrelation parameter. A larger $\delta$ will generate spatial patterns with greater connectivity. Subsequently, for a target population, scCube randomly samples the same proportion of grids based on the proportion of this population in the generated data, and assigns all cells (or spots) which belongs to this population to the spatial positions in the sampled grids. This step repeats $c$ times ($c$ is set by users and does not exceed the total number of populations in the generated data), and cells (or spots) in the remaining population are randomly assigned to spatial positions in the remaining grids. Based on the above strategy, users can flexibly choose to simulate a unique spatial distribution pattern for each population, or simulate the same spatial co-localization distribution patterns for multiple populations simultaneously. Furthermore, scCube also provides an additional parameter, $\lambda$, to generate spatial patterns with varying degrees of fuzziness by randomly swapping the spatial coordinates of $1 - \lambda$ percent of the cells.

**2) Customized spatial pattern generation.** Inspired by spaSim[54], a recently published simulator of tissue spatial data, scCube provides four functions to optionally generate a series of biologically interpretable spatial basis patterns for each population, including unstructured mixed cell populations, clustered cell populations, cell rings encircling tissue structures, and some external structures such as vessels. Furthermore, scCube also provides an additional function to combine the different types and numbers of basis patterns mentioned above to generate more complex structures.

**Unstructured mixed cell populations simulation.** For unstructured mixed cell populations simulation, scCube randomly samples positions with a specific proportion and assigns designated cell populations to these positions. The number and proportion of cell populations can be set by users.

**Clustered cell populations simulation.** For clustered cell populations simulation, scCube applies geometric shapes (including circles, ovals, and other irregular shapes formed by combining circles or ovals) to delineate regions of clustered cell populations. Positions inside the margin of geometric shapes are defined as clustered structures and assigned designated cell populations, while positions outside the margin are defined as the background. Specifically, the number and parameters (such as center location, size, and direction) of geometric shapes can be customized. Given the center location $(x_0, y_0)$ and the direction $\theta$ of the region for simulation, scCube first converts the coordinate $(x, y)$ of each position as below:

$$\begin{aligned} x_{rotated} &= (x - x_0)\cos(\theta) + (y - y_0)\sin(\theta) \\ y_{rotated} &= -(x - x_0)\sin(\theta) + (y - y_0)\cos(\theta) \end{aligned} \tag{8}$$

Subsequently, scCube will apply circular or elliptical coordinate equations:

$$\frac{x^2}{a^2} + \frac{y^2}{b^2} = 1 \tag{9}$$

Where $a$ and $b$ are the major axis and minor axis, respectively, and $a = b$ when the regions are circles. For each position, if $\frac{x_{rotated}^2}{a^2} + \frac{y_{rotated}^2}{b^2} < 1$, the position is inside the region, otherwise it's outside the region. In addition, scCube further considers the presence of other infiltrating cells in the clustered cell populations, and the cell type and proportion of infiltrating cells can be specified by users.

**Cell rings simulation.** scCube applies concentric circles to simulate the cell single or double rings. Similar to clustered cell populations simulation, the number and parameter of concentric circles as well as the cell type and proportion of infiltrating cells can also be specified by users. Moreover, scCube provides an additional parameter $d$ to control the widths of cell rings. Taking the cell single rings simulation as an example, for each position, if $\frac{x_{rotated}^2}{a^2} + \frac{y_{rotated}^2}{b^2} < 1$, the position is inside the inner cluster; if $\frac{x_{rotated}^2}{a^2} + \frac{y_{rotated}^2}{b^2} > 1$ and $\frac{x_{rotated}^2}{(a+d)^2} + \frac{y_{rotated}^2}{(b+d)^2} < 1$, the position is inside the outer ring; otherwise it's outside the outer ring and is defined as the background.

**Vessels simulation.** scCube applies stripes to simulate the blood or lymphatic vessels. The number and parameter (including length, width, and direction) of stripes as well as the cell type and proportion of infiltrating cells can be specified by users. Given two center endpoints $(x_1, y_1)$ and $(x_2, y_2)$ and the width $d$ of the stripe for simulation, scCube first calculates the normalized perpendicular vector $\boldsymbol{p}$:

$$\boldsymbol{p} = \left[ \frac{-(y_2 - y_1)}{l}, \frac{(x_2 - x_1)}{l} \right] \tag{10}$$

Where $l = \sqrt{(x_2 - x_1)^2 + (y_2 - y_1)^2}$ is the length of the stripe, and the endpoints on the margin of stripe thus can be defined as $(x_1 + d\frac{\boldsymbol{p}}{2}, y_1 + d\frac{\boldsymbol{p}}{2})$ or $(x_1 - d\frac{\boldsymbol{p}}{2}, y_1 - d\frac{\boldsymbol{p}}{2})$. For each position $(x, y)$, scCube further calculates its projection lengths $u$ and $v$ on the direction vector and the perpendicular vector of the stripe, respectively:

$$\begin{aligned} u &= \left(x - \left(x_1 + d\tfrac{\boldsymbol{p}}{2}\right)\right)\left(\tfrac{x_2 - x_1}{l}\right) + \left(y - \left(y_1 + d\tfrac{\boldsymbol{p}}{2}\right)\right)\left(\tfrac{y_2 - y_1}{l}\right) \\ v &= -\left(x - \left(x_1 + d\tfrac{\boldsymbol{p}}{2}\right)\right)\left(\tfrac{y_2 - y_1}{l}\right) + \left(y - \left(y_1 + d\tfrac{\boldsymbol{p}}{2}\right)\right)\left(\tfrac{x_2 - x_1}{l}\right) \end{aligned} \tag{11}$$

If $0 \le u \le l$ and $|v| \le \frac{d}{2}$, the position is inside the stripe, otherwise it's outside the stripe and is defined as the background.

## Comparison with other simulation methods

We compared the simulation performance of scCube with other existing simulators using the real SRT datasets. In detail, we considered the following eight simulators, two of which are SRT simulators and six are single-cell simulators:

(1) SRTsim[36]: SRT simulator, implemented in the R package *SRTsim*, with the count matrix as inputs.
(2) scDesign3[37]: SRT simulator, implemented in the R package *scDesign3*, with the count matrix as inputs.
(3) scDesign2[44]: single-cell simulator, implemented in the R package *scDesign2*, with the count matrix as inputs.
(4) SymSim[45]: single-cell simulator, implemented in the R package *SymSim*, with the count matrix as inputs.
(5) ZINB-WaVE[46]: single-cell simulator, implemented in the R package *Splatter*, with the count matrix as inputs.
(6) Splat[47]: single-cell simulator, implemented in the R package *Splatter*, with the count matrix as inputs.
(7) Splat Simple[47]: single-cell simulator, implemented in the R package *Splatter*, with the count matrix as inputs.
(8) Kersplat[47]: single-cell simulator, implemented in the R package *Splatter*, with the count matrix as inputs.

All compared simulation methods were set with default parameters and we did not perform any pre-normalized step on the raw count matrices.

## Evaluation metrics

**PCC_GEV.** The Pearson correlation coefficient (PCC) values between the normalized expression vector for each gene across spatial positions in real and simulated data. The $PCC_{GEV}$ value was calculated using the

following equation:

$$PCC_{GEV} = \frac{\sum_{j=1}^{n}(x_{ij} - \bar{x})(y_{ij} - \bar{y})}{\sqrt{\sum_{j=1}^{n}(x_{ij} - \bar{x})^2}\sqrt{\sum_{j=1}^{n}(y_{ij} - \bar{y})^2}} \quad (12)$$

Where $n$ is the number of spots/cells, $x_{ij}$ and $y_{ij}$ are the expression value of gene $i$ in spot/cell $j$ in the real data and the simulated data, respectively; $\bar{x}$ and $\bar{y}$ are the average expression value of gene $i$ across all spots/cells in the real data and the simulated data, respectively.

**MAE$_{GEV}$.** The mean absolute error (MAE) values between the normalized expression vector for each gene across spatial positions in real and simulated data. The MAE$_{GEV}$ value was calculated using the following equation:

$$MAE_{GEV} = \frac{1}{n}\sum_{j=1}^{n}|x_{ij} - y_{ij}| \quad (13)$$

Where $n$ is the number of spots/cells, $x_{ij}$ and $y_{ij}$ are the expression value of gene $i$ in spot/cell $j$ in the real data and the simulated data, respectively.

## PCC$_{GBM}$

Considering that direct comparison between the real and simulated gene expression vectors is overly stringent and leads to insufficient comparison results when the training data and the test data are different, according to Song et al.[37], we further calculated the Pearson correlation coefficient (PCC) values between the expression value for each gene from the simulated data's spatial locations predicted by the two generalized boosted regression models (GBMs) trained on the real data and simulated data separately. The GBMs were implemented in the R package *caret*. Compared with PCC$_{GEV}$, this alternative evaluation metric is more robust and informative (Supplementary Fig. S56). The PCC$_{GBM}$ value was calculated using the Eq. 12 above, where $n$ is the number of spots/cells, $x_{ij}$ and $y_{ij}$ are the expression value of gene $i$ in spot/cell $j$ predicted by the GBMs trained on the real data and the simulated data, respectively; $\bar{x}$ and $\bar{y}$ are the average expression value of gene $i$ across all spots/cells predicted by the GBMs trained on the real data and the simulated data, respectively.

## Benchmarking spot deconvolution methods

For the spot deconvolution methods benchmarking, we first used scCube to generate the simulated SRT data with single-cell resolution from a mouse brain scRNA-seq dataset, each cell type has a specific spatial distribution pattern. For this single-cell SRT data, all cells were split according to the fixed spatial distance which was determined by the setting average number of cells per spot $n$ and then merged into simulated spots as the benchmarking datasets. We simulated six spot-based SRT datasets (n = 5, 10, 20, 30, 50, and 100) in total and benchmarked the following nine spot deconvolution methods with the default parameters: (1) Cell2location[19] implemented in the Python package *cell2location*; (2) DestVI[21] implemented in the Python package *scvi-tools*; (3) DSTG[22] implemented in the Python package *DSTG*; (4) RCTD[25] implemented in the R package *spacexr*; (5) Seurat[29] implemented in the R package *Seurat*; (6) spatialDWLS[24] implemented in the R package *Giotto*; (7) SPOTlight[23] implemented in the R package *SPOTlight*; (8) Stereoscope[26] implemented in the Python package *Stereoscope*; (9) Tangram[27] implemented in the Python package *Tangram*.

## Evaluation metrics

**PCC.** The Pearson correlation coefficient (PCC) was calculated using the Eq. 12 above, where $n$ is the number of cell types, $x_{ij}$ and $y_{ij}$ are the proportion value of cell type $j$ in spot $i$ in the ground truth and the predicted result, respectively; $\bar{x}$ and $\bar{y}$ are the average proportion value across all cell types in spot $i$ in the ground truth and the predicted result, respectively.

**SRCC.** The Spearman rank-order correlation coefficient (SRCC) was calculated using the following equation:

$$SRCC = 1 - \frac{6\sum_{j=1}^{n}d_{ij}^2}{n(n^2 - 1)} \quad (14)$$

Where $n$ is the number of cell types, $d_{ij}$ is the rank value of the proportion value of cell type $i$ in spot $i$ in the ground truth and the predicted result, respectively.

**RMSE.** The root mean squared error (RMSE) was calculated using the following equation:

$$RMSE = \sqrt{\frac{1}{n}\sum_{j=1}^{n}(x_{ij} - y_{ij})^2} \quad (15)$$

Where $n$ is the number of cell types, $x_{ij}$ and $y_{ij}$ are the proportion value of cell type $j$ in spot $i$ in the ground truth and the predicted result, respectively.

**JS.** The Jensen-Shannon divergence (JS) was calculated using the following equation:

$$JS = \frac{1}{2}KL\left(P_i \middle\| \frac{P_i + Q_i}{2}\right) + \frac{1}{2}KL\left(Q_i \middle\| \frac{P_i + Q_i}{2}\right) \quad (16)$$

$$KL(x_i\|y_i) = \sum_{j=1}^{n}\left(x_{ij}\log\frac{x_{ij}}{y_{ij}}\right) \quad (17)$$

Where $n$ is the number of cell types, $x_{ij}$ and $y_{ij}$ are the proportion value of cell type $j$ in spot $i$ in the ground truth and the predicted result, respectively; $P_i$ and $Q_i$ are the distribution probability vectors of cell type proportion in spot $i$ in the ground truth and the predicted result, respectively.

**AS.** The accuracy score (AS) proposed by Li et al.[56] was calculated using the following equation:

$$AS = \frac{1}{4}\left(RANK_{PCC} + RANK_{SRCC} + RANK_{RMSE} + RANK_{JS}\right) \quad (18)$$

## Benchmarking gene imputation methods

For the gene imputation methods benchmarking, we also used scCube to generate the new scRNA-seq data with whole transcriptomes from the mouse brain scRNA-seq dataset. Since none of the existing gene imputation methods make use of the spatial information of single cells, we thus did not simulate the spatial coordinates for each generate cells. On the contrary, we focused on the effect of the quality of detected transcriptomes in image-based SRT data on the gene imputation methods. To do so, we utilized scCube to generate three image-based SRT data with 200 randomly selected genes, 200 highly variable genes, and 200 cell type marker genes, respectively, and benchmarked the following eight gene imputation methods with the default parameters: (1) SpaGE[28] implemented in the Python package *SpaGE*; (2) Tangram[27] implemented in the Python package *Tangram*; (3) stPlus[31] implemented in

the Python package *stPlus*; (4) gimVI[57] implemented in the Python package *scvi-tools*; (5) Seurat[29] implemented in the R package *Seurat*; (6) SpaOTsc[58] implemented in the Python package *SpaOTsc*; (7) novoSpaRc[59] implemented in the Python package *novoSpaRc*; (8) LIGER[30] implemented in the R package *liger*.

### Evaluation metrics

The PCC, SRCC, RMSE, JS, and AS mentioned above were employed to evaluate the gene imputation performance, and the input $n$ is the number of spots/cells, $x_{ij}$ and $y_{ij}$ are the expression value of gene $i$ in spot/cell $j$ in the in the ground truth and the predicted result, respectively.

### Benchmarking resolution enhancement methods

For the resolution enhancement methods benchmarking, we simulated two pairs of spot-based SRT data with low (121 spots) and high (1089 spots) cellular resolution using scCube and benchmarked the following two existing methods with the default parameters: (1) BayesSpace[49] implemented in the R package *BayesSpace*; (2) CARD[60] implemented in the R package *CARD*.

### Evaluation metrics

The PCC mentioned above were employed to evaluate the accuracy of high-cellular resolution spatial maps of gene expression predicted by BayesSpace and CARD, and the input $n$ is the number of spots, $x_{ij}$ and $y_{ij}$ are the expression value of gene $i$ in spot $j$ in the in the ground truth and the predicted result, respectively. Besides, since CARD also predicted the cell type composition of spots for the both low- and high-cellular resolution SRT data, we thus also evaluated the deconvolution result using the PCC mentioned above, and the input $n$ is the number of cell types, $x_{ij}$ and $y_{ij}$ are the proportion value of cell type $j$ in spot $i$ in the ground truth and the predicted result, respectively.

### Benchmarking spatial domain identification methods

For the spatial domain identification methods benchmarking, we selected a 10X Visium SRT dataset from the human dorsolateral prefrontal cortex (DLPFC) containing 12 tissue slices[38] and generated the corresponding simulated SRT data for each slice using the reference-based strategy of scCube as the benchmarking datasets. The cortex layer label of each spot in these 12 simulated datasets was known and can be used as the ground truth. Five spatial domain identification methods (including (1) BayesSpace[49] implemented in the R package *BayesSpace*; (2) SpaGCN[18] implemented in the Python package *SpaGCN*; (3) STAGATE[33] implemented in the Python package *STAGATE*; (4) DR-SC[32] implemented in the R package *DR.SC*; (5) stLearn[77] implemented in the Python package *stLearn*), as well as one non-spatial unsupervised clustering method (Seurat[29] implemented in the R package *Seurat*) were benchmarked with the default parameters.

### Evaluation metrics

**ARI.** The adjusted rand index (ARI) was calculated using the following equation:

$$ARI = \frac{\sum_{ij}\binom{n_{ij}}{2} - \left[\sum_i\binom{a_i}{2}\sum_j\binom{b_j}{2}\right]/\binom{n}{2}}{\frac{1}{2}\left[\sum_i\binom{a_i}{2} + \sum_j\binom{b_j}{2}\right] - \left[\sum_i\binom{a_i}{2} + \sum_j\binom{b_j}{2}\right]\binom{n}{2}} \quad (19)$$

Where $n$ is number of spots, $a_i$ and $b_j$ are the number of spots belonging to class $i$ in the ground truth and cluster $j$ in the predicted result, respectively, and $n_{ij}$ is number of spots belonging to class $i$ and cluster $j$ simultaneously.

**NMI.** The normalized mutual information (NMI) was calculated using the following equation:

$$NMI = \frac{\sum_{ij}\left(\frac{n_{ij}}{n}\right)\log\left(\frac{n \times n_{ij}}{a_i \times b_j}\right)}{\frac{1}{2}\left[-\sum_i\frac{a_i}{n}\log\left(\frac{a_i}{n}\right) - \sum_j\frac{b_j}{n}\log\left(\frac{b_j}{n}\right)\right]} \quad (20)$$

Where the notation is the same as that in ARI.

**HS.** The homogeneity score (HS) was calculated using the following equation:

$$HS = 1 - \frac{H(C|K)}{H(C)} \quad (21)$$

Where $H(C|K)$ is the conditional entropy of the classes given the cluster assignments and is given by:

$$H(C|K) = -\sum_{c=1}^{|C|}\sum_{k=1}^{|K|}\frac{n_{c,k}}{n}\cdot\log\left(\frac{n_{c,k}}{n_k}\right) \quad (22)$$

and $H(C)$ is the entropy of the classes and is given by:

$$H(C) = -\sum_{c=1}^{|C|}\frac{n_c}{n}\cdot\log\left(\frac{n_c}{n}\right) \quad (23)$$

Where $n$ is number of spots, $n_c$ and $n_k$ are the number of spots belonging to class $c$ in the ground truth and cluster $k$ in the predicted result, respectively, $n_{c,k}$ is the number of spots from class $c$ assigned to cluster $k$.

**FMI.** The Fowlkes-Mallows score (FMI) was calculated using the following equation:

$$FMI = \sqrt{Precision \times Recall} \quad (24)$$

Where $Precision = \frac{TP}{TP+FP}$ is proportion of spots that are correctly classified in the predicted result, and $Recall = \frac{TP}{TP+FP}$ is proportion of spots that are correctly classified in the ground truth.

### Benchmarking spatial cell-cell interaction inference methods

For the spatial cell-cell interaction inference benchmarking, we selected a STARmap SRT data from the mouse V1 neocortex[3] and generated the corresponding simulated SRT data using the reference-based strategy of scCube as the benchmarking datasets. One recently published spatially resolved cell-cell communication inference method (SpaTalk[34] implemented in the R package *SpaTalk*) as well as other seven scRNA-seq cell-cell communication inference methods (including (1) CellCall[78] implemented in the R package *CellCall*; (2) CellChat[79] implemented in the R package *CellChat*; (3) CellPhoneDB[80] implemented in the Python package *cellphonedb*; (4) CytoTalk[81] implemented in the R package *CytoTalk*; (5) Giotto[82] implemented in the R package *Giotto*; (6) NicheNet[83] implemented in the R package *nichenetr*; (7) SpaOTsc[58] implemented in the Python package *SpaOTsc*) were benchmarked with the default parameters.

### Evaluation metrics

The spatial proximity significance and the co-expression level on the cell-cell spatial graph network of the inferred ligand-receptor interactions (LRIs) proposed by Shao et al.[34] were employed to evaluate the accuracy of all methods in spatial cell-cell interaction inference.

## Statistics & reproducibility

No statistical method was used to predetermine the sample size. There were no data exclusions except for the filtering of 5 "blank" barcodes and the Fos gene in the MERFISH data. The experiments were not randomized. The investigators were not blinded to allocation during experiments and outcome assessment. Python (version 3.8.5) and R (version 4.1.0) are used for the statistical analysis.

## Reporting summary

Further information on research design is available in the Nature Portfolio Reporting Summary linked to this article.

## Data availability

No new data was generated for this study. All data used in this study is publicly available and can be accessed through the following links: (1) the human dorsolateral prefrontal cortex (DLPFC) 10X Visium dataset [http://spatial.libd.org/spatialLIBD/][38]; (2) the mouse hypothalamus MERFISH dataset [https://doi.org/10.5061/dryad.8t8s248][69]; (3) the mouse neocortex V1 STARmap dataset [https://zenodo.org/record/7830764#.ZDpObi-1HUI][3]; (4) the human HER2 breast cancer "Spatial Transcriptomics (ST)" dataset [https://zenodo.org/record/5511763#.Y6kMduxBzUI][70]; (5) the human skin squamous cell carcinoma (SCC) "Spatial Transcriptomics (ST)" dataset [GSE144240][71]; (6) the human breast cancer 10X Xenium dataset [https://www.10xgenomics.com/products/xenium-in-situ/preview-dataset-human-breast][72]; (7) the zebrafish embryo Stereo-seq dataset [https://gene.ai.tencent.com/SpatialOmics/dataset?datasetID=83][73]; (8) the human breast cancer 10X Visium and paired scRNA-seq dataset [https://doi.org/10.5281/zenodo.4739739][75]; (9) the he Tabula Muris scRNA-seq dataset [https://figshare.com/projects/Tabula_Muris_Transcriptomic_characterization_of_20_organs_and_tissues_from_Mus_musculus_at_single_cell_resolution/27733][40]; (10) the Tabula Sapiens scRNA-seq dataset [https://figshare.com/projects/Tabula_Sapiens/100973][41]; (11) the MCA scRNA-seq dataset [https://figshare.com/articles/MCA_DGE_Data/5435866][42]; (12) the HCL scRNA-seq dataset [https://figshare.com/articles/HCL_DGE_Data/7235471][43]; (13) the human TNBC Multiplexed Imaging (MIBI) channel images [https://mibi-share.ionpath.com][55]. Source data are provided with this paper.

## Code availability

scCube is an open-access python package available in the GitHub repository [https://github.com/ZJUFanLab/scCube][84].

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

## Acknowledgements

This work is supported by the National Natural Science Foundation of China (U23A20513, X.F.), Ningbo Top Medical and Health Research Program (No. 2022030309, X.F.), the Fundamental Research Funds for the Central Universities (226-2024-00001, X.F.). The authors thank the High-Performance Computing Cluster of Zhejiang University Innovation Center of Yangtze River Delta for their technical support and thank spaSim for its inspiration in the development of reference-free spatial pattern simulation for this project.

## Author contributions

X.F. and Y.C. conceived the study. J.Q., Y.F., Z.C., and C.L. implemented the scCube model. J.Q., H.B., X.S., J.L., W.G., Y.H., A.L., and Y.Y. collected datasets involved in this article and performed benchmarking experiments. J.Q. and H.B. wrote the manuscript, and all authors edited and revised the manuscript.

## Competing interests

The authors declare no competing interests.
