## [Peer Review File · Nature Communications]

Simulating multiple variability in spatially resolved transcriptomics with scCubeReviewer #1 (Remarks to the Author):

Accurate simulation of pseudo spatial transcriptomics (ST) datasets is crucial for enhancing downstream data analyses. This study proposed a simulator called scCube for simulating multiple spatial variability in spatial resolved transcriptomics and producing unbiased simulated ST data. Benchmark results indicated that the variational autoencoder (VAE) framework-based model outperformed other previously published models. The manuscript shows an interesting methodology for ST analysis. However, the reviewer has the following concerns that need to be addressed:

1. In reference strategy simulation, the authors selected a 10X Visium dataset (Figure S3) as the benchmark dataset. However, their test strategy is similar to the MERFISH dataset because they treated each Visium spot as an individual entity to generate new data. It's worth noting that each spot in the Visium dataset is a mixture of multiple cells. In many scenarios, our reference Visium slice lacks annotation information, whereas we only have annotated scRNA-seq reference data (an example of which can be found in DOI: <https://doi.org/10.1038/s41588-021-00911-1> or PMID: 34493872, as the authors provided both Visium data and scRNA-seq data from human breast cancer samples). I am curious how scCube leverages these references to simulate new similar pseudo slices that closely resemble the Visium data.
2. [line 578] "..., whose numbers are determined by the ground truth in the spatial reference data." How do the authors define the "ground truth" in this context, especially when we do not have annotations in the spatial reference data as I mentioned in comment 1?
3. The count matrix or logarithmized matrix of many scRNA-seq data or spatial transcriptomics data is highly sparse. I am curious whether the authors have performed any calibrations to induce sparsity in the generated gene expression data.
4. The examples designed in the sections titled "Using scCube to benchmark spot deconvolution methods", "Using scCube to benchmark gene imputation methods" and "Using scCube to benchmark resolution enhancement methods" make no sense to readers. A primary objective of these sections is to demonstrate the utility of scCube-simulated data for benchmarking spot deconvolution, gene imputation and resolution enhancement. To achieve this, the authors can first use a real-world data, such as MERFISH shown in Figure 2, as well as a pseudo data simulated by scCube for model benchmark, respectively. By comparing the benchmark results between the real-world data and scCube-simulated data, they can establish whether scCube-simulated data is biologically interpretable. If the benchmark results between the real-world data and scCube-simulated data are similar, it would support the conclusion that scCube-simulated data is biologically meaningful.
5. In the section "Simulating multiple variability in SRT data by the reference-free strategy with scCube", the authors should consider simulating ST examples that are more biologically interpretable, such as a spherical tumor tissue surrounded by normal tissues.
6. The descriptions for data simulation still lack detailed information in terms of Figure 1 and the Methods section. For instance, it is not clear how the authors implemented cell type features to simulate ST data.
7. The Figure 1 can be improved by including more information, such as how scCube integrates gene expression simulation and spatial pattern simulation for downstream analysis.
8. In Figure 3a and 3b, a 5x5 image matrix displaying the different combinations of λ and δ will better illustrate how these two parameters impact the simulation results.
9. [line 17-19] "The rapid advancement of computational methods for spatially resolved transcriptomic (SRT) data analysis raises the need for unbiased simulated datasets for evaluation and validation." The causation appears somewhat unclear. The authors should offer a more compelling justification to clarify the necessity of simulating SRT data.
10. [line 145] "PCC" should be defined at its first occurrence.

Reviewer #2 (Remarks to the Author):

This manuscript introduces scCube, a simulator for spatially resolved transcriptomics (SRT) data, offering versatile ground truths with editable resolution, spatial pattern continuity, and tissue dimension. scCube is designed for benchmarking SRT computational tools, including deconvolution,

gene imputation, and resolution enhancement. While scCube holds promise for the SRT community, the manuscript requires refinement for a clearer understanding of the methodology. Additionally, suggestions for both methodology and results are provided below.

****Major Points:****

****Results:****

1. Define the evaluation metric PCC used in the comparison with other simulators. The simulators in row 1 of Fig. 2 perform similarly but show different PCC quantities. Consider incorporating multiple metrics to assess how well synthetic data mimics real data.
2. Evaluate scCube's performance on other SRT sequencing technologies like GeoMX and 10X Xenium, not just 10x Visium and MERFISH.
3. Explore if scCube can help benchmark computational tools for spatial domain identification and spatial cell-cell interaction inference.
4. In Fig. S11, compare scCube's performance with other simulators in the same scenario for clarity.

****Methodology:****

1. Assess if scCube allows manual specification of patterns for specific genes in SRT data simulation.
2. Consider incorporating flexibility in scCube for generating within-cell-type spatial patterns to account for intra-cell-type gene expression variation.
3. Clarify the reference-free spatial pattern simulation process. Provide a visual aid (e.g., a plot similar to Fig. S1) to enhance understanding, especially for the representation of the overall distribution denoted as V in line 604.
4. Explain the choice of the VAE model in the initial step of generating scRNA-seq data. Address why the VAE model was chosen and whether alternative models could be considered.

****Minor Points:****

1. Correct a typo in Fig. 2d: "Bregma -0.09" should be "Bregma -0.29."
2. In gene expression simulation (line 530), correct the statement: the conditional distribution should be $q_{\phi}(z|x)$, not $p_{\phi}(z|x)$.
3. Rectify the x-axis range in the bottom-right density plot in Figure 5.a for accuracy.

Reviewer #3 (Remarks to the Author):

This paper presents an algorithm for simulating spatially resolved transcriptomics data. The algorithm aims to preserve spatial expression patterns and generate simulated data with various spatial patterns, resolutions, tissue slice dimensions, and more. The study comprehensively compares the performance of scCube with existing algorithms in terms of the aforementioned spatial variations. Additionally, the paper demonstrates the usefulness of scCube in three benchmarking applications. The idea behind scCube is straightforward and logical, and it demonstrates its ability to simulate unbiased SRT data. However, there are a few points that need to be addressed.

1. scCube consists of two steps: 1) single-cell gene expression simulation and 2) spatial pattern simulation. These steps appear to be independent of each other. It would be interesting to compare the performance of the first step with that of existing single-cell simulation tools. Can existing simulation tools be used as a substitute for the first step of the scCube algorithm?
2. The authors tested scCube using Slide-seq, 10X, and ST datasets. However, these datasets are relatively low-resolution and do not provide cell-level gene expression. It would be interesting to

see how scCube performs when simulating high-resolution datasets that do provide cell-level gene expression, such as stereo-seq datasets.

3. Many existing tools for simulating spatial transcriptomics data take a very long time and sometimes fail to complete the simulation. It would be helpful if the authors could provide the execution time for scCube in these simulated datasets.

4. In Figure 7a, it would be helpful if the authors could explain why the UMAP plot, calculated based on randomly selected genes, is still able to accurately distinguish cell types.

Response to reviewers

Overview of Changes

We sincerely appreciate the reviewers' constructive comments and positive feedback. Our work has been much improved based on their valuable suggestions. We have tried our best to address the concerns raised by the reviewers point by point. Compared to the original version, the revised manuscript adds substantial benchmark results, adequately describes the methodology, provides more spatial pattern simulation functions, and demonstrates the utility of scCube in benchmarking other SRT computational tools. The main changes are summarized as follows.

1. The Method section and the schematic workflow in Figure 1 of the manuscript have been reworked to include more detailed information for a clearer understanding of the design concept and methodology of scCube.
2. The simulation performance of scCube is benchmarked more systematically. Additional SRT datasets from other sequencing platforms (10X Xenium, Stereo-seq, STARmap, and ST) are included to evaluate the scalability of scCube. Multiple evaluation metrics are applied to assess the simulation performance of scCube and other simulators. In addition, the benchmark results of the relevant SRT computational tools on the real-world data and scCube-simulated data are compared to further demonstrate the accuracy of scCube simulations.
3. The reference-free spatial pattern simulation of scCube is more diverse. First, except for the spatial autocorrelation function in the original version, the updated scCube provides a series of customized spatial pattern simulation functions to simulate complex spatial structures that are more biologically interpretable, such as mixed populations, clusters, cell rings, blood vessels, and so on. Second, the updated scCube also provides a cell subtype spatial pattern simulation function for generating within-cell-type spatial patterns to account for intra-cell-type gene expression variation.
4. The applications of scCube in benchmarking SRT computational tools are more comprehensive. In addition to spot deconvolution, gene imputation, and resolution enhancement methods, scCube has also been applied to benchmark other methods such as spatial domain identification and cell-cell interaction inference, further demonstrating the broad application potential of scCube.

We hope this edition will satisfy the reviewers and address all the concerns to win their approval for the publication of our manuscript. Please find detailed responses below.

Reviewer #1 (Remarks to the Author):

Reviewer's summary

Accurate simulation of pseudo spatial transcriptomics (ST) datasets is crucial for enhancing downstream data analyses. This study proposed a simulator called scCube for simulating multiple spatial variability in spatial resolved transcriptomics and producing unbiased simulated ST data. Benchmark results indicated that the variational autoencoder (VAE) framework-based model outperformed other previously published models. The manuscript shows an interesting methodology for ST analysis. However, the reviewer has the following concerns that need to be addressed:

Response: We thank the reviewer for the positive feedback and constructive critiques. We have significantly improved scCube based on your professional comments and suggestion. All significant modifications are marked in blue in the revised manuscript. We hope this edition will address your concerns.

1. In reference strategy simulation, the authors selected a 10X Visium dataset (Figure S3) as the benchmark dataset. However, their test strategy is similar to the MERFISH dataset because they treated each Visium spot as an individual entity to generate new data. It's worth noting that each spot in the Visium dataset is a mixture of multiple cells. In many scenarios, our reference Visium slice lacks annotation information, whereas we only have annotated scRNA-seq reference data (an example of which can be found in DOI: <https://doi.org/10.1038/s41588-021-00911-1> or PMID: 34493872, as the authors provided both Visium data and scRNA-seq data from human breast cancer samples). I am curious how scCube leverages these references to simulate new similar pseudo slices that closely resemble the Visium data.

Response: Thanks for your constructive comments. Sorry that we had not clearly described how scCube simulates the spot-based SRT data (e.g., ST and 10X Visium) in the reference-based simulation. Although in the manuscript we chose a DLPFC 10X

Visium SRT dataset with spatial domain annotations as the benchmark dataset and used the spatial domain annotations as the “ground truth” in the simulations, we would like to emphasize that the reference-based simulation step of scCube does not necessarily require such annotations. When the spatial reference lacks these annotation information (e.g., spatial domain or pathological annotation), users can perform the unsupervised or spatial domain clustering steps first to provide the cluster label for each spot in the spatial reference, and then perform the reference-based simulation of scCube later. We demonstrated this in detail on the breast cancer 10X Visium SRT datasets you mentioned.

Figure R1. The 10X Visium SRT data of breast cancer generated by Wu et.al. The spots were colored with the pathological annotation provided by the authors, the unsupervised clustering results with different resolutions, and the spatial domain clustering results.

As illustrated in **Figure R1**, we selected four SRT datasets (CID44971, CID4535, CID4465, and CID4290) and performed the reference-based simulation of scCube using the pathological annotation provided by the authors, three unsupervised clustering results with different clustering resolutions (implemented by Seurat,

resolution = 0.3, 0.5, and 0.7), and one spatial domain clustering result (implemented by BayesSpace) as the “ground truth”, respectively. As shown in **Figure R2-4**, scCube accurately simulated the spatial expression patterns of genes in all four SRT datasets, whether using the annotation information or the clustering results as the “ground truth”, and the average PCC values between the real spatial expression patterns of all genes and the corresponding simulated spatial expression patterns are both greater than 0.9 (**Figure R5** and **Table R1**).

Figure R2. The spatial expression patterns of eight representative genes in the ground truth

(CID44971) and the simulated data generated by scCube using different annotation information.

Figure R3. The spatial expression patterns of eight representative genes in the ground truth (CID4535) and the simulated data generated by scCube using different annotation information.

Figure R4. The spatial expression patterns of four representative genes in the ground truth (a, CID4465; b, CID4290) and the simulated data generated by scCube using different annotation information.

Figure R5. Boxplots of PCC values between the real spatial expression patterns of all genes and the corresponding simulated spatial expression patterns across four breast cancer 10X Visium SRT datasets. The simulated data were generated by scCube using different annotation information.

Table R1. The average PCC values between the real spatial expression patterns of all genes and the corresponding simulated spatial expression patterns across four breast cancer 10X Visium SRT datasets.

Datasets	Pathological annotation	Louvain (res = 0.3)	Louvain (res = 0.5)	Louvain (res = 0.7)	BayesSpace
CID44971	0.923	0.918	0.926	0.934	0.923
CID4535	0.945	0.934	0.943	0.933	0.938
CID4465	0.938	0.936	0.933	0.941	0.940
CID4290	0.946	0.944	0.948	0.948	0.941

In addition, we fully agree with you that each spot in the Visium dataset is a mixture of multiple cells. Therefore, in order to further verify the accuracy of the simulated data generated by scCube, we utilized RCTD to deconvolute the real-world data and scCube-simulated data, respectively. As shown in **Figure R6**, the deconvolution results on the real-world data and scCube-simulated data are highly consistent, with the average PCC values greater than 0.98.

In summary, all these results suggest that scCube is robust to the choice of “ground truth” and can still accurately simulate the SRT data whose annotation information is lacking without the need for additional data, such as the annotated scRNA-seq reference data. The results have been included in the supplementary materials and discussed in regard in the manuscript.

Figure R6. Performance evaluation of scCube based on the spot deconvolution results. a, The deconvolution results by RCTD on the real-world data and scCube-simulated data. **b,** Boxplots of PCC values between the deconvolution results on the real and scCube-simulated data.

2. [line 578] "..., whose numbers are determined by the ground truth in the spatial

reference data.” How do the authors define the “ground truth” in this context, especially when we do not have annotations in the spatial reference data as I mentioned in comment 1?

Response: Thanks for your constructive comments. As answered in Question 1, the “ground truth” can be defined as the spatial domain or pathological annotation (if provided), or just as cluster labels obtained by the unsupervised or spatial domain clustering steps. We have shown that scCube is robust to the choice of “ground truth”. In addition, we have also revised the Method section and the schematic workflow in Figure 1 of the manuscript to include more detailed information for a clearer understanding of the design concept and methodology of scCube (See response to Question 6 for details).

3. The count matrix or logarithmized matrix of many scRNA-seq data or spatial transcriptomics data is highly sparse. I am curious whether the authors have performed any calibrations to induce sparsity in the generated gene expression data.

Response: Thanks for your professional comments. We applied the variational autoencoder (VAE) model to simulate the gene expression profiles of the scRNA-seq or SRT data in the gene expression simulation step, and didn’t perform any calibrations to induce sparsity in the generated gene expression data.

As a deep generative model, VAEs have become a popular tool for single-cell omics data modeling (Lopez et al., *Nat Methods*, 2018; Lotfollahi et al., *Nat Methods*, 2019; Grønbech et al., *Bioinformatics*, 2020; Liao et al., *Nat Commun*, 2022). VAEs learn a compressed latent space representation of the input data through the encoder network, which captures the underlying structure and patterns of the original data, and then reconstructs the data from the latent space back to the original data space through the decoder network. Based on the encoder-decoder architecture, VAEs can accurately simulate various characteristics of the original data, including sparsity. Furthermore, in contrast to classical autoencoders (AEs), VAEs directly learn the posterior distribution over the latent space representation, instead of learning the reconstruction of the original data, which ensures the latent space exhibits favorable continuity characteristics, enabling the generation of new data points that are similar to the original data while also exhibiting differences.

Moreover, we also compared the sparsity of the real and scCube-simulated data across 60 SRT datasets from 3 different sequencing platforms. As shown in **Figure R7-9**, the gene expression data simulated by scCube exhibited a high degree of consistency with the real data in terms of sparsity, further indicating that scCube can accurately preserve the characteristics of the original data without any additional calibrations. The results have been included in the supplementary materials.

Figure R7. Sparsity comparison between real data and scCube-simulated data across 36 human breast cancer ST data.

Figure R8. Sparsity comparison between real data and scCube-simulated data across 12 human DLPFC 10X Visium data.

Figure R9. Sparsity comparison between real data and scCube-simulated data across 12 mouse hypothalamus MERFISH data.

4. The examples designed in the sections titled “Using scCube to benchmark spot deconvolution methods”, “Using scCube to benchmark gene imputation methods” and “Using scCube to benchmark resolution enhancement methods” make no sense to readers. A primary objective of these sections is to demonstrate the utility of scCube-simulated data for benchmarking spot deconvolution, gene imputation and resolution enhancement. To achieve this, the authors can first use a real-world data, such as MERFISH shown in Figure 2, as well as a pseudo data simulated by scCube for model benchmark, respectively. By comparing the benchmark results between the real-world data and scCube-simulated data, they can establish whether scCube-simulated data is biologically interpretable. If the benchmark results between the real-world data and

scCube-simulated data are similar, it would support the conclusion that scCube-simulated data is biologically meaningful.

Response: Thanks for your professional comments. Let me explain, the purpose of providing the three benchmark examples in the manuscript is **to further demonstrate the unbiased reference-free simulation framework of scCube, which starts from scRNA-seq data**, and its **application** in benchmarking different SRT computational methods.

Currently, due to the lack of ground truth, a widely used approach for assessing the performance of computational methods for the integration of scRNA-seq and SRT data such as spot deconvolution and gene imputation involves utilizing simulated SRT data constructed from scRNA-seq data. However, in most previous studies, this strategy varies in the selection of scRNA-seq data and the approach of constructing the simulated data, and lacks the reproducibility in the description of most simulation steps. Moreover, the existing SRT simulators such as scDesign3 and SRTsim cannot implement this strategy since both of them make the simulations based on real SRT data, which leads to inherent limitations (for example, the limited gene detection and low cellular resolution) of the generated data.

scCube, on the other hand, provides an unbiased framework to help users easily achieve this simulation process. In addition, A desirable feature of scCube is its ability to simulate a series of data only for one specific variable, which allows users to investigate the impact of certain variables on the performance of different computational methods flexibly. For example, in the “Using scCube to benchmark spot deconvolution methods” example provided by the manuscript, we demonstrated in detail how scCube was utilized to generate a series of simulated spot-based SRT data with varying resolutions, enabling a systematic evaluation of the impact of resolution on the performance of different spot deconvolution methods. Therefore, for the above reasons, we believe it makes sense to provide these three benchmark examples in the manuscript to help readers more intuitively understand and use the reference-free strategy of scCube.

Nevertheless, we fully agree with you that the biologically interpretable of scCube-simulated data could be further **evaluated** by comparing the benchmark results between the real-world data and scCube-simulated data. To achieve this, we chose three kinds of spatial computation methods: spot deconvolution, gene

imputation, and spatial domain identification, and comprehensively compared the benchmark results between the real-world data and scCube-simulated data.

As shown in **Figure R10a**, for the spot deconvolution computational methods benchmarking, we aggregated the real and scCube-simulated MERFISH data (Bregma: +0.06) as the SRT data to be deconvolved, respectively, and utilized the MERFISH data from another slice (Bregma: -0.29) as the single-cell reference. Nine spot deconvolution methods, including Cell2location, DestVI, DSTG, RCTD, Seurat, spatialDWLS, SPOTlight, Stereoscope, and Tangram were benchmarked. As shown in **Figure R10b-c**, each method presented a similar deconvolution result in the real and scCube-simulated data, and the benchmark results of all methods between the real data and scCube-simulated data are also similar.

Figure R10. Comparison of benchmark results of spot deconvolution methods between the real and scCube-simulated data. **a**, Schematic of benchmarking spot deconvolution methods. **b**, The deconvolution results of each method over the real (top) and scCube-simulated (bottom) SRT data. **c**, The benchmark results of all methods over the real (top) and scCube-simulated (bottom) SRT data.

As shown in **Figure R11a**, for the gene imputation computational methods benchmarking, we selected the real and scCube-simulated STARmap data (Replicate 1) as the SRT data to be imputed, respectively, and utilized the STARmap data from another replicate (Replicate 2) as the single-cell reference. Seven gene imputation methods, including SpaGE, Tangram, stPlus, Seurat, SpaOTsc, novoSpaRc, and LIGER were benchmarked. As shown in **Figure R11b-c**, each method presented a similar gene imputation result in the real and scCube-simulated data, and the benchmark results of all methods between the real data and scCube-simulated data are also similar.

Figure R11. Comparison of benchmark results of gene imputation methods between the real and scCube-simulated data. **a**, Schematic of benchmarking gene imputation methods. **b**, The gene imputation results of each method over the real (top) and scCube-simulated (bottom) SRT data. **c**, The benchmark results of all methods over the real (top) and scCube-simulated (bottom) SRT data.

For the spatial domain identification computational methods benchmarking, we selected a 10X Visium SRT dataset from the human dorsolateral prefrontal cortex (DLPFC) containing 12 tissue slices and generated the corresponding simulated SRT data for each slice using scCube. Five spatial domain identification methods, including BayesSpace, SpaGCN, STAGATE, DR-SC, and stLearn, as well as one non-spatial unsupervised clustering method Seurat were benchmarked. As shown in **Figure R12a-b**, each method presented a similar clustering result in the real and scCube-simulated data, and the benchmark results of all methods between the real data and scCube-simulated data are also similar.

All the above results indicated that the simulated SRT data generated by scCube is biologically meaningful. The results have been included in the supplementary materials.

Figure R12. Comparison of benchmark results of spatial domain identification methods between the real and scCube-simulated data. a, The clustering results of each method over the real (top) and scCube-simulated (bottom) SRT data (slice: 151670). **b**, The benchmark results of all methods over the real (top) and scCube-simulated (bottom) SRT data.

5. In the section “Simulating multiple variability in SRT data by the reference-free strategy with scCube”, the authors should consider simulating ST examples that are more biologically interpretable, such as a spherical tumor tissue surrounded by normal tissues.

Response: Thanks for your constructive suggestion and we fully agree with you that the more biologically interpretable spatial patterns should be considered in the reference-free simulation. Inspired by spaSim (Feng et al., *Nat Commun*, 2023), a recently published simulator of tissue spatial data, we have added five new functions named ‘generate_pattern_custom_mixing’, ‘generate_pattern_custom_cluster’, ‘generate_pattern_custom_ring’, ‘generate_pattern_custom_stripes’, and ‘generate_pattern_custom_complex’ for user to simulate more realistic tissue cell spatial structures (**Figure R13**) (See Line 748-800 of the Method section in the revised manuscript for details).

```

a
def generate_pattern_custom_mixing(self,
    sc_adata: AnnData,
    generate_cell_num: int = 5000,
    celltype_key: str = 'Cell_type',
    cell_key: str = 'Cell',
    set_seed: bool = False,
    seed: int = 12345,
    spatial_size: int = 30,
    select_celltype: Optional[list] = None,
    prop_list: Optional[list] = None,
    hidden_size: int = 128,
    load_path: str = '',
    used_device: str = 'cuda:0'
):

b
def generate_pattern_custom_cluster(self,
    sc_adata: AnnData,
    generate_cell_num: int = 5000,
    celltype_key: str = 'Cell_type',
    cell_key: str = 'Cell',
    set_seed: bool = False,
    seed: int = 12345,
    spatial_size: int = 30,
    select_celltype: Optional[list] = None,
    cluster_size_list: list = [],
    cluster_prop_list: list = [],
    infiltration_prop_list: list = [],
    background_celltype: list = [],
    background_prop: Optional[list] = None,
    hidden_size: int = 128,
    load_path: str = '',
    used_device: str = 'cuda:0'
):

c
def generate_pattern_custom_ring(self,
    sc_adata: AnnData,
    generate_cell_num: int = 5000,
    celltype_key: str = 'Cell_type',
    cell_key: str = 'Cell',
    set_seed: bool = False,
    seed: int = 12345,
    spatial_size: int = 30,
    select_celltype: Optional[list] = None,
    ring_size_list: list = [],
    ring_prop_list: list = [],
    infiltration_prop_list: list = [],
    background_celltype: list = [],
    background_prop: Optional[list] = None,
    hidden_size: int = 128,
    load_path: str = '',
    used_device: str = 'cuda:0'
):

d
def generate_pattern_custom_stripes(self,
    sc_adata: AnnData,
    generate_cell_num: int = 5000,
    celltype_key: str = 'Cell_type',
    cell_key: str = 'Cell',
    set_seed: bool = False,
    seed: int = 12345,
    spatial_size: int = 30,
    select_celltype: Optional[list] = None,
    v1_list: list = [None, None],
    v2_list: list = [None, None],
    stripe_size_list: list = [],
    stripe_celltype_list: list = [],
    stripe_purity_list: list = [],
    infiltration_celltype_list: list = [],
    infiltration_prop_list: list = [],
    background_celltype: list = [],
    background_prop: Optional[list] = None,
    hidden_size: int = 128,
    load_path: str = '',
    used_device: str = 'cuda:0'
):

e
def generate_pattern_custom_complex(self,
    sc_adata: AnnData,
    spa_pattern_base: DataFrame,
    spa_pattern_add: DataFrame,
    celltype_key: str = 'Cell_type',
    cell_key: str = 'Cell',
    background_celltype: list = [],
    hidden_size: int = 128,
    load_path: str = '',
    used_device: str = 'cuda:0'
):
    
```

Figure R13. New functions in the reference-free spatial pattern simulation of scCube.

As illustrated in **Figure R14a**, users can apply the first four functions to generate a series of biologically interpretable spatial basis patterns, including unstructured mixed cell populations, clustered cell populations, cell rings encircling tissue structures, and some external structures such as vessels (Feng et al., *Nat Commun*, 2023). In addition, users can further apply the ‘generate_pattern_custom_complex’ function to flexibly simulate the highly customized complex spatial patterns by combining different types as well as numbers of basis patterns (**Figure R14b**).

We next applied scCube to the simulation of the tumor-immune microenvironment of 3 triple negative breast cancer (TNBC) patients, which corresponded to three archetypical subtypes of tumor-immune interactions: cold, mixed, and compartmentalized, respectively (**Figure R14c**). It has been reported that the cold subtype shows the low immune infiltration, the mixed subtype shows the high mixability between tumor and immune cells, and the compartmentalized subtype shows a series of regions comprised predominantly of either immune or tumor cells (Keren et al., *Cell*, 2018). As shown in **Figure R14c**, the spatial patterns simulated by scCube were very similar to the corresponding tumor-immune microenvironments. Furthermore, unlike spaSim, which can only generate spatial pattern images, scCube also combines this customized reference-free spatial pattern simulation step with the gene expression simulation step to generate the corresponding gene expression profiles that match the simulated spatial patterns, which more comprehensively simulates the real-world SRT data (**Figure R14d-e**).

We have updated the codes and documents on GitHub, and the results have been included in the revised manuscript. We have also acknowledged the spaSim in the Acknowledgements section in the revised manuscript.

Figure R14. Using scCube to flexibly simulate the biologically interpretable spatial patterns. a and b, The biologically interpretable spatial basis (a) and complex (b) patterns simulated by scCube. **c,** The real and scCube-simulate tumor-immune microenvironments of mixed, compartmentalized, and cold subtypes of TNBC. **d and e,** The spatial expression patterns of immune (d) and tumor (e) marker genes in the scCube-simulated SRT data.

6. The descriptions for data simulation still lack detailed information in terms of Figure 1 and the Methods section. For instance, it is not clear how the authors implemented cell type features to simulate ST data.

Response: Thanks for your treasurable comments and we are sorry for the lack of detailed information in the Method section and Figure 1. We have extensively reorganized the schematic workflow of scCube as follows.

As shown in **Figure R15** below, the scCube model consists of two components, 1) gene expression simulation, and 2) spatial pattern simulation. The spatial pattern simulation further includes two strategies: reference-based and reference-free. In the gene expression simulation step, scCube applies the variational autoencoder (VAE) model to generate the gene expression profiles of different cell/spot populations (for example, the cell types, spatial domains, pathological regions, etc.). Notably, for each population, scCube can generate gene expression profiles with a customized number of cells (or spots) based on user settings (**Figure R15a**).

In the spatial pattern simulation step, if the spatial reference is provided, scCube will apply the reference-based strategy to construct a mapping between the cells (or spots) in the generated data and the positions in the spatial reference through the optimal transport algorithm and map the generated cells (or spots) to positions with the maximum likelihood of spatial origin (**Figure R15b**). Otherwise, scCube will apply the reference-free strategy to simulate specific spatial patterns for each population in the generated data. As shown in **Figure R15c**, after randomly generating spatial positions with the same number as the cells (or spots) in the generated data, users can further choose to generate random or customized spatial patterns. For the former, scCube first uniformly divides the spatial range containing all cell positions into $n \times n$ grids, and the overall distribution V of all grids is assumed to follow a multivariate normal distribution. Subsequently, for a target population, scCube randomly samples the same proportion of grids based on the proportion of this population in the

generated data, and assigns all cells (or spots) which belong to this population to the spatial positions in the sampled grids. This step repeats c times (c is set by users and does not exceed the total number of populations in the generated data), and cells (or spots) in the remaining population are randomly assigned to spatial positions in the remaining grids. For the latter, scCube provides four functions to optionally generate a series of biologically interpretable spatial basis patterns for each population, including unstructured mixed cell populations, clustered cell populations, cell rings encircling tissue structures, and some external structures such as vessels, and users can further flexibly simulate the highly customized complex spatial patterns by combining different types as well as numbers of basis patterns.

The detailed description of the data simulation can be found in Line 638-800 of the Method section in the revised manuscript.

Figure R15. Schematic workflow of scCube. **a**, Conceptual framework of gene expression simulation step with scCube. A variational autoencoder (VAE) model is applied to simulate the gene expression profiles of cell (or spot) populations in scRNA-seq (or SRT) data. **b** and **c**, Conceptual framework of spatial pattern simulation step with scCube. In the reference-based simulations (**b**),

a mapping between the cells (or spots) in generated data and the positions in the spatial reference is first constructed using the optimal transport algorithm, and the generated cells (or spots) then are mapped to positions with the maximum likelihood of spatial origin. In the reference-free simulations (c), the spatial positions with the same number as the cells (or spots) in the generated data are randomly generated first. Then, scCube generates the random spatial patterns with the default spatial autocorrelation function, or the customized complex spatial patterns by combining different types as well as numbers of basis patterns.

7. The Figure 1 can be improved by including more information, such as how scCube integrates gene expression simulation and spatial pattern simulation for downstream analysis.

Response: Thanks for your valuable suggestion. As answered in Question 6, we have extensively reorganized the schematic workflow of scCube, and included detailed information about the data simulation of scCube in Figure 1 and the Method section of the revised manuscript. Besides, we have also added a flow diagram to visually show how scCube simulated SRT data based on the gene expression simulation and spatial pattern simulation.

As illustrated in **Figure R16** below, by combining the simulated gene expression profiles and spatial patterns, scCube can generate unbiased SRT data corresponding to diverse spatial variations, such as the shape and continuity of the spatial pattern of cell (or spot) populations, the resolution for spot-based SRT data, the targeted gene number and type for imaging-based SRT data, the dimension of the tissue slice, and so on.

The flow diagram has been included in the supplementary materials.

Figure R16. Flow diagram to simulate various SRT data by scCube.

8. In Figure 3a and 3b, a 5x5 image matrix displaying the different combinations of λ and δ will better illustrate how these two parameters impact the simulation results.

Response: Thanks for your professional comments. We have updated the Figure in the

manuscript, as reflected below:

Figure R17. The spatial patterns with different fuzzy degree and continuity generated by scCube.

As illustrated in **Figure R17**, the different combinations of λ and δ will form spatial patterns with various fuzzy degree and the continuity.

9. [line 17-19] “The rapid advancement of computational methods for spatially resolved transcriptomic (SRT) data analysis raises the need for unbiased simulated datasets for evaluation and validation.” The causation appears somewhat unclear. The authors should offer a more compelling justification to clarify the necessity of simulating SRT data.

Response: Thanks for your professional comments. We have described it clearly in the

revised manuscript, as reflected below:

“A pressing challenge in spatially resolved transcriptomic (SRT) is to benchmark the computational methods. A widely-used approach involves utilizing simulated data. However, biases exist in terms of the currently available simulated SRT data, which seriously affects the accuracy of method evaluation and validation. Herein, we present scCube...”

10. [line 145] “PCC” should be defined at its first occurrence.

Response: Thanks for your kind reminding. We have defined “PCC” at its first occurrence, as reflected in the revised manuscript:

“The simulation performance was evaluated by the Pearson correlation coefficient (PCC)...”

Reviewer #2 (Remarks to the Author):

Reviewer's summary

This manuscript introduces scCube, a simulator for spatially resolved transcriptomics (SRT) data, offering versatile ground truths with editable resolution, spatial pattern continuity, and tissue dimension. scCube is designed for benchmarking SRT computational tools, including deconvolution, gene imputation, and resolution enhancement. While scCube holds promise for the SRT community, the manuscript requires refinement for a clearer understanding of the methodology. Additionally, suggestions for both methodology and results are provided below.

Response: We thank the reviewer for the positive feedback and constructive critiques. We have significantly improved scCube based on your professional comments and suggestion. All significant modifications are marked in blue in the revised manuscript. We hope this edition will address your concerns.

****Major Points:****

****Results:****

1. Define the evaluation metric PCC used in the comparison with other simulators. The simulators in row 1 of Fig. 2 perform similarly but show different PCC quantities. Consider incorporating multiple metrics to assess how well synthetic data mimics real data.

Response: Thanks for your valuable advice. We apologize for the lack of a clear definition of the evaluation metric PCC. As shown in **Figure R18**, the evaluation metric PCC is defined as “the Pearson correlation coefficient between the gene expression vector for each gene across spatial positions in real and simulated data”.

Figure R18. Schematic workflow of performance evaluation of different simulators.

Figure R19. The spatial expression patterns of three representative marker genes of ependymal cells in the ground truth and the simulated data generated by scCube, SRTsim, scDesign3, and ZINB-WaVE.

Also, we are sorry for the visual misunderstanding of Figure 2 in the original

manuscript. In Figure 2a of the original manuscript, we showed the spatial expression pattern of the marker gene of inhibitory neurons, *Gad1*, on the simulated data generated by different simulators. Since inhibitory neurons make up the majority of this benchmark dataset and are widely and uniformly distributed throughout the space, the spatial expression patterns of their marker genes appeared visually similar across different simulated datasets. Conversely, as illustrated in **Figure R19** above, for other cell types with more specific spatial distribution (for example, ependymal cells in this dataset), differences in the spatial expression patterns of their marker genes across different simulated data can be visually observed more clearly.

In addition, we fully agree with you that considering incorporating multiple metrics to comprehensively evaluate the simulation performance of scCube with other simulators. To achieve this, we introduced the mean absolute error (MAE) to further evaluate the similarity of the real and simulated data. Specifically, we calculated the PCC and MAE values between the gene expression vector for each gene across spatial positions in real and simulated data generated by each simulator across seven benchmark datasets. As shown in **Figure R20-21**, scCube outperformed other SRT and single-cell simulators over most of the benchmark datasets, with the highest average PCC values and the lowest average MAE values (except for the human DLPPC 10X Visium and zebrafish embryo Stereo-seq datasets) between the real spatial expression patterns of all genes and the corresponding simulated spatial expression patterns. These results demonstrated that scCube can accurately simulate SRT data based on the reference-based strategy.

Figure R20. Performance comparison of scCube with other simulators across seven benchmark datasets. **a** and **b**, The average PCC (**a**) and MAE (**b**) values between the real spatial expression patterns of all genes and the corresponding simulated spatial expression patterns generated by each simulator. The asterisk represents the top-ranked method for each dataset. NA, not available.

Figure R21. Boxplots of PCC and MAE values between the real spatial expression patterns of all genes and the corresponding simulated spatial expression patterns generated by scCube and other simulators across seven benchmark datasets. The simulation results for all genes of scDesign3 are not provided in the Stereo-seq data and ST data due to speed constraints of the training step.

2. Evaluate scCube's performance on other SRT sequencing technologies like GeoMX and 10X Xenium, not just 10x Visium and MERFISH.

Response: Thanks for your professional comments. As you suggested, we have compared the simulation performance of scCube with other existing simulators on the SRT data generated by other sequencing technologies, including the 10X Xenium data from human breast cancer (BC), the Stereo-seq data from zebrafish embryo, the STARmap data from mouse V1 neocortex, and the two ST data from human HER2-positive breast cancer (BC) and human skin squamous cell carcinoma (SCC) (**Figure R22**), and the benchmark results were shown in **Figure R20-21** above (See response to Question 1 for details).

We also demonstrated the spatial expression patterns of several representative marker genes in the real and simulated data. As shown in **Figure R23-27**, scCube most accurately preserved the spatial expression patterns of these genes across all five different benchmark datasets.

All the results indicated that scCube is well scalable and can accurately simulate SRT data generated by different sequencing technologies. Relevant analysis and discussion have been included in the revised manuscript.

Figure R22. The five benchmark datasets generated by other SRT sequencing technologies.

Figure R23. Performance comparison of scCube with other simulators over the mouse V1 neocortex STARmap dataset. The spatial expression patterns of six representative cell type marker

genes in the ground truth and the simulated data generated by scCube, SRTsim, scDesign3, ZINB-WaVE, SymSim, scDesign2, Splat, Kersplat, and Splat Simple.

Figure R24. Performance comparison of scCube with other simulators over the human breast cancer (BC) 10X Xenium dataset. The spatial expression patterns of seven representative cell type marker genes in the ground truth and the simulated data generated by scCube, SRTsim, scDesign3, ZINB-WaVE, SymSim, scDesign2, Splat, Kersplat, and Splat Simple.

Figure R25. Performance comparison of scCube with other simulators over the zebrafish embryo Stereo-seq dataset. The spatial expression patterns of seven representative cluster marker genes in the ground truth and the simulated data generated by scCube, SRTsim, scDesign3, ZINB-WaVE, SymSim, scDesign2, Splat, Kersplat, and Splat Simple.

Figure R26. Performance comparison of scCube with other simulators over the HER2-positive breast cancer (BC) ST dataset. The spatial expression patterns of six representative domain marker genes in the ground truth and the simulated data generated by scCube, SRTsim, scDesign3, ZINB-WaVE, SymSim, scDesign2, Splat, Kersplat, and Splat Simple.

Figure R27. Performance comparison of scCube with other simulators over the human skin squamous cell carcinoma (SCC) ST dataset. The spatial expression patterns of eight representative domain marker genes in the ground truth and the simulated data generated by scCube, SRTsim, scDesign3, ZINB-WaVE, SymSim, scDesign2, Splat, Kersplat, and Splat Simple.

3. Explore if scCube can help benchmark computational tools for spatial domain identification and spatial cell-cell interaction inference.

Response: Thanks for your constructive comments. As you suggested, we have benchmarked the spatial domain identification and spatial cell-cell interaction inference methods using scCube. For the spatial domain identification computational methods benchmarking, we selected a 10X Visium SRT dataset from the human dorsolateral prefrontal cortex (DLPFC) containing 12 tissue slices and generated the corresponding simulated SRT data for each slice using the reference-based strategy of scCube. Five spatial domain identification methods, including BayesSpace, SpaGCN, STAGATE, DR-SC, and stLearn, as well as one non-spatial unsupervised clustering method Seurat were benchmarked.

Figure R28. Using scCube to benchmark spatial domain identification methods. **a**, The clustering results of each method over the scCube-simulated SRT data (slice: 151670). **b**, Boxplots of adjust rand index (ARI), normalized mutual information (NMI), homogeneity score (HS), and Fowlkes-Mallows score (FMI) values of the five spatial domain identification methods and one non-spatial unsupervised clustering method over the 12 scCube-simulated SRT datasets.

As shown in **Figure R28**, most spatial domain identification methods (STAGATE,

BayesSpace, SpaGCN, and DR-SC) identified a clear layered structure corresponding to different cortical layers in the ground truth. In contrast, the inferred spatial clusters by stLearn and Seurat were distributed almost randomly across spatial positions. These results suggested that the spatial information can help improve the clustering accuracy in spatial domain identification. Overall, STAGATE, BayesSpace, and SpaGCN had the top 1, 2, and 3 ranking average adjusted rand index (ARI), normalized mutual information (NMI), homogeneity score (HS), and Fowlkes-Mallows score (FMI) values over 12 simulated SRT datasets (0.463/0.610/0.639/0.582, 0.416/0.573/0.594/0.548, and 0.394/0.535/0.563/0.526, respectively), followed by DR-SC (0.339/0.489/0.501/0.492), Seurat (0.314/0.433/0.451/0.466), and stLearn (0.201/0.327/0.340/0.376).

For the spatial cell-cell interaction inference methods benchmarking, we selected a STARmap SRT dataset from the mouse V1 neocortex and generated the corresponding simulated SRT data using the reference-based strategy of scCube. One recently published spatially resolved cell-cell communication inference method, SpaTalk, as well as other seven scRNA-seq cell-cell communication inference methods, including CellCall, CellChat, CellPhoneDB, CytoTalk, Giotto, NicheNet, and SpaOTsc were benchmarked. To evaluate the accuracy of different methods in spatial cell-cell interaction inference, we compared the spatial proximity significance and the co-expression level on the cell-cell spatial graph network of the ligand–receptor interactions (LRIs) inferred by each method (Shao et al., *Nat Commun*, 2022).

Figure R29. Using scCube to benchmark spatial cell-cell interaction inference methods.

As shown in **Figure R29**, SpaTalk obtained the highest average $-\log_{10}P$ -value of spatial proximity significance (19.932) and co-expression percent (0.409) for all

inferred LRIs, outperforming other scRNA-seq cell-cell communication inference methods. These results suggested that the spatial information can help to improve the accuracy of inferring spatially proximal LR pairs that mediating cell-cell communication in space.

The benchmark results of spatial domain identification methods and spatial cell-cell interaction inference methods have been included in the supplementary materials.

4. In Fig. S11, compare scCube's performance with other simulators in the same scenario for clarity.

Response: Thanks for your professional comments. We fully agree to compare the performance of scCube with other simulators in the same scenario for clarity. As shown in **Figure R20, R21, R26, and R27** above, scCube still outperformed other simulators over both the two cancer SRT datasets, with the highest average PCC values and the lowest average MAE values between the real spatial expression patterns of all genes and the corresponding simulated spatial expression patterns. These results demonstrated that scCube can also accurately preserve the spatial heterogeneity of gene expression in a much more complex spatial architecture, such as the tumor microenvironment, further suggesting its widespread applicability.

****Methodology:****

1. Assess if scCube allows manual specification of patterns for specific genes in SRT data simulation.

Response: Thanks for your constructive comments. It does sound interesting that try to manually specify patterns for specific genes in SRT data simulation. However, we should point out first that this simulation strategy does not fit the design concept of scCube. As mentioned in the manuscript, the SRT data simulation of scCube starts with the real scRNA-seq or SRT data. scCube first simulates the gene expression profiles of cells (or spots) of specific cell types (or spatial domains) in the gene expression simulation, and then assigns a specific spatial distribution to each cell type (or spatial domain) through the reference-based or reference-free strategy. This “cell type (or spatial domain)-based” simulation workflow of scCube can preserve the gene-gene

relationships present in the real data within the simulated SRT data. For example, as shown in **Figure R30**, CD3D and CD3E, both immune cell markers, showed similar spatial expression patterns consistent with expectations in the scCube-simulated SRT data, as did the tumor cell markers KRT18 and KRT19.

Figure R30. The scCube-simulated SRT data preserves the gene-gene relationships present in the real data.

However, if the spatial expression patterns of a set of specific genes are directly manual-specified in the simulation, the following two questions inevitably arise: 1) How to ensure that all genes within the selected gene sets are indeed correlated to each other in the real data? and 2) How to simulate the spatial expression patterns of genes other than the selected genes? For the above reasons, we didn't provide this function in scCube.

Nevertheless, we did consider how to specify the presence or absence of spatial expression patterns for a specific set of genes through a more reasonable strategy when designing scCube. In brief, scCube achieves this function by specifying specific cell types rather than specific genes (See Line 267-289 and Figure 3b-f in the revised manuscript). As shown in **Figure R31** below, scCube provides a parameter named 'spatial_cell_type' in the 'generate_pattern_random' function for users to simulate spatial patterns only for a specific set of cell types, which in turn achieves the manual specification of the types of spatial expression patterns for the marker genes of these cell types.

```

def generate_pattern_random(self,
                            generate_sc_data: DataFrame,
                            generate_sc_meta: DataFrame,
                            celltype_key: str = 'Cell_type',
                            set_seed: bool = False,
                            seed: int = 12345,
                            spatial_cell_type: Optional[list] = None,
                            spatial_dim: int = 2,
                            spatial_size: int = 30,
                            delta: float = 25,
                            lamda: float = 0.75,
                            is_split: bool = True,
                            split_coord: str = 'point_z',
                            slice_num: int = 5,
                            ):

```

Figure R31. The parameters in the 'generate_pattern_random' function.

As illustrated in **Figure R32a**, we used this function of scCube to generate two simulated SRT datasets, in which astrocytes were manually specified to be present (Dataset 1) and absent (Dataset 2) spatial patterns, respectively. In this way, the spatial expression patterns of marker genes of astrocytes were also manually specified into two types. As illustrated in **Figure R32b-c**, compared with Dataset2, the marker genes of astrocytes in Dataset1 showed obvious spatial expression patterns and had significantly higher Moran's I values. Overall, this is a fallback but more reasonable strategy compared with manually specifying a set of specific genes directly.

Figure R32. Using scCube to manually specified spatial patterns for specific cell types and their marker genes. **a.** Two simulated SRT datasets generated by scCube, in which astrocytes were manually specified to be present (Dataset 1) and absent (Dataset 2) spatial patterns. **b.** Boxplots of Moran's I values for spatial expression patterns of marker genes of astrocytes in two simulated datasets. The P -value is calculated with the two-sided Wilcoxon rank-sum test. **c.** The spatial expression patterns of marker genes of astrocytes in Dataset 1 (top) and Dataset 2 (bottom).

2. Consider incorporating flexibility in scCube for generating within-cell-type spatial patterns to account for intra-cell-type gene expression variation.

Response: Thanks for your valuable comment. As you suggested, we have added a new function named 'generate_subtype_pattern_random' for users to flexibly simulate within-cell-type spatial patterns (**Figure R33a**). As illustrated in **Figure R33b-c**, we chose a scRNA-seq dataset from MCA (Han et al., *Cell*, 2018) as a demonstration, in which macrophages exist in two subtypes: 'macrophage_Ace high' and 'macrophage_S100a4 high'.

In the previous version, scCube simulated a specific spatial pattern for each cell

type without considering heterogeneity with the cell types, so that the spatial distribution of each cell subtype and the spatial expression patterns of its marker genes followed random distributions (**Figure R33d-e top**). In the updated version, the ‘generate_subtype_pattern_random’ function provided by scCube can help users perform further spatial pattern simulation for each cell subtype within the specific cell type to better account for intra-cell-type gene expression variation in space (**Figure R33d-e bottom**). Moreover, as this function can be added separately, users can flexibly choose whether to consider the heterogeneity of cell subtypes when performing the spatial pattern simulation for cell types.

We have updated the codes and documents on GitHub, and the results have been included in the revised manuscript.

Figure R33. Using scCube to flexibly simulate the within-cell-type spatial patterns. **a.** The New function for cell subtype spatial pattern simulation of scCube. **b.** The simulated gene expression data of a scRNA-seq dataset from MCA by scCube, in which macrophages contain two subtypes. **c.** The simulated spatial pattern of cell types generated by scCube. **d** and **e.** The spatial distributions of macrophage subtypes (**d**) and the spatial expression patterns of their marker genes (**e**) in the settings of “cell subtypes without spatial patterns” (top) and “cell subtypes with spatial patterns” (bottom).

3. Clarify the reference-free spatial pattern simulation process. Provide a visual aid (e.g., a plot similar to Fig. S1) to enhance understanding, especially for the

representation of the overall distribution denoted as V in line 604.

Response: Thanks for your treasurable comments and we apologize for the lack of detail information on reference-free spatial pattern simulation process. We have extensively reorganized the schematic workflow of scCube and added a visual aid of the reference-free spatial pattern simulation process to enhance understanding (**Figure R34** below).

Figure R34. Schematic workflow of scCube. **a**, Conceptual framework of gene expression simulation step with scCube. A variational autoencoder (VAE) model is applied to simulate the gene expression profiles of cell (or spot) populations in scRNA-seq (or SRT) data. **b** and **c**, Conceptual framework of spatial pattern simulation step with scCube. In the reference-based simulations (**b**), a mapping between the cells (or spots) in generated data and the positions in the spatial reference is first constructed using the optimal transport algorithm, and the generated cells (or spots) then are mapped to positions with the maximum likelihood of spatial origin. In the reference-free simulations (**c**), the spatial positions with the same number as the cells (or spots) in the generated data are randomly generated first. Then, scCube generates the random spatial patterns with the default spatial autocorrelation function, or the customized complex spatial patterns by combining different types as well as numbers of basis patterns.

As shown in **Figure R34c**, scCube further provides two strategies in the reference-free spatial pattern simulation. After randomly generating spatial positions with the same number as the cells (or spots) in the generated data, users can either choose to generate the random or customized spatial patterns. For the former, scCube uniformly divides the spatial range containing all cell positions into $n \times n$ grids first. To encourage neighborhood similarity in the composition of population in grids, we assume that the overall distribution V of grids follows a multivariate normal distribution:

$$V \sim MVN(0, \Sigma)$$

Where the $n \times n$ covariance matrix Σ models the spatial correlation among the grids based on the Euclidean distance between their spatial coordinates:

$$\Sigma = e^{-\left(\frac{D}{\delta}\right)^2}$$

Where D is the Euclidean distance matrix and δ is the autocorrelation parameter. A larger δ will generate spatial patterns with greater connectivity. Subsequently, for a target population, scCube randomly samples the same proportion of grids based on the proportion of this population in the generated data, and assigns all cells (or spots) which belongs to this population to the spatial positions in the sampled grids. This step repeats c times (c is set by users and does not exceed the total number of populations in the generated data), and cells (or spots) in the remaining population are randomly assigned to spatial positions in the remaining grids. Furthermore, scCube also provides an additional parameter, λ , to generate spatial patterns with varying degrees of fuzziness by randomly swapping the spatial coordinates of $1 - \lambda$ percent of the cells.

For the latter, scCube provides four functions to optionally generate a series of more biologically interpretable spatial basis patterns for each population, which were suggested by the Reviewer 1 (See response to Question 5 for Reviewer 1 on Page 16-18 for details), including unstructured mixed cell populations, clustered cell populations, cell rings encircling tissue structures, and some external structures such as vessels (Feng et al., *Nat Commun*, 2023). Moreover, users can further flexibly simulate the highly customized complex spatial patterns by combining different types as well as numbers of basis patterns.

The detailed description of the data simulation can be found in Line 638-800 of

the Method section in the revised manuscript.

4. Explain the choice of the VAE model in the initial step of generating scRNA-seq data. Address why the VAE model was chosen and whether alternative models could be considered.

Response: Thanks for your constructive comments. As a deep generative model, VAEs have become a popular tool for single-cell omics data modeling (Lopez et al., *Nat Methods*, 2018; Lotfollahi et al., *Nat Methods*, 2019; Grønbech et al., *Bioinformatics*, 2020; Liao et al., *Nat Commun*, 2022). VAEs learn a compressed latent space representation of the input data through the encoder network, which captures the underlying structure and patterns of the original data, and then reconstructs the data from the latent space back to the original data space through the decoder network. Based on the encoder-decoder architecture, VAEs can accurately simulate various characteristics of the original data. Furthermore, in contrast to classical autoencoders (AEs), VAEs directly learn the posterior distribution over the latent space representation, instead of learning the reconstruction of the original data, which ensures the latent space exhibits favorable continuity characteristics, enabling the generation of new data points that are similar to the original data while also exhibiting differences. For the above two reasons, we chose the VAE model to generate the simulated scRNA-seq or SRT data in the gene expression simulation step.

In addition to VAEs, another class of deep generative models, generative adversarial networks (GANs) and a series of its variants such as conditional GANs (CGANs) and Wasserstein GANs (WGANs) have also been applied for single-cell omics data modeling (Marouf et al., *Nat Commun*, 2020; Xu et al., *Nucleic Acids Res*, 2020; Wang et al., *Bioinformatics*, 2022; Huang et al., *Brief Bioinform*, 2023). To this end, we considered the CGANs model and explored its ability to simulate scRNA-seq or SRT data. As illustrated in **Figure R35a-b and R36a-b**, we simulated the mouse hypothalamus MERFISH data and the mouse V1 neocortex STARmap data using the VAEs and CGANs models, respectively. The results showed that both of these two models could achieve accurate gene expression simulation, where the gene expression profiles of each cell type were highly correlated between the real and generated data.

We further applied the reference-based spatial pattern simulation strategy of scCube to generate the corresponding simulated SRT data from the gene expression

data simulated by these two models. As illustrated in **Figure R35c and R36c**, both of these two simulated SRT data accurately preserved the spatial expression patterns of genes. All the results show that the CGANs model has the same potential to simulate SRT data as the VAEs model.

Figure R35. Simulation of the mouse hypothalamus MERFISH data with the VAEs and CGANs models. a. UMAP visualization of the real and simulated mouse hypothalamus MERFISH data, colored by cell types (left) and data types (right). **b.** The heatmap of the correlation of gene expression profiles for each cell type between real and generated data. **c.** The spatial expression patterns of *Gad1*, *Nnat*, and *Mbp* in the ground truth and the simulated data generated by VAE, and CGAN.

Figure R36. Simulation of the mouse V1 neocortex STARmap data with the VAEs and CGANs models. a. UMAP visualization of the real and simulated mouse V1 neocortex STARmap data, colored by cell types (left) and data types (right). **b.** The heatmap of the correlation of gene expression profiles for each cell type between real and generated data. **c.** The spatial expression

patterns of *Lamp5*, *Mbp*, and *Pcp4* in the ground truth and the simulated data generated by VAE, and CGAN.

We have updated the codes of scCube to add two new functions named 'train_cgan_and_generate_cell' and 'load_cgan_and_generate_cell' for users to try training or loading the CGANs model for gene expression simulation (**Figure R37**). However, it is worth noting that we did not conduct more comprehensive experiments to evaluate the simulation performance of the CGANs model, which is beyond the scope of the main study of this manuscript. Therefore, by default, we still recommend using the VAEs model for gene expression simulation.

```

a
def train_cgan_and_generate_cell(self,
    sc_adata: AnnData,
    celltype_key: str,
    cell_key: str,
    target_num: Optional[dict] = None,
    batch_size: int = 512,
    epoch_num: int = 5000,
    lr: float = 0.0001,
    SemSize: int = 256,
    NoiseSize: int = 256,
    save_model: bool = True,
    save_path: str = '',
    project_name: str = '',
    used_device: str = 'cuda:0'
):
    """
    train CGAN model to generate cells
    :param sc_adata: AnnData of data
    """

b
def load_cgan_and_generate_cell(self,
    sc_adata: AnnData,
    celltype_key: str,
    cell_key: str,
    target_num: Optional[dict] = None,
    batch_size: int = 512,
    SemSize: int = 256,
    NoiseSize: int = 256,
    load_path: str = '',
    used_device: str = 'cuda:0'
):
    """
    Load trained CGAN model to generate cells
    :param sc_adata: AnnData of data
    :param celltype_key: the column name of 'cell types' in meta
    :param cell_key: the column name of 'cell' in meta
    :param target_num: target number of cells to generate, if 'target_num'
    of cell types of the input data
    """
  
```

Figure R37. Two new functions in gene expression simulation of scCube.

****Minor Points:****

1. Correct a typo in Fig. 2d: "Bregma -0.09" should be "Bregma -0.29."

Response: Thanks for your comments, we have corrected the typo in the revised manuscript.

2. In gene expression simulation (line 530), correct the statement: the conditional distribution should be $q_{\phi}(z|x)$, not $p_{\phi}(z|x)$.

Response: Thanks a lot for your kind reminding, we have corrected it in the revised manuscript.

3. Rectify the x-axis range in the bottom-right density plot in Figure 5.a for accuracy.

Response: Thanks a lot for your kind reminding, we have revised the x-axis range in the bottom-right density plot, as reflected below:

Figure R38. The frequency histograms of the cell number per spot for the simulate spot-based SRT data (n=100) generated by scCube.

Reviewer #3 (Remarks to the Author):

Reviewer's summary

This paper presents an algorithm for simulating spatially resolved transcriptomics data. The algorithm aims to preserve spatial expression patterns and generate simulated data with various spatial patterns, resolutions, tissue slice dimensions, and more. The study comprehensively compares the performance of scCube with existing algorithms in terms of the aforementioned spatial variations. Additionally, the paper demonstrates the usefulness of scCube in three benchmarking applications. The idea behind scCube is straightforward and logical, and it demonstrates its ability to simulate unbiased SRT data. However, there are a few points that need to be addressed.

Response: We thank the reviewer for the positive feedback and constructive critiques. We have significantly improved scCube based on your professional comments and suggestion. All significant modifications are marked in blue in the revised manuscript. We hope this edition will address your concerns.

1. scCube consists of two steps: 1) single-cell gene expression simulation and 2) spatial pattern simulation. These steps appear to be independent of each other. It would be interesting to compare the performance of the first step with that of existing single-cell simulation tools. Can existing simulation tools be used as a substitute for the first step of the scCube algorithm?

Response: Thanks for your constructive comments. You are right that the two steps in scCube are independent of each other, and it's indeed very interesting to compare the performance of the gene expression simulation step of scCube with that of existing single-cell simulation tools. As you suggested, we compared the performance of the gene expression simulation step of scCube with other six single-cell simulators (scDesign2, ZINB-WaVE, SymSim, Splat, Kersplat, and Splat Simple) on the mouse hypothalamus MERFISH data and the mouse V1 neocortex STARmap data.

As shown in **Figure R39**, the first step of scCube outperformed other single-cell simulators, where cells belonging to the same cell type in the simulated and real data are highly overlapping on the UMAP plot. In contrast, other single-cell simulators, except scDesign2, failed to accurately simulate the gene expression profiles of each

cell type, possibly because they model the scRNA-seq data as a whole rather than model each cell type individually (Zappia et al., *Genome Biol*, 2017).

Figure R39. Performance comparison of the gene expression simulation step of scCube with other single cell simulators over the mouse hypothalamus MERFISH data (a) and the mouse V1 neocortex STARmap data (b). The UMAP visualization of the real and simulated data are colored by cell types (top) and data types (bottom).

We further applied the reference-based spatial pattern simulation strategy of scCube to generate the corresponding simulated SRT data from the gene expression data simulated by scDesign2 and the first step of scCube, respectively. As illustrated in **Figure R40-41**, although the simulation performance of the “scDesign2 + the second step of scCube” strategy was greatly improved compared with scDesign2 alone, the simulated SRT data generated by the whole workflow of scCube still best preserved the spatial expression patterns of genes in the real data, exhibiting the highest average PCC values for all genes. In summary, we recommend using the whole workflow of scCube to simulate SRT data for accuracy and convenience.

Figure R40. Performance comparison of scCube with scDesign2 and “scDesign2 + the second step of scCube” over the mouse hypothalamus MERFISH data. a. The spatial expression patterns of *Gad1*, *Nnat*, and *Mbp* in the ground truth and the simulated data generated by scCube, “scDesign2 + the second step of scCube”, and scDesign2. b. Boxplots of PCC values for the spatial expression patterns of all genes between the real and simulated data generated by scCube, “scDesign2 + the second step of scCube”, and scDesign2 (scDesign2_o, “scDesign2 only”; scDesign2_c, “scDesign2 + the second step of scCube”).

Figure R41. Performance comparison of scCube with scDesign2 and “scDesign2 + the second step of scCube” over the mouse V1 neocortex STARmap data. a. The spatial expression patterns of *Gad1*, *Nnat*, and *Mbp* in the ground truth and the simulated data generated by scCube, “scDesign2 + the second step of scCube”, and scDesign2. b. Boxplots of PCC values for the spatial expression patterns of all genes between the real and simulated data generated by scCube, “scDesign2 + the second step of scCube”, and scDesign2 (scDesign2_o, “scDesign2 only”; scDesign2_c, “scDesign2 + the second step of scCube”).

2. The authors tested scCube using Slide-seq, 10X, and ST datasets. However, these

datasets are relatively low-resolution and do not provide cell-level gene expression. It would be interesting to see how scCube performs when simulating high-resolution datasets that do provide cell-level gene expression, such as stereo-seq datasets.

Response: Thanks for your insightful comments. We fully agree to evaluate the simulation performance of scCube on high-resolution datasets. As you suggested, we further selected other high-resolution SRT datasets, including the 10X Xenium data from human breast cancer (BC), the STARmap data from mouse V1 neocortex, and the Stereo-seq data from zebrafish embryo (**Figure R42**).

Figure R42. The three additional high-resolution benchmark datasets.

As shown in **Figure R43**, scCube outperformed other SRT and single-cell simulators on all three high-resolution SRT datasets, with the highest average PCC values between the real spatial expression patterns of all genes and the corresponding simulated spatial expression patterns.

Figure R43. Boxplots of PCC values between the real spatial expression patterns of all genes and the corresponding simulated spatial expression patterns generated by scCube and other simulators across three high-resolution benchmark datasets. The simulation results for all genes of scDesign3 are not provided in the Stereo-seq data due to speed constraints of the training step.

We further demonstrated the spatial expression patterns of several representative marker genes in the real and simulated data. As shown in **Figure R44-46**, scCube most accurately preserved the spatial expression patterns of these genes across the three high-resolution benchmark datasets.

All the results indicated that scCube is well scalable and can also accurately simulate high-resolution SRT data.

Figure R44. Performance comparison of scCube with other simulators over the human breast cancer (BC) 10X Xenium dataset. The spatial expression patterns of seven representative cell type marker genes in the ground truth and the simulated data generated by scCube, SRTsim, scDesign3, ZINB-WaVE, SymSim, scDesign2, Splat, Kersplat, and Splat Simple.

Figure R45. Performance comparison of scCube with other simulators over the mouse V1 neocortex STARmap dataset. The spatial expression patterns of six representative cell type marker genes in the ground truth and the simulated data generated by scCube, SRTsim, scDesign3, ZINB-WaVE, SymSim, scDesign2, Splat, Kersplat, and Splat Simple.

Figure R46. Performance comparison of scCube with other simulators over the zebrafish embryo Stereo-seq dataset. The spatial expression patterns of seven representative cluster marker genes in the ground truth and the simulated data generated by scCube, SRTsim, scDesign3, ZINB-WaVE, SymSim, scDesign2, Splat, Kersplat, and Splat Simple.

3. Many existing tools for simulating spatial transcriptomics data take a very long time

and sometimes fail to complete the simulation. It would be helpful if the authors could provide the execution time for scCube in these simulated datasets.

Response: Thanks for your professional comments and we fully agree with you that provide the execution time for scCube in these datasets is very helpful. Specifically, there are two factors that affect the execution time of scCube: the size of the dataset and the number of epochs for training the model in the gene expression simulation. In general, the execution time of scCube increases with the larger number of cells or genes in the simulated data and the greater number of training epochs. However, with larger training epochs, the model is often able to learn the features of the data more effectively and thus achieve more accurate simulation.

Therefore, for the above reasons, we systematically provide the execution time and simulation performance of scCube with different number of training epochs on seven datasets of different sizes. As illustrated in **Figure R47**, with a 24 GB NVIDIA A10 GPU, the execution time of scCube on all datasets increased linearly with the number of training epochs. Specifically, for the mouse V1 neocortex STARmap data (Replicate 1), human BC ST data (Patient G replicate 1), mouse hypothalamus MERFISH data (Bregma: +0.06), human SCC ST data (Patient 2 replicate 1), human DLPFC 10X Visium data (Slice 151673), zebrafish embryo Stereo-seq data (24-hpf), and human BC 10X Xenium data, the execution times of scCube to reach a relatively stable simulation performance (10,000 epochs) were approximately 3.38, 6.12, 10.46, 14.08, 54.94, 58.26, and 564.77 minutes, respectively.

Figure R47. The execution time and simulation performance of scCube with different number of training epochs across seven benchmark datasets of different sizes. The grey background highlights the number of training epochs when scCube reach a relatively stable simulation performance.

Moreover, as shown in **Figure R48**, the execution time of scCube mainly concentrated in the training step. Therefore, if there are trained models provided by scCube (<https://github.com/ZJUFanLab/scCube/blob/main/tutorial/statistics.md>) matching the target SRT data, users can directly download the corresponding models to generate the simulated data within a very short period of time without additional training steps. The results have been included in the supplementary materials.

Figure R48. The execution time of training and generation steps of scCube with different number of training epochs on seven datasets of different sizes.

4. In Figure 7a, it would be helpful if the authors could explain why the UMAP plot, calculated based on randomly selected genes, is still able to accurately distinguish cell types.

Response: Thanks for your thought-provoking comments. It's indeed interesting that the UMAP plot calculated based on randomly selected genes can still accurately distinguish cell types. To explain it, we performed 10 repetitions on the random gene selection step and investigated the expression of these genes in each cell type. As shown in **Figure R49**, even though these 200 genes were randomly selected, many of them still showed robust differential expression in different cell types (average 146.5, 86.3, 39.5, and 17.6 genes when the \log_2FC threshold was 0.25, 0.5, 1.0, and 1.5, respectively), which may be the reason why the UMAP plot is still able to distinguish different cell types.

Figure R49. The number of genes in common between the differentially expressed genes (DEGs) of cell types calculated with Seurat and the randomly selected genes.

To prove our conjecture, we further excluded these shared genes and performed UMAP on the remaining genes. As shown in **Figure R50**, when these differentially expressed genes (DEGs) were excluded, the ability of UMAP to distinguish different cell types started to decline; and, when the threshold of \log_2FC dropped to 0.25, the UMAP calculated based on the remaining genes could no longer distinguish cell types. All the results suggested that it was the DEGs among the 200 randomly selected genes that enabled different cell types to still be accurately distinguished on the UMAP plot.

Figure R50. The UMAP plot calculated based on remaining genes after excluding differentially expressed genes (DEGs) according to different \log_2FC thresholds.

Reviewer #1 (Remarks to the Author):

The manuscript has significantly improved and has addressed the majority of the previous comments and concerns. However, there are still two minor concerns that need to be addressed before it can be published:

1. [My previous comment 4] I think the authors didn't understand my point of view. Now I clarify it more clearly and I hope the authors will address it in this revision:

While the authors provide a detailed explanation, their argument relies on the assumption that scCube simulated data accurately mimics real-world ST data. Therefore, it is crucial for them to substantiate this assumption. As a suggestion, I propose that the authors consider selecting MERFISH data as an example. MERFISH data encompasses both cell annotations and spatial information simultaneously, allowing the creation of a real-world resampled ST dataset with ground truth. By comparing benchmark results between the real-world resampled ST data and scCube-simulated data, we can assess the similarity between the two. Consistent results would indicate that scCube-simulated data closely resembles real data.

The objective of my suggestion is not to determine which deconvolution methods are superior but to leverage these methods to demonstrate that data simulated by scCube is biologically more meaningful. If the authors fail to demonstrate the superiority of scCube, the question arises: why choose scCube for simulating data in the deconvolution benchmark? It's noteworthy that in many deconvolution algorithm papers, authors often randomly simulate ST spots using scRNA-seq data without assessing whether their simulated data aligns with real-world ST data.

2. [My previous comment 3] I have a straightforward suggestion regarding calibration: the authors can set a counts/normalized count threshold and reset counts below this threshold to zero. The specific threshold can be defined based on the characteristics of your ST platform.

Reviewer #1 (Remarks on code availability):

I believe the code is OK and is capable of generating the results presented in the manuscript. The README file has included enough instructions for installing and running the application.

Reviewer #2 (Remarks to the Author):

Please see the attached pdf.

Reviewer #2 (Remarks on code availability):

Please see the review pdf file.

Reviewer #2 Attachment on the following page

Our major concern is that the comparison results in the manuscript are not informative nor fair. There are two major reasons. First, the evaluation favors overfitting of the training data, which is treated as the ground truth by the proposed simulator scCube and an existing simulator SRTsim. Hence, these two simulators far outperformed the other simulators that do not assume the training data to be ground truths. Second, the Pearson correlation coefficient is not an informative metric because it has a low value even between two splitted datasets with the same underlying spatial pattern (if random noise is removed). The first reason also made us wonder how scCube and SRTsim can be useful because if users want to use training data as the ground truth, they can just use the training data instead of the simulated data.

1. Major

- a. It is unclear how the proposed simulator scCube can assist method benchmarking. In the comparison results, scCube and a previous simulator SRTsim performed way better than other simulators. The reason is that only scCube and SRTsim assign simulated single cells to spatial locations by mimicking real spatial data, assuming that the real data is noiseless. In other words, scCube and SRTsim treats real spatial data as the ground truth. Then it is expected that the simulated spatial data well mimic the real spatial data. However, this approach has the overfitting issue. We verified this issue by a data splitting experiment (code at the bottom of this review). In the experiment, we split a real spatial dataset into two datasets using a statistical method countsplit ([Neufeld et al., **Biostatistics**, 2022]), so the two datasets are expected to have the same true gene expression. However, when we calculated the Pearson correlation coefficient (PCC), the major evaluation metric used in the scCube manuscript, the PCC between the two datasets was extremely low (less than 0.1), even though the spatial patterns of these two samples resemble each other (see the figures at the bottom of this review). This indicates that the high PCC values obtained by SRTsim and scCube were due to overfitting; that is, the simulated spatial data resembles the training data even more than the test data does.
- b. Because of this overfitting issue, it is unclear why we need simulators like scCube and SRTsim, because they do not provide any more ground truths beyond the real spatial data they are trained on. The fundamental question is: how can scCube assist method benchmarking better than using real data? If real data is considered as the ground truth, why do we need simulation?
- c. For the above reason, among the simulators compared in the manuscript, only SRTsim is comparable to scCube because they both treat the real spatial data as the ground truths and aim to mimic the training data as much as possible (the overfitting issue). As our experiment indicates, data splitting should be used to evaluate the overfitting issue, and any evaluation metric should be applied to the test data for a fair comparison.
- d. Moreover, the PCC is not a suitable evaluation metric because it is well known to be sensitive to outlying values. Also, our experiment above shows that the PCC

is not informative because it is so low even between two datasets splitted from one dataset.

- e. The implementation of baseline simulators is unfair. We examined the code used by the authors for generating synthetic data using scCube and other baseline simulators. We found that the input real data were pre-normalized (and/or log-transformed) and thus unsuitable for methods directly modeling read counts. We deem it necessary for the authors to provide details of input data and the normalization process for each simulator in the comparison.
2. Minor
- a. Provide an explanation for the presence of NAs in scDesign3's PCC and MAE values in Figure 2.a and Figure R20.
 - b. In Figure 2.a, the author uses color to indicate rank, which does not accurately reflect the actual differences in PCC. For example, a PCC value of $2.50e-6$ is represented by a slightly darker blue, which could give a misleading impression of its actual weakness.
 - c. Line 187 should be "Bregma +0.06" slice.
 - d. The ground truth provided by scCube is limited to the cell-type level since scCube assigns each spot a cell type when generating synthetic data. We suggest the authors address this limitation in the discussion section.

Experiment:

1. Round the values of the real dataset in Supplementary Information page 6.
2. Utilize the **countsplit** method to divide the real dataset into two datasets
3. Plot the gene expression and calculate the PCC.

Results:

It is evident that even though the splitted real dataset exhibits a similar visual spatial pattern, the PCC values are quite low. This suggests that PCC may not be a suitable metric for this analysis.

Code:

```

library(SeuratDisk)
library(Seurat)
library(countsplit)
library(readr)
library(ggExtra)

# This dataset is downloaded from the url for
# mouse_hypothalamus_MERFISH_Animal1_Bregma_0.06_adata.h5ad
# (https://pan.baidu.com/s/1u6h_9YxFZwhkNgl7U8D9cg?pwd=cube#list/path=%2F), which is
# provided on scCube's GitHub page (under scCube/tutorial/statistics.md).
# The CSV files used below are obtained by reading the h5ad file as an anndata object in
# Python and saving the .X and .obs from the anndata object as CSV files.

data = read_csv("~/scCube/data/merfish066_with_names.csv", col_types = cols(...1 =
col_skip())) # the skipped columns for cell IDs
data = as.data.frame(data)
data = round(data)
covariate <- read_csv("~/scCube/data/merfish066_covariate.csv", col_types = cols(...1 =
col_skip())) # the skipped columns for cell IDs
covariate = as.data.frame(covariate)
rownames(data) = paste0("Cell_", 1:dim(data)[1])

set.seed(123)
overdisps.est = sctransform::vst(t(data))$model_pars[,1]
splitNB.est = countsplit(data, overdisps=overdisps.est)
Xtrain_nb = splitNB.est[[1]]
Xtest_nb = splitNB.est[[2]]

all(colnames(Xtrain_nb) == colnames(Xtest_nb))
cor_all_nb = sapply(colnames(Xtrain_nb), function(x){
  cor(Xtrain_nb[,x], Xtest_nb[,x])
})
names(cor_all_nb) = colnames(Xtrain_nb)
mean(na.omit(cor_all_nb))
genes = c("Gad1", "Mbp", "Nnat", "Aqp4", "Slc17a6", "Fn1", "Pdgfra", "Selplg", "Myh11")
cor_all_nb[genes]

genes1 = genes[1:3]
data_train = data.frame(Xtrain_nb[,genes1]) %>% as_tibble() %>% dplyr::mutate(X =
covariate$x, Y = covariate$y) %>% tidyr::pivot_longer(-c("X", "Y"), names_to = "Gene",
values_to = "Expression") %>% dplyr::mutate(Method = "Train")

```

```
data_test = data.frame(Xtest_nb[,genes1]) %>% as_tibble() %>% dplyr::mutate(X =
covariate$x, Y = covariate$y) %>% tidyr::pivot_longer(-c("X", "Y"), names_to = "Gene",
values_to = "Expression") %>% dplyr::mutate(Method = "Test")
data = bind_rows(data_train, data_test) %>% dplyr::mutate(Method = factor(Method, levels =
c("Train", "Test")))
```

```
row_labels = paste0(genes1,"(PCC = ",round(cor_all_nb[genes1], 3) ,)")
names(row_labels) = genes1
p = data %>% ggplot(aes(x=X, y=Y, color = Expression)) + geom_point(size=0.5) +
facet_grid(Method ~ Gene, labeller = labeller(Gene = row_labels)) + viridis::scale_color_viridis()
+ coord_fixed()
p
```

```
cowplot::ggsave2("~/scCube/data/genes_pattern.png")
```

```
genes2 = genes[4:9]
data_train2 = data.frame(Xtrain_nb[,genes2]) %>% as_tibble() %>% dplyr::mutate(X =
covariate$x, Y = covariate$y) %>% tidyr::pivot_longer(-c("X", "Y"), names_to = "Gene",
values_to = "Expression") %>% dplyr::mutate(Method = "Train")
data_test2 = data.frame(Xtest_nb[,genes2]) %>% as_tibble() %>% dplyr::mutate(X =
covariate$x, Y = covariate$y) %>% tidyr::pivot_longer(-c("X", "Y"), names_to = "Gene",
values_to = "Expression") %>% dplyr::mutate(Method = "Test")
data2 = bind_rows(data_train2, data_test2) %>% dplyr::mutate(Method = factor(Method, levels
= c("Train", "Test")))
```

```
row_labels2 = paste0(genes2,"(PCC = ",round(cor_all_nb[genes2], 3) ,)")
names(row_labels2) = genes2
p = data2 %>% ggplot(aes(x=X, y=Y, color = Expression)) + geom_point(size=0.5) +
facet_grid(Method ~ Gene, labeller = labeller(Gene = row_labels2)) +
viridis::scale_color_viridis() + coord_fixed()
p
```

Reviewer #3 (Remarks to the Author):

The author offers thorough experiments and analyses. All my concerns have been addressed.

Response to reviewers

Overview of Changes

We sincerely appreciate the reviewers' constructive comments and positive feedback. Our work has been much improved based on their valuable suggestions. We have tried our best to address the concerns raised by the reviewers point by point. Compared to the original version, the revised manuscript adds additional experiments to demonstrate the biological interpretability of scCube-simulated data, utilizes more robust and informative evaluation metrics, and adds substantial benchmark results to evaluate the overfitting issue of scCube. We hope this edition will satisfy the reviewers and address all raised concerns to win their approval for the publication of our manuscript. Please find detailed responses below.

Reviewer #1 (Remarks to the Author):

Reviewer's summary

The manuscript has significantly improved and has addressed the majority of the previous comments and concerns. However, there are still two minor concerns that need to be addressed before it can be published:

Response: Thanks for your positive comments.

1. [My previous comment 4] I think the authors didn't understand my point of view. Now I clarify it more clearly and I hope the authors will address it in this revision:

While the authors provide a detailed explanation, their argument relies on the assumption that scCube simulated data accurately mimics real-world ST data. Therefore, it is crucial for them to substantiate this assumption. As a suggestion, I propose that the authors consider selecting MERFISH data as an example. MERFISH data encompasses both cell annotations and spatial information simultaneously, allowing the creation of a real-world resampled ST dataset with ground truth. By comparing benchmark results between the real-world resampled ST data and scCube-simulated data, we can assess the similarity between the two. Consistent results would

indicate that scCube-simulated data closely resembles real data.

The objective of my suggestion is not to determine which deconvolution methods are superior but to leverage these methods to demonstrate that data simulated by scCube is biologically more meaningful. If the authors fail to demonstrate the superiority of scCube, the question arises: why choose scCube for simulating data in the deconvolution benchmark? It's noteworthy that in many deconvolution algorithm papers, authors often randomly simulate ST spots using scRNA-seq data without assessing whether their simulated data aligns with real-world ST data.

Response: Thanks for your professional comments. We are very sorry that we didn't understand your point of view in the last revision. In this revision, as you suggested, we first aggregated the MERFISH data (Bregma: +0.06) as the SRT data to be deconvolved, and treated this SRT data as the real-world resampled SRT dataset with ground truth. Next, based on this SRT dataset, we further generated the simulated SRT dataset by scCube. We also generated another simulated SRT dataset as a control using the random strategy (**Figure R1a**). The MERFISH data from another slice (Bregma: -0.29) was utilized as the scRNA-seq reference.

We ran nine spot deconvolution methods on the real-world resampled SRT dataset, the scCube-simulated SRT dataset, and the random-simulated SRT dataset, respectively. As shown in **Figure R1b**, compared with the random-simulated SRT dataset, the deconvolution results of each spot were more similar between the scCube-simulated SRT dataset and real-world resampled SRT dataset, reaching higher Pearson correlation coefficient values. We further compared the benchmark results of all methods between the real-world and simulated data, and the result also showed that the benchmark results of all methods between the scCube-simulated SRT dataset and the real-world resampled SRT dataset were more similar regardless of the benchmarking metrics utilized (**Figure R1c**), suggesting that the scCube-simulated SRT data is biologically more meaningful compared with the random strategy. The results have been included in the supplementary materials.

Figure R1. Comparison of benchmark results of spot deconvolution methods between the real and simulated data. **a**, Schematic of benchmarking spot deconvolution methods. The real-world resampled SRT dataset is aggregated from the MERFISH data (Bregma: +0.06). Next, based on this SRT dataset, two simulated SRT datasets are generated by scCube and the random strategy, respectively. The random-simulated dataset is generated by combining corresponding number and types of cells randomly selected from the MERFISH (Bregma: +0.06) data according to the ground truth of the composition of cells in each spot. The MERFISH data from another slice (Bregma: -0.29) is utilized as the scRNA-seq reference. **b**, The comparison of the deconvolution results of each spot by each method between the real-world and simulated data. **c**, The comparison of the benchmarking results of each method between the real-world and simulated data. Four different benchmarking metrics were utilized.

2. [My previous comment 3] I have a straightforward suggestion regarding calibration: the authors can set a counts/normalized count threshold and reset counts below this threshold to zero. The specific threshold can be defined based on the characteristics of your ST platform.

Response: Thanks for your constructive comments. We have added a new parameter named 'count_threshold' in the 'generate_spot_data_random' and 'generate_image_data_random' functions (**Figure R2**). The default value of 'count_threshold' is 'None', in which case scCube will not perform any calibrations on the simulated data. In addition to this, if users set a specific value of 'count_threshold', scCube will reset the expression values below this threshold to zero to better match the sparsity of the SRT data generated by different sequencing platforms. The codes on GitHub have been updated.

```

def generate_spot_data_random(solv):
    generate_sc_data: DataFrame,
    generate_sc_meta: DataFrame,
    platform: str = '10x',
    gene_type: str = 'whole',
    min_cell: int = 10,
    n_gene: Optional[int] = None,
    n_cell: int = 10,
    count_threshold: Optional(float) = None):
    """
    generate spot-based data
    :param generate_sc_data: DataFrame of generated sc data
    :param generate_sc_meta: DataFrame of generated sc meta
    :param platform: only works when '10x_spot4v2', '10x' -- square neighborhood structure;
    '10xhd' -- hexagonal neighborhood structure; '10xhd' -- random neighborhood structure
    :param gene_type: the type of genes to generate, 'whole' -- the whole genes;
    'hvg' -- the highly variable genes; 'marker' -- the marker genes of each cell type;
    'random' -- the randomly selected genes
    :param min_cell: filter the genes expressed in fewer than 'min_cell' cells before selected genes,
    only works when 'gene_type'='random', 'hvg', or 'marker'
    :param n_gene: the number of genes to select, only works when 'gene_type'='random', 'hvg', or 'marker'
    :param n_cell: the mean number of cells per spot
    :param count_threshold: sparsity calibration, reset the expression values below this threshold to zero
    returns: st_data, st_meta, st_index
    """

```

```

def generate_image_data_random(solv):
    generate_sc_data: DataFrame,
    generate_sc_meta: DataFrame,
    gene_type: str = 'whole',
    min_cell: int = 10,
    n_gene: Optional[int] = None,
    count_threshold: Optional(float) = None):
    """
    generate image-based data
    :param generate_sc_data: DataFrame of generated sc data
    :param generate_sc_meta: DataFrame of generated sc meta
    :param gene_type: the type of genes to generate, 'whole' -- the whole genes;
    'hvg' -- the highly variable genes; 'marker' -- the marker genes of each cell type;
    'random' -- the randomly selected genes
    :param min_cell: filter the genes expressed in fewer than 'min_cell' cells before selected genes,
    only works when 'gene_type'='random', 'hvg', or 'marker'
    :param n_gene: the number of genes to select, only works when 'gene_type'='random', 'hvg', or 'marker'
    :param count_threshold: sparsity calibration, reset the expression values below this threshold to zero
    returns: st_data, st_meta, st_index
    """

```

Figure R2. The newly-added parameter 'count_threshold' in the 'generate_spot_data_random' and 'generate_image_data_random' functions.

Reviewer #2 (Remarks to the Author):

Reviewer's summary

Our major concern is that the comparison results in the manuscript are not informative nor fair. There are two major reasons. First, the evaluation favors overfitting of the training data, which is treated as the ground truth by the proposed simulator scCube and an existing simulator SRTsim. Hence, these two simulators far outperformed the other simulators that do not assume the training data to be ground truths. Second, the Pearson correlation coefficient is not an informative metric because it has a low value even between two splitted datasets with the same underlying spatial pattern (if random noise is removed). The first reason also made us wonder how scCube and SRTsim can be useful because if users want to use training data as the ground truth, they can just use the training data instead of the simulated data.

Response: We thank the reviewer for the professional comments. We have provided more experimental results to make sure the comparisons in the manuscript are informative and fair. All significant modifications are marked in blue in the revised manuscript. We hope this edition will address your concerns.

1. Major

a. It is unclear how the proposed simulator scCube can assist method benchmarking. In the comparison results, scCube and a previous simulator SRTsim performed way better than other simulators. The reason is that only scCube and SRTsim assign simulated single cells to spatial locations by mimicking real spatial data, assuming that the real data is noiseless. In other words, scCube and SRTsim treats real spatial data as the ground truth. Then it is expected that the simulated spatial data well mimic the real spatial data. However, this approach has the overfitting issue. We verified this issue by a data splitting experiment (code at the bottom of this review). In the experiment, we split a real spatial dataset into two datasets using a statistical method countsplit ([Neufeld et al., Biostatistics, 2022]), so the two datasets are expected to have the same true gene expression. However, when we calculated the Pearson

correlation coefficient (PCC), the major evaluation metric used in the scCube manuscript, the PCC between the two datasets was extremely low (less than 0.1), even though the spatial patterns of these two samples resemble each other (see the figures at the bottom of this review). This indicates that the high PCC values obtained by SRTsim and scCube were due to overfitting; that is, the simulated spatial data resembles the training data even more than the test data does.

Response: Thanks for your professional comments and we apologize for the lack of evaluation of the overfitting issue in the simulator benchmarking. We fully agree with you that the overfitting issue should be considered. To achieve this, we compared scCube with the other two SRT simulators, SRTsim and scDesign3, in the following three scenarios:

- 1) We split a real SRT dataset into two datasets using the statistical method `countsplit` (Neufeld et al., *Biostatistics*, 2024) as you mentioned, one as the training dataset and the other as the test dataset;
- 2) We split a real SRT dataset into two datasets using the random splitting strategy, one as the training dataset and the other as the test dataset;
- 3) We selected two tissue slices from the same samples or experiments, one as the training dataset and the other as the test dataset:
 - a) the mouse hypothalamus MERFISH datasets (same sample): training datasets – Bregma: +0.06; test datasets – Bregma: -0.29;
 - b) the human DLPFC 10X Visium datasets (same experiment): training datasets – Slice 151507; test datasets – Slice 151676.

Each simulator was trained on the training data, and the simulated data generated by different simulators was subsequently compared with the test data for overfitting evaluation.

Additionally, based on your comments, we didn't directly compare the PCC values between the gene expression vector for each gene across spatial positions in real and simulated data in this revision. As an alternative, we compared supervised learners trained on real data and simulated data separately to evaluate the similarity of spatial patterns between the two data (Song et al., *Nat Biotechnol*, 2023). Specifically, for every gene, we first treated it as the outcome and trained a generalized boosted regression model (GBM) on the real data and simulated data separately to predict the

gene expression values from the spatial locations. We then compared the PCC values between the two GBMs' predicted gene expression values from the simulated data's spatial locations. A high PCC value means that the two GBMs are similar, i.e., the spatial patterns between the real data and simulated data are similar. Compared with the direct comparison of the PCC value of the gene expression vectors across spatial positions between the real data and simulated data, this alternative evaluation metric is more robust and informative (**Figure R3**).

Figure R3. Comparison of two evaluation metrics for spatial patterns. The training dataset and test dataset is split from the mouse hypothalamus MERFISH data (Bregma: +0.06) using countsplit. PCC_{GEV} : the Pearson correlation coefficient values between the gene expression vector for each gene across spatial positions in real and simulated data. PCC_{GBM} : the Pearson correlation coefficient values between the two GBMs' predicted gene expression values from the simulated data's spatial locations.

Figure R4. Boxplots of PCC values between the two generalized boosted regression models' predicted gene expression values from the simulated data's spatial locations across seven benchmark datasets (scenario 1). The two generalized boosted regression models are trained on the real data and the simulated data generated by scCube and other two SRT simulators separately. The simulation results for all genes of scDesign3 are not provided in the Stereo-seq data, 10X Visium, and ST data due to speed constraints of the training step.

As shown in **Figure R4** above, in the first scenario, scCube still outperformed other SRT simulators across most of the benchmark datasets split by countsplit (except for the human BC 10X Xenium dataset). We also demonstrated the spatial expression patterns of several representative marker genes in the training data, test data (ground truth), and simulated data generated by different SRT simulators, respectively. As shown in **Figure R5-11**, all three SRT simulators accurately preserved the spatial expression patterns of these genes across seven different benchmark datasets. These results suggested that scCube and SRTsim do not have significant overfitting issues in this scenario.

Figure R5. Performance comparison of scCube with other two SRT simulators over the mouse hypothalamus MERFISH (Bregma: +0.06) dataset split by countsplit.

Figure R6. Performance comparison of scCube with other two SRT simulators over the mouse V1 neocortex STARmap (Replicate 1) dataset split by countsplit.

Figure R7. Performance comparison of scCube with other two SRT simulators over the human breast cancer (BC) 10X Xenium (Sample 1 Replicate 1) dataset split by countsplit.

Figure R8. Performance comparison of scCube with other two SRT simulators over the zebrafish embryo Stereo-seq (24-hpf) dataset split by countsplit.

Figure R9. Performance comparison of scCube with other two SRT simulators over the human DLPFC 10X Visium (Slice 151507) dataset split by countsplit.

Figure R10. Performance comparison of scCube with other two SRT simulators over the HER2-positive breast cancer (BC) ST (Patient G Replicate 1) dataset split by countsplit.

Figure R11. Performance comparison of scCube with other two SRT simulators over the skin squamous cell carcinoma (SCC) ST (Patient 2 Replicate 1) dataset split by countsplit.

Figure R12. Boxplots of PCC values between the two generalized boosted regression models' predicted gene expression values from the simulated data's spatial locations across seven benchmark datasets (scenario 2). The two generalized boosted regression models are trained on the real data and the simulated data generated by scCube and SRTsim separately. The simulation results of scDesign3 are not provided since it is unable to simulate the SRT data with new spatial locations.

The comparison results of scCube and SRTsim in the second scenario are illustrated in **Figure R12** above, which indicates that scCube achieved better simulation performance on all seven different benchmark datasets compared with SRTsim. Furthermore, both scCube and SRTsim still relatively well preserved the spatial expression patterns of several representative marker genes across different benchmark datasets split by the random splitting strategy (**Figure R13-19**).

Figure R13. Performance comparison of scCube with SRTsim over the mouse V1 neocortex STARmap (Replicate 1) dataset split by the random splitting strategy.

Figure R14. Performance comparison of scCube with SRTsim over the mouse hypothalamus MERFISH (Bregma: +0.06) dataset split by the random splitting strategy.

Figure R15. Performance comparison of scCube with SRTsim over the human breast cancer (BC) 10X Xenium (Sample 1 Replicate 1) dataset split by the random splitting strategy.

Figure R16. Performance comparison of scCube with SRTsim over the zebrafish embryo Stereo-seq (24-hpf) dataset split by the random splitting strategy.

Figure R17. Performance comparison of scCube with SRTsim over the human DLPFC 10X Visium (Slice 151507) dataset split by the random splitting strategy.

Figure R18. Performance comparison of scCube with SRTsim over the HER2-positive breast cancer (BC) ST (Patient G Replicate 1) dataset split by the random splitting strategy.

Figure R19. Performance comparison of scCube with SRTsim over the skin squamous cell carcinoma (SCC) ST (Patient 2 Replicate 1) dataset split by the random splitting strategy.

Figure R20. Boxplots of PCC values between the two generalized boosted regression models' predicted gene expression values from the simulated data's spatial locations across two benchmark datasets (scenario 3). The two generalized boosted regression models are trained on the real data and the simulated data generated by scCube and SRTsim separately. The simulation results of scDesign3 are not provided since it is unable to simulate the SRT data with new spatial locations.

In the third scenario, we considered a more general case where the training data and the test data come from different tissue slices of the same sample or experiment, rather than just being split from the same datasets. As shown in **Figure R20** above, scCube still accurately simulated the spatial expression patterns of each gene in the test datasets. In contrast, SRTsim suffered from significant overfitting, with the median PCC values between the two GBMs' predicted gene expression values from the simulated data's spatial locations approaching zero. The results of spatial expression patterns of several representative marker genes in the simulated data generated by scCube and SRTsim also showed that SRTsim failed to preserve these spatial expression patterns when simulating SRT data with a large discrepancy in tissue shape or cell type spatial distribution from the spatial reference due to the heavy influence of the spatial reference, while scCube avoided such overfitting issue (**Figure R21-22**).

In summary, the results of all three different scenarios above indicated that scCube does not have the obvious overfitting issue. Relevant analysis and discussion have been included in the revised manuscript.

Figure R21. Performance comparison of scCube with SRTsim over the mouse hypothalamus MERFISH (Bregma: -0.29) dataset using the mouse hypothalamus MERFISH (Bregma: +0.06) dataset as the spatial reference.

Figure R22. Performance comparison of scCube with SRTsim over the human DLPFC 10X Visium (Slice 151676) dataset using the human DLPFC 10X Visium (Slice 151507) dataset as the spatial reference.

b. Because of this overfitting issue, it is unclear why we need simulators like scCube and SRTsim, because they do not provide any more ground truths beyond the real spatial data they are trained on. The fundamental question is: how can scCube assist method benchmarking better than using real data? If real data is considered as the ground truth, why do we need simulation?

Response: Thanks for your professional comments. As answered in Question a, our results suggested that scCube does not exhibit obvious overfitting issues. In addition, we would like to clarify that, in our opinion, one of the fundamental functionalities of an SRT simulator should be the ability to accurately simulate real SRT data. Therefore, by considering the real data as the ground truth and comparing the simulated data with the ground truth in simulator benchmarking, we could quantitatively evaluate the simulation performance of different simulators. However, this does not mean that

scCube can only generate the same simulated data as the training data in practice. Actually, scCube is a general simulation framework and can flexibly generate diverse simulated SRT data from the real SRT or scRNA-seq data through reference-based or reference-free strategies. Moreover, in the reference-based simulation, scCube also does not treat the real SRT data as the ground truth but rather as the spatial reference. We provided detailed demonstrations in the following examples.

1. Simulating SRT data by the reference-based strategy with scCube

In the first example, we trained scCube on the mouse hypothalamus (Bregma: +0.06) MERFISH dataset, and then utilized other slices sequentially as the spatial reference to simulate SRT data of other slices by the reference-based strategy with scCube. As illustrated in **Figure R23**, scCube accurately simulated the spatial expression patterns of genes in other tissue slices beyond the SRT data (Bregma: +0.06) that was trained on.

Figure R23. Simulating SRT data of different slices of the mouse hypothalamus by the reference-based strategy with scCube. The model is trained on the mouse hypothalamus (Bregma: +0.06) MERFISH dataset and other slices are utilized sequentially as the spatial reference.

In the second example, we trained scCube on the mouse neocortex scRNA-seq dataset (Tasic et al., Nature, 2018), and then utilized the mouse V1 neocortex STARmap (Replicate 1) dataset as the spatial reference to simulate SRT data by the reference-based strategy with scCube. The STARmap dataset was filtered to exclude two cell types (HPC and Reln) that do not exist in the scRNA-seq dataset. As illustrated in **Figure R24**, scCube accurately simulated the spatial expression patterns of genes already measured in the spatial reference and additionally provided the spatial expression patterns of novel genes beyond the targeted RNA species.

Figure R24. Simulating SRT data using the mouse neocortex scRNA-seq data by the reference-based strategy with scCube. The model is trained on the mouse neocortex scRNA-seq dataset and the mouse V1 neocortex STARmap (Replicate 1) dataset is utilized as the spatial reference.

2. Simulating SRT data by the reference-free strategy with scCube

In the third example, we demonstrated how to simulate SRT data by the reference-free strategy with scCube. Specifically, we applied scCube to the simulation of the tumor-immune microenvironment of 3 triple negative breast cancer (TNBC) patients, which corresponded to three archetypical subtypes of tumor-immune interactions: cold, mixed, and compartmentalized, respectively (**Figure R25a**). As shown in **Figure R25a**, the spatial patterns simulated by scCube closely resembled the corresponding tumor-immune microenvironments. Furthermore, scCube also combines this customized reference-free spatial pattern simulation step with the gene expression simulation step to generate the corresponding gene expression profiles that match the simulated spatial patterns, which provides more information about the spatial expression of external genes compared with the original Multiplexed Imaging (MIBI) images (**Figure R25b-c**).

The detailed description of the reference-free strategy of scCube can also be found in Line 248-402 in the revised manuscript.

Figure R25. Simulating SRT data of different slices of the mouse hypothalamus by the reference-free strategy with scCube. a, The real and scCube-simulated tumor-immune microenvironments of mixed, compartmentalized, and cold subtypes of TNBC. **b and c,** The spatial expression patterns of immune (**b**) and tumor (**c**) marker genes in the scCube-simulated SRT data.

c. For the above reason, among the simulators compared in the manuscript, only SRTsim is comparable to scCube because they both treat the real spatial data as the ground truths and aim to mimic the training data as much as possible (the overfitting issue). As our experiment indicates, data splitting should be used to evaluate the overfitting issue, and any evaluation metric should be applied to the test data for a fair comparison.

Response: Thanks for your professional comments. As answered in Question a, we have fully evaluated the overfitting issue in three scenarios. In addition to two data splitting methods (countsplit in scenario 1, and the random splitting strategy in scenario 2), we further considered a more general case where the training data and the test data come from different tissue slices of the same sample or experiment, rather than just being split from the same datasets (scenario 3). Each simulator was trained on the training data, and the simulated data generated by different simulators was subsequently compared with the test data for overfitting evaluation. Both the original and new metrics were applied to the test data for a fair comparison. All the results suggested that scCube does not exhibit any obvious overfitting issues.

Furthermore, we would like to clarify that scCube does not treat the real SRT data as the ground truth but rather as the spatial reference. The purpose of this strategy is not to mimic the spatial reference as much as possible, but rather to learn the underlying spatial patterns present in the spatial reference and subsequently simulate realistic SRT data based on these patterns. As a result, this approach does not lead to severe overfitting issues.

d. Moreover, the PCC is not a suitable evaluation metric because it is well known to be sensitive to outlying values. Also, our experiment above shows that the PCC is not informative because it is so low even between two datasets splitted from one dataset.

Response: Thanks for your professional comments. You're right that the PCC_{GEV} (the Pearson correlation coefficient values between the expression vector for each gene across spatial positions in real and simulated data) is overly stringent to be an informative evaluation metric when the training data and the test data are different. As answered in Question a, we have applied a new evaluation metric PCC_{GBM} (the Pearson correlation coefficient values between the expression value for each gene from the simulated data's spatial locations predicted by the two generalized boosted regression models trained on the real data and simulated data separately) proposed by Song et al. (Song et al., Nat Biotechnol, 2023) to evaluate the simulation performance of each simulator. Compared with the direct comparison of the PCC value of the gene expression vectors across spatial positions between the real data and simulated data, this alternative evaluation metric is more robust and informative (**Figure R3** above).

e. The implementation of baseline simulators is unfair. We examined the code used by the authors for generating synthetic data using scCube and other baseline simulators. We found that the input real data were pre-normalized (and/or log-transformed) and thus unsuitable for methods directly modeling read counts. We deem it necessary for the authors to provide details of input data and the normalization process for each simulator in the comparison.

Response: Thanks for your professional comments. We are sorry that we didn't provide details of input data and the normalization process for each simulator in the

comparison. We would like to clarify first that since the input of scCube requires the normalized data, the .h5ad files we provided on GitHub are all pre-normalized to facilitate direct use by users. However, in the benchmarking experiment, for methods directly modeling read counts, we did not do any pre-processing on the input data. In other words, the input is the raw count data (except for the MERFISH data, which already contains decimals in its raw form). Subsequently, in the performance evaluation process, we first normalized the real data and simulated data generated by these methods separately, and then calculated the PCC values between the gene expression vector for each gene across spatial positions in real and simulated data to be consistent with scCube. Thus, we believe the comparison is fair enough.

We have uploaded the raw count data used in the comparison (<https://pan.baidu.com/s/1yBKFXKNZ2zydFjMdLwDWPA?pwd=cube>). In addition, we also have added the details of input data and the normalization process for each simulator in the comparison in the revised manuscript (See Line 659-660 and Line 824-857 of the Method section in the revised manuscript).

2. Minor

a. Provide an explanation for the presence of NAs in scDesign3's PCC and MAE values in Figure 2.a and Figure R20.

Response: Thanks for your careful comments. The results in Figure 2a and Figure R20 are the average PCC and MAE values of all genes. For scDesign3, we were unable to finish the simulation of all genes in the 10X Visium data (18,094 genes), ST BC data (14,992 genes), ST SCC data (16,772 genes), and Stereo-seq data (12,838 genes) due to speed constraints of the training step. Therefore, we cannot provide the average PCC and MAE values of all genes in the four scDesign3-simulated data. We have added this description to the legend of Figure 2a.

b. In Figure 2.a, the author uses color to indicate rank, which does not accurately reflect the actual differences in PCC. For example, a PCC value of $2.50e-6$ is represented by a slightly darker blue, which could give a misleading impression of its

actual weakness.

Response: Thanks a lot for your kind reminding, we have revised the heatmap in Figure 2a to boxplot, which can be found in Figure 2a-b in the revised manuscript.

c. Line 187 should be “Bregma +0.06” slice.

Response: Thanks a lot for your kind reminding, we have corrected the typo in the revised manuscript.

d. The ground truth provided by scCube is limited to the cell-type level since scCube assigns each spot a cell type when generating synthetic data. We suggest the authors address this limitation in the discussion section.

Response: Thanks for your professional comments. We have discussed this limitation in the discussion section, which can be found in Line 553-561 in the revised manuscript.

Reviewer #3 (Remarks to the Author):

Reviewer's summary

The author offers thorough experiments and analyses. All my concerns have been addressed.

Response: Thanks for your positive comments.

Reviewer #1 (Remarks to the Author):

The authors have addressed all of my concerns.

Reviewer #1 (Remarks on code availability):

The code comes with a README file containing sufficient instructions for installing and running the application. I have tested the code.

Reviewer #2 (Remarks to the Author):

The authors have addressed most of our comments, however, our primary concern regarding the potential usage of scCube stays unsolved (see comment 2 below). In addition, we have the following comments 1 and 3 regarding the evaluation of scCube.

1. When addressing the overfitting issue in R2.1, the authors designed three different scenarios. We agreed that the first two scenarios could provide substantial evidence the scCube does not have the overfitting issue. However, in the third scenario, scCube needs to use both training and testing SRT gene expression data during the training process in order to match the generated gene expression with the testing data spatial coordinates. Thus, using the testing dataset during training created "information leakage", making the performance of scCube biased. Therefore, this third scenario should be fixed.

2. Regarding comment R2.2 (how can scCube assist method benchmarking better than using real data? If real data is considered as the ground truth, why do we need simulation?), the authors' claim that generating synthetic data mimicking real data so that we could quantitatively evaluate the simulation performance of different simulators doesn't make sense to us. In such cases, there are plenty of real datasets that can be used to fulfill such goals. In other words, what would be the fundamental difference between the real data and the synthetic data generated by scCube toward this benchmarking goal? Ultimately, what additional information can the synthetic data of scCube provide that real data cannot?

3. The authors did not include the comparison between different evaluation metrics within the article. It is necessary to assert that "the PCC_GEV is overly stringent to be an informative evaluation metric when the training data and the test data are different" as mentioned in their response. For instance, Figure R3 should be included to demonstrate this point effectively.

Response to reviewers

Reviewer #1 (Remarks to the Author):

Reviewer's summary

The authors have addressed all of my concerns.

Response: Thanks for your positive comments.

Reviewer #2 (Remarks to the Author):

Reviewer's summary

The authors have addressed most of our comments, however, our primary concern regarding the potential usage of scCube stays unsolved (see comment 2 below). In addition, we have the following comments 1 and 3 regarding the evaluation of scCube.

Response: We thank the reviewer for the positive comments.

1. When addressing the overfitting issue in R2.1, the authors designed three different scenarios. We agreed that the first two scenarios could provide substantial evidence the scCube does not have the overfitting issue. However, in the third scenario, scCube needs to use both training and testing SRT gene expression data during the training process in order to match the generated gene expression with the testing data spatial coordinates. Thus, using the testing dataset during training created “information leakage”, making the performance of scCube biased. Therefore, this third scenario should be fixed.

Response: Thanks for your comments. In the third scenario, for the mouse hypothalamus MERFISH datasets, we selected the “Bregma: +0.06” slice as the training dataset and the “Bregma: -0.29” slice as the test dataset, and for the human DLPFC 10X Visium datasets, we selected the “Slice 151507” slice as the training dataset and the “Slice 151676” slice as the test dataset. Thus, the training and test datasets do not share information as they are from different tissue slices.

Figure R1. Workflow of scCube in the third scenario.

In addition, we would like to clarify that the training process of scCube only exists

in the gene expression simulation step. In the third scenario, we only used the training SRT dataset to train the VAE model for generating the gene expression profiles of new cells/spots. The test SRT dataset was only used as the spatial reference in the subsequent reference-based spatial pattern simulation step to construct the mapping between the generated cells/spots and the positions in the spatial reference (**Figure R1** above). Therefore, we believe that there is also no “information leakage” present in scCube in the third scenario.

2. Regarding comment R2.2 (how can scCube assist method benchmarking better than using real data? If real data is considered as the ground truth, why do we need simulation?), the authors’ claim that generating synthetic data mimicking real data so that we could quantitatively evaluate the simulation performance of different simulators doesn’t make sense to us. In such cases, there are plenty of real datasets that can be used to fulfill such goals. In other words, what would be the fundamental difference between the real data and the synthetic data generated by scCube toward this benchmarking goal? Ultimately, what additional information can the synthetic data of scCube provide that real data cannot?

Response: Thanks for your comments. We need to clarify two concepts here, namely “spatial reference” and “ground truth”. The “ground truth” refers to the real SRT data in the simulator benchmarking section. In this context, we simulated the real SRT data with scCube and other simulators respectively, and then quantitatively evaluated the simulation performance of different simulators by considering the real SRT data as the “ground truth”. On the other hand, the “spatial reference” is specifically related to the reference-based simulation of scCube, i.e., scCube does not treat the real SRT data as the ground truth in the simulation, but rather as the spatial reference. The purpose of this strategy is to learn the underlying spatial patterns present in the spatial reference and subsequently simulate realistic SRT data based on these patterns. Therefore, scCube is not designed to mimic the training data as closely as possible.

As responded in the last revision, we provided three examples to demonstrate that the scCube-simulated data can provide the additional information that the real data cannot, thus assisting method benchmarking studies better than only using the real data. We would like to clarify it more clearly in this revision.

1. Increase the size and diversity of the benchmarking datasets

In this example, we only utilized the mouse hypothalamus (Bregma: +0.06) MERFISH dataset to train scCube, thus, the generated gene expression profiles only contain the information from the training dataset (Bregma: +0.06). Subsequently, we sequentially applied other slices (Bregma: +0.11, +0.16, +0.21, +0.26, -0.04, -0.14, -0.19, -0.24, -0.29) as the spatial reference, aiming to simulate corresponding spatial patterns for the generated gene expression data by learning their underlying spatial patterns. As illustrated in **Figure R2**, scCube accurately simulated the spatial expression patterns of genes. On the other hand, due to the provision of additional information that is distinct from both the training data (additional spatial patterns) and the spatial reference (additional gene expression profiles), these scCube-simulated data can serve as supplementary datasets in method benchmarking studies, thereby increasing the size and diversity of benchmark datasets.

Figure R2. Simulating SRT data of different slices of the mouse hypothalamus by the reference-based strategy with scCube. The model is trained on the mouse hypothalamus (Bregma: +0.06) MERFISH dataset and other slices are utilized sequentially as the spatial reference.

2. Provide the additional information that is not feasible to obtain from the real data due to technical limitations

We demonstrated two examples in this section. For the former example, we trained scCube on the mouse neocortex scRNA-seq dataset (Tasic et al., Nature, 2018), and then utilized the mouse V1 neocortex STARmap (Replicate 1) dataset as the spatial reference to simulate SRT data by the reference-based strategy with scCube. As illustrated in **Figure R3**, scCube accurately simulated the spatial expression patterns of genes already measured in the spatial reference and additionally provided the spatial expression patterns of novel genes beyond the targeted RNA species. The scCube-simulated data exhibited a significant increase in the number of genes compared with

the spatial reference (from 996 to 3,715), which helps to better assist benchmarking studies of SRT methods such as spatial cell-cell interaction inference.

Figure R3. Simulating SRT data using the mouse neocortex scRNA-seq data by the reference-based strategy with scCube. The model is trained on the mouse neocortex scRNA-seq dataset and the mouse V1 neocortex STARmap (Replicate 1) dataset is utilized as the spatial reference.

For the latter example, we trained scCube on the human breast cancer scRNA-seq dataset (Wu et al., *Genome Med.*, 2021), and then simulated three archetypical subtypes of the tumor-immune microenvironment of 3 triple negative breast cancer (TNBC) patients by the reference-free strategy with scCube. As shown in **Figure R4**, the spatial patterns simulated by scCube closely resembled the corresponding tumor-immune microenvironments. Furthermore, scCube also combines this customized reference-free spatial pattern simulation step with the gene expression simulation step to generate the corresponding gene expression profiles (17,100 genes) that match the simulated spatial patterns, which provides more information about the spatial expression of external genes compared with the original Multiplexed Imaging (MIBI) images (only 36 proteins, Keren et al., *Cell*, 2018).

In summary, our results indicated that scCube is not designed to mimic the training data as closely as possible, and the scCube-simulated data offer additional information that is not available in the real data, which assists method benchmarking studies better than only using the real data.

Figure R4. Simulating SRT data of different slices of the mouse hypothalamus by the reference-free strategy with scCube. a, The real and scCube-simulate tumor-immune microenvironments of mixed, compartmentalized, and cold subtypes of TNBC. **b and c,** The spatial expression patterns of immune (**b**) and tumor (**c**) marker genes in the scCube-simulated SRT data.

3. The authors did not include the comparison between different evaluation metrics within the article. It is necessary to assert that "the PCC_GEV is overly stringent to be an informative evaluation metric when the training data and the test data are different" as mentioned in their response. For instance, Figure R3 should be included to demonstrate this point effectively.

Response: Thanks for your kind reminding, we have included the comparison between different evaluation metrics within the article, which can be found in Line 851-859 in the revised manuscript and Supplementary Figure S54 in the supplementary information.

Reviewer #1 (Remarks to the Author):

The authors have addressed all my concerns.

Reviewer #2 (Remarks to the Author):

I am afraid that the scCube authors and we have fundamental differences in our definition of ground truths in simulators (see below). Hence, the authors' revision still did not address our fundamental concern: what benchmark studies can scCube be used for?

1. The authors claim that the "ground truth" refers to the real spatially resolved transcriptomic (SRT) data in the simulator benchmarking section. However, we argue that simulators are needed because the real SRT data does not contain the ground truths for benchmarking purposes.

2. We think the ground truths should be the parameters or some properties of the model used for generating the simulated data. The ground truths are needed for any inferential analysis that aims to infer biological patterns from noisy SRT data, including the differentially expressed genes between spatial domains or spatially variable genes.

Reviewer #4 (Remarks to the Author):

The scCube paper discusses two important tasks involving the definition of 'ground truth':

1. the first task is benchmarking different simulators (scCube, SRTsim, scDesign3, etc.), and
2. the second task is benchmarking downstream analysis methods using synthetic data generated by these simulators (such as cell type deconvolution, gene imputation, resolution enhancement, and spatial domain detection).

For the first task, the goal is to generate synthetic data that resembles real SRT datasets. This means the synthetic data should retain the characteristics of the real data, such as gene-wise and location-wise summary statistics (mean, variance, coefficient of variation, sparsity, library size, etc., as used in the SRTsim paper, scCube only compared sparsity with real data); and more directly, the correlation calculated between synthetic and real testing data (such as the PCC_GBM value used in scCube and the scDesign3 paper). In this context, using observed gene expression values to quantify how well simulators can resemble real data looks reasonable. Although overfitting is indeed a concern, the scCube authors designed three scenarios to prove the method's generalizability:

- 1)splitting the real data into training and testing datasets using countsplit;
- 2)splitting the real data into training and testing datasets randomly;
- 3)using two different tissue slices (training and testing) from the same samples/experiments, and the results are reasonable to me.

For the second task, 'ground truth' would be spatial domain labels for benchmarking spatial clustering methods, cell type proportions on low-resolution spatial spots for cell deconvolution, or gene expression values for unmeasured genes in gene imputation, etc. The scCube paper demonstrated its utility in various downstream analysis.

To clarify for the readers, it would be beneficial to more explicitly define ground truth for each task. Additionally, including detailed information about which datasets are used as training and testing data in the first task, as well as how ground truth is determined for the second task (such as spatial domain labels, cell type proportions, and gene expression values of unmeasured genes), would enhance understanding. In the Methods section, instead of merely listing the metrics used for comparison, a deeper explanation of which data vectors are employed and how these evaluation metrics are calculated could significantly improve understanding.

Response to reviewers

Reviewer #1 (Remarks to the Author):

Reviewer's summary

The authors have addressed all of my concerns.

Response: Thanks for your positive comments.

Reviewer 1 thinks that the revised manuscript illustrates that, in certain comparisons, scCube can indeed fulfill benchmarking functions. They suggested that it would be beneficial for ensuring rigor in their research if the authors discuss any limitations associated with scCube in benchmarking functions.

Response: Thanks for your professional comments. We have discussed the potential limitations associated with scCube in benchmarking functions, which can be found in Line 561-571 in the revised manuscript.

Reviewer #2 (Remarks to the Author):

Reviewer's summary

I am afraid that the scCube authors and we have fundamental differences in our definition of ground truths in simulators (see below). Hence, the authors' revision still did not address our fundamental concern: what benchmark studies can scCube be used for?

1. The authors claim that the "ground truth" refers to the real spatially resolved transcriptomic (SRT) data in the simulator benchmarking section. However, we argue that simulators are needed because the real SRT data does not contain the ground truths for benchmarking purposes.

2. We think the ground truths should be the parameters or some properties of the model used for generating the simulated data. The ground truths are needed for any inferential analysis that aims to infer biological patterns from noisy SRT data, including the differentially expressed genes between spatial domains or spatially variable genes.

Response: We appreciate the reviewer for kindly pointing out the differences in the definitions of "ground truth". As summarized by Reviewer #4, there were two important tasks involving the definition of "ground truth" in our previous manuscript:

- (1) the first task is benchmarking scCube and other different simulators. The purpose of this task is to evaluate the reasonableness of simulating real SRT data of each simulator. Therefore, we defined the test data as "ground truth" to quantitatively evaluate the similarity between the synthetic data generated by the simulators and the test data. To avoid ambiguity, we have revised the "ground truth" to "real data" in this task in the revision, and the related figures have also been updated.
- (2) the second task is benchmarking downstream analysis methods using synthetic data generated by scCube. In this task, the "ground truth" was specifically defined as the spatial domain labels (in the spatial domain identification methods benchmarking study), the cell type proportions within spots (in the spot deconvolution methods benchmarking study), and the gene expression values for unmeasured genes (in the gene imputation methods benchmarking study), etc., and we believe the definition of "ground truth" used in this task is consistent with the interpretation provided in your

Comment 2.

In conjunction with the clarifications provided above about the definition of "ground truth", we can now more clearly address your fundamental concern - what additional information can the synthetic data of scCube provide that real data cannot? We provided detailed demonstrations in the following two examples.

1. scCube allows varying the number of genes when generating the SRT data

scCube allows users to generate simulated SRT data with a specified number or type of genes. This simulation strategy enables scCube to be used for benchmarking gene imputation methods, as it provides the "ground truth" of the gene expression values for unmeasured genes that cannot be accessed from the real imaging-based SRT data such as STARmap. As shown in **Figure R1a-b**, the scCube-simulated SRT data contains a variable number of genes, which not only preserves the spatial expression patterns of the genes already measured in the real data, but also simulates the spatial expression patterns of unmeasured genes beyond the target RNA species.

Figure R1. scCube generates simulated SRT data with various gene numbers. a, The scCube-simulated data provides additional "ground truth" of the gene expression values for unmeasured genes compared with the real data. **b,** The number of genes in the real and scCube-simulated data.

2. scCube allows varying the number and size of spatial patterns when generating the SRT data

scCube allows users to generate simulated SRT data with a specified number or size of spatial patterns. This simulation strategy enables scCube to be used for benchmarking spatial domain identification methods, as it provides the additional "ground truth" that can accommodate varied distributions of spatial patterns. In contrast, the real data cannot provide the "ground truth" for spatial patterns of differing numbers or sizes. As shown in **Figure R2a-b**, the scCube-simulated SRT data contains a variable number or size of each domain compared with the real data.

Figure R2. scCube generates simulated SRT data with various spatial pattern numbers and sizes. a, The scCube-simulated data provides additional "ground truth" of the spatial patterns compared with the real data. **b,** The number of spots of each domain in the real and scCube-simulated data.

In addition to the two examples mentioned above, we have also demonstrated in detail the utility of scCube in benchmark studies of SRT methods in our previous manuscript, which can be found in Line 407-544 in the revised manuscript. Taking the benchmark study of spot deconvolution methods as an example, scCube simulated a series of SRT data with different resolutions and provided the "ground truth" of the cell type composition within each spot at different resolutions, which assisted method benchmarking studies better than only using the real data.

In summary, our results indicated that scCube can accommodate complex benchmarking studies. scCube enables generating simulated SRT data in a user-specified manner that meets diverse benchmarking purposes and provides additional "ground truth" that is not present in real data. Relevant results and discussion have been included in the revised manuscript.

"1. What additional information does scCube's simulated data provide in addition to the real SRT data? To us, all the benchmark studies that can be done with scCube's simulated data can also be done with the real SRT data. We think answering this question is crucial to the application of scCube for benchmarking purposes.

Response: Thanks for your comments. As answered above, we have demonstrated that the scCube-simulated data can provide additional "ground truth" that is not present in the real data, which assists method benchmarking studies better than only using the real data.

2. Why does scCube use simulated single-cell data by the Variational Autoencoder (VAE), instead of using real single-cell data? As a two-step procedure, scCube first simulates single-cell data using the VAE and then matches the simulated cells to spatial spots to simulate SRT data. To us, if the goal is just to simulate SRT data to resemble real SRT data, scCube can well use real single-cell data instead of simulated single-cell data in the second step. Hence, simulating single-cell data seems unnecessary."

Response: Thanks for your comments. Specifically, as a deep generative model, VAEs learn a compressed latent space representation of the input data (scRNA-seq or SRT data) which captures the underlying structure and patterns of the original data, rather than merely replicating it. Consequently, users can conveniently generate or augment scRNA-seq or SRT data using the latent space representation learned by VAEs, such as

the conditional generation of specific rare cell populations (Marouf M et al., *Nat Commun*, 2020). As illustrated in **Figure R2b**, in the second example, we used VAEs to generate a specific number of spots for Layer 3, Layer 5, and WM, to match the predefined spatial locations. In contrast, this simulation would be unattainable using the real data, as the number of simulated spatial locations exceeds the number of spots contained in the real data. In summary, using VAEs enables the flexible simulation of SRT data that satisfies diverse experimental design requirements based on accurately modeling real data.

Reviewer #4 (Remarks to the Author):

Reviewer's summary

The scCube paper discusses two important tasks involving the definition of 'ground truth':

1. the first task is benchmarking different simulators (scCube, SRTsim, scDesign3, etc.), and

2. the second task is benchmarking downstream analysis methods using synthetic data generated by these simulators (such as cell type deconvolution, gene imputation, resolution enhancement, and spatial domain detection).

For the first task, the goal is to generate synthetic data that resembles real SRT datasets. This means the synthetic data should retain the characteristics of the real data, such as gene-wise and location-wise summary statistics (mean, variance, coefficient of variation, sparsity, library size, etc., as used in the SRTsim paper, scCube only compared sparsity with real data); and more directly, the correlation calculated between synthetic and real testing data (such as the PCC_GBM value used in scCube and the scDesign3 paper). In this context, using observed gene expression values to quantify how well simulators can resemble real data looks reasonable. Although overfitting is indeed a concern, the scCube authors designed three scenarios to prove the method's generalizability:

1)splitting the real data into training and testing datasets using countsplit;

2)splitting the real data into training and testing datasets randomly;

3)using two different tissue slices (training and testing) from the same samples/experiments, and the results are reasonable to me.

For the second task, 'ground truth' would be spatial domain labels for benchmarking spatial clustering methods, cell type proportions on low-resolution spatial spots for cell deconvolution, or gene expression values for unmeasured genes in gene imputation, etc. The scCube paper demonstrated its utility in various downstream analysis.

Response: We greatly thank the reviewer for the positive comments and professional summary of our work.

To clarify for the readers, it would be beneficial to more explicitly define ground truth for each task. Additionally, including detailed information about which datasets are used as training and testing data in the first task, as well as how ground truth is determined for the second task (such as spatial domain labels, cell type proportions, and gene expression values of unmeasured genes), would enhance understanding. In the Methods section, instead of merely listing the metrics used for comparison, a deeper explanation of which data vectors are employed and how these evaluation metrics are calculated could significantly improve understanding.

Response: Thanks for your professional comments. We have revised the "ground truth" to "real data" in the first task to avoid ambiguity. Additionally, we have defined the "ground truth" for the second task more explicitly and provided detailed information about how the "ground truth" is determined in various downstream analysis, which can be found in Line 412-414, Line 464-470, Line 501-504, and Line 971-973 in the revised manuscript. Detailed information about datasets that were used as training and testing data in the first task can also be found in Supplementary Data 3.

Furthermore, based on your advice, we have also provided deeper explanations and formulas for all the evaluation metrics employed in two tasks, which can be found in Line 859-887, Line 904-929, Line 946-950, Line 957-966, Line 980-1004, and Line 1018-1022 of the Method section in the revised manuscript.

The new Reviewer #4 suggested that to make it clearer to the readers, it might be helpful if the authors could separate and clearly state the definitions of ground truths in the two important tasks mentioned in their report.

Response: Thanks for your comments. As answered above, we have revised the "ground truth" to "real data" in the first task to avoid ambiguity, and we have also provided the clear definitions of the "ground truth" for the second task, which can be found in Line 412-414, Line 464-470, Line 501-504, and Line 971-973 in the revised manuscript.

Reviewer #2 (Remarks to the Author):

We thank the authors for their added text to clarify the ground truths scCube can provide. However, we find the terms such as "spatial components" and "sizes" still a bit confusing. Hence, we suggest the authors use the following phrases for the ground truths scCube can provide:

the number of genes, the numbers and areas of spatial domains, and the cell type composition at each spot

We suggest that the authors use these terms consistently throughout the manuscript to avoid confusion.

Reviewer #4 (Remarks to the Author):

The authors have addressed all of my concerns.

Reviewer #4 (Remarks on code availability):

The tutorial is easy to follow. With enough instructions for installing and running the application.

Response to reviewers

Reviewer #2 (Remarks to the Author):

Reviewer's summary

We thank the authors for their added text to clarify the ground truths scCube can provide. However, we find the terms such as “spatial components” and “sizes” still a bit confusing. Hence, we suggest the authors use the following phrases for the ground truths scCube can provide:

the number of genes, the numbers and areas of spatial domains, and the cell type composition at each spot

We suggest that the authors use these terms consistently throughout the manuscript to avoid confusion.

Response: Thanks for your professional comments. We have used these terms you suggested consistently throughout the manuscript to avoid confusion.

Reviewer #4 (Remarks to the Author):

Reviewer's summary

The authors have addressed all of my concerns.

Response: Thanks for your positive comments.

Reviewer #4 (Remarks on code availability):

Reviewer's summary

The tutorial is easy to follow. With enough instructions for installing and running the application.

Response: Thanks for your positive comments.